# Using an ordinary differential equation model to separate rest and task signals in fMRI

Amrit Kashyap [1] ✉, Eloy Geenjaar[2,3], Patrik Bey [1], Kiret Dhindsa[1], Katharina Glomb[1], Sergey Plis [3], Shella Keilholz [2,4] & Petra Ritter [1]

Cortical activity results from the interplay between network-connected regions that integrate information and stimulus-driven processes originating from sensory motor networks responding to specific tasks. Separating the information due to each of these components has been challenging, and the relationship as measured by fMRI in each of these cases Rest (network) and Task (stimulus-driven) remains a significant open question in the study of large-scale brain dynamics. In this study, we develop a network ordinary differential equation (ODE) model using advanced system identification tools to analyze fMRI data from both rest and task conditions. We demonstrate that task-specific ODEs are essentially a subset of rest-specific ODEs across four different tasks from the Human Connectome Project. By assuming that task activity is a relative complement of rest activity, our model significantly improves predictions of reaction times on a trial-by-trial basis, leading to a 9% increase in explanatory power ($R^2$) across the 14 sub-tasks tested. We have additionally shown that these results hold for predicting missing trials and accuracy on a per individual basis as well as classifying Tasks trajectories or resulting dynamic Task functional connectivity. Our findings establish the principle of the Active Cortex Model, which posits that the cortex is always active and that Rest State encompasses all processes, while certain subsets of processes get elevated to perform specific task computations. Thus, this study is an important milestone in the development of the fMRI equation - to causally link large-scale brain activity, brain structural connectivity, and behavioral variables within a single framework.

One of the main successes of fMRI is measuring noninvasively how the cortex responds to particular stimuli and performs cognitive tasks, i.e., Task fMRI. Over 100,000 Task studies had been performed and analyzed as of 2023 to understand how different components of the nervous system interact to perform specific behavioral tasks and to address a diverse set of cognitive and clinical questions[1]. However, a key challenge in these studies has been characterizing the changes in activity triggered by task onset relative to the large nonstationary baseline cortical activity[2,3]. This baseline activity, also known as Resting State fMRI, has been studied in the absence of stimulus and portions of the activity have been linked to numerous different sources, such as unobserved spontaneous thought processes or introspective processes[4,5]. However, considering the substantial energy demands of activating large neural regions and the persistent rhythmic nature of

[1]Charite University Medizin, Berlin, Germany. [2]Georgia Institute of Technology, Atlanta, US. [3]Georgia State University, Atlanta, US. [4]Emory University, Atlanta, US. ✉e-mail: amrit.kashyap@charite.de

these resting-state processes during Task activation, studies have suggested that these processes may rather play a role in homeostatic functions, such as priming plasticity, maintaining functional interactions between regions, and optimizing generative models for future interactions with the environment[6–8]. To explain Task activation, studies have proposed that certain Resting State sub-networks are functionally activated with a higher weight during Task activation, while others have also shown that sub-networks of the white matter structural connectivity are predictive of Task performance[9,10]. However, there is uncertainty of what occurs to the other Rest processes within Task, as many of these Resting State networks are also shared with the Task networks[11]. Mechanistically, these quasi-rhythmic processes observed in Rest are thought to originate from intrinsic cortical loops that exist in the white matter networks that are stimulated by spontaneous activity and thought to persist during Task processes[12,13]. Determining this exact relationship between Rest and Task would not only further our fundamental understanding of how the nervous system operates but would also have practical consequences in improving our ability to analyze and decode Task fMRI activity.

In the following study, we extend our current generative Ordinary Differential Equations (ODE) models for fMRI activity, and model the components of the Rest signal during the Task activation[14]. Our results reinforce the idea that Task processes are a subset of Rest processes. More importantly, we present a practical approach to isolating Task-specific signals, which enhances correlations with behavioral measures across various tasks. This method is not only applicable to future studies but can also refine analyses of past research.

In the context of fMRI as a measurement modality, the observed system is believed to be represented by stochastic differential equations, where part of the dynamics is constrained by a rigid network structure, while the remainder evolves due to unknown spontaneous inputs originating from unidentified neural sources[15,16]. This approach has led to the development of Brain Network Models (BNMs), which leverage estimates of white matter connectivity, and Neural Field Equations (NFEs) that utilize the cortical surface structure as macro-structures to model Resting State activity[17–19]. To compare the measured signal with the stochastic differential models, noisy inputs are integrated on the basis of the macrostructure, yielding comparable metrics between the model and empirical observations limited to steady-state properties like functional connectivity. However, this approach is poorly suited to address questions about transient dynamics, such as those that occur during task switching that require analysis of a particular trajectory rather than steady-state dynamics[17]. To address this issue, we propose a framework for using ODEs with neural data. Instead of constructing the biophysical model from primitives, we fit an ODE directly to the measured fMRI data to model the network component of the measured signal, enabling us to interpret individual trajectories during Rest and Task epochs with reference to the structured activity. In this manner, we provide a reference dynamics framework for the processes in Rest and Task, enabling the interpretation of single trials based on their structured network component and an unstructured component representing unknown neural activity.

To this aim, we use the Sparse Identification of Nonlinear Dynamics (SINDy), an algorithm that approximates the underlying network ODE from measured timeseries using regression[20]. Our approach bears similarities to one of the earliest methods used for fMRI data analysis, Dynamic Causal Modeling (DCM), which relies on Bayesian inference. However, since SINDy utilizes regression, the algorithm scales much better than DCM while providing stable, consistent solutions, unlike comparable deep learning algorithms that fit an ODE[21,22]. We use SINDy to construct a network model, where each edge $E$ between the brain regions $a$ and $b$ is approximated using a Taylor polynomial of order $p$ with corresponding

coefficients $\Xi_n$: $E_{ab}(X_b) = \sum_{n=1}^{p} \Xi_n X_b^n$, giving us the change in activity with respect to the network neighbors. The derivative of a single region is estimated as a sum across all edges minus the decay rate $\dot{X}_a = \sum E_{ab}(X_b) - \Lambda(X_a)$. The decay function can be estimated from the Hemodynamic Response Function (HRF); however, we demonstrate that it can also be accurately derived through a data-driven approach. The network activity is defined here as the component of the signal that can be explained by the activity of neighboring nodes and can be constructed separately for the Rest and Task datasets to capture the processes specific to each dataset. Overall, ridge regression results in a trade-off between three key outcomes: producing coefficients that reflect a more biologically plausible network when compared to tractography-based estimates, achieving a better fit to the predicted dynamic time series, and generating a realistic Hemodynamic Response Function (HRF) based on the system's impulse response. Ensuring these three criteria during hyperparameter tuning collectively guides the model toward biophysically plausible and accurate predictions.

Using the same hyperparameters, separate network ODEs were trained for Rest and Task fMRI using different datasets from the Human Connectome Project (HCP)—specifically Rest, Gambling (GB), Working Memory (WM), Emotion (EM), and Relational (REL)[23]. This resulted in a distinct model for each dataset, with each model capturing the dominant processes of the dynamical system observed during the respective epochs.

Interpreting a single Task trial using these models in terms of stimulus-dependent and stimulus-independent processes is challenging, as stimulus-related activity may be embedded within the identified network processes of each model and/or reflected in the unstructured residual activity. We hypothesize that stimulus-independent process can be represented by a linear combination of the Rest and Task ODE models, capturing the structured background activity that is unrelated to the performed Task. The candidate background activity is then subtracted from the measured signal and evaluated to determine whether it enhances the isolation of stimulus-dependent processes. To test the separated signal in relation to behavioral outcomes, we used five different behavioral measures (reaction time per trial, trajectory classification per trial, accuracy and missing trials across individual Task, functional connectivity across individual Task) across fourteen sub-tasks to test which separation framework increases the brain-to-behavior relationship. We demonstrate that accurately separating the signal is achievable only under the assumption of the Active Cortex Model, where the Task-specific network ODEs are a subset of the Rest network ODEs, and the background, stimulus-independent activity is approximated by the difference between the Rest and Task ODE models. We show that the Task network activity is specific to each type of Task, but they are all subsets of the Rest ODE network model. This supports the conclusion of previous literature that modeled Task activity as weighted components of functional and structural sub-networks of Rest processes[10,24]. Thus, we conclude that the primary organization of the Rest and Task can be approximated by the Active Cortex Model, aptly named as all processes are active during Rest and Task, but a certain subset are elevated to perform the particular Task.

This work contributes to the field by applying an ODE model with biophysical properties to interpret the Task fMRI signal. To gain real insight into brain dynamics, structure, and behavior, it is best to address it by creating a single framework where all these elements can interact in a causal manner. Only when all three of these properties are represented simultaneously can we hope to build a working model of the cortex in the future, as the space for possible solutions is vast. This is our main contribution: while elements of these have been modeled before, our method provides a unified framework that incorporates all of these components.

## Results

The results section is organized into four subsections. The first subsection introduces how we construct and validate network ODEs to represent network activity of Rest and Task fMRI data. The second subsection shows that during the Task, the stimulus couples with a component of the Rest network. The Task activity can be separated from background processes only by removing the processes present in the Rest network activity but not present in the Task network activity. The third subsection quantifies how much variance is explained by removing the background processes and which brain regions change to reflect the increase in brain-to-behavior correlation. The fourth subsection examines the relationship between different Tasks and shows that by removing the shared background processes, the distinct timeseries trajectories and Task FCs diverge in hyper-dimensional space as the processes common to them are reduced.

### Network model for fMRI

**Structural.** To construct a network model for fMRI, it is assumed that the measured data is composed of both structured network activity and unstructured residual activity, as illustrated in Fig. 1A. The network activity represents structured interactions and is modeled as

relationships between distinct nodes. To construct this network, we fit a separate function for each edge in the network using a Taylor series polynomial $p^n$ via ridge regression and define the change in activity in a node as the sum of all its edges. A sparsity parameter determines the number of edges used to represent the ODE. To test whether these estimated edges are biophysically relevant, we examine the distribution of edge strengths with other measures of network connectivity using Structural Connectivity (SC) estimated from Diffusion Tensor Imaging (DTI). The coefficients are highly correlated (>0.5 on average) for the odd orders of the polynomial for the edges within a hemisphere (intrahemisphere) while the even orders are smaller in magnitude and have low correlations (<0.05 on average) with SC as shown in Fig. 1C. We plotted the resulting odd edge functions using a derivative plot in Fig. 1D. They are divided into four categories: (1) diagonal (self edges), (2) intrahemispheric, (3) contralateral (edges between homologous brain regions), and interhemispheric (excluding contralateral edges) in Fig. 1D. They all consist of sigmoid functions, where the relationship is linear around the origin and plateaus at the extrema. The intrahemispheric and contralateral edges are positive sigmoid functions, but the interhemispheric connections (non-contralateral) have an opposite sign. The main difference between our identified network ODE and

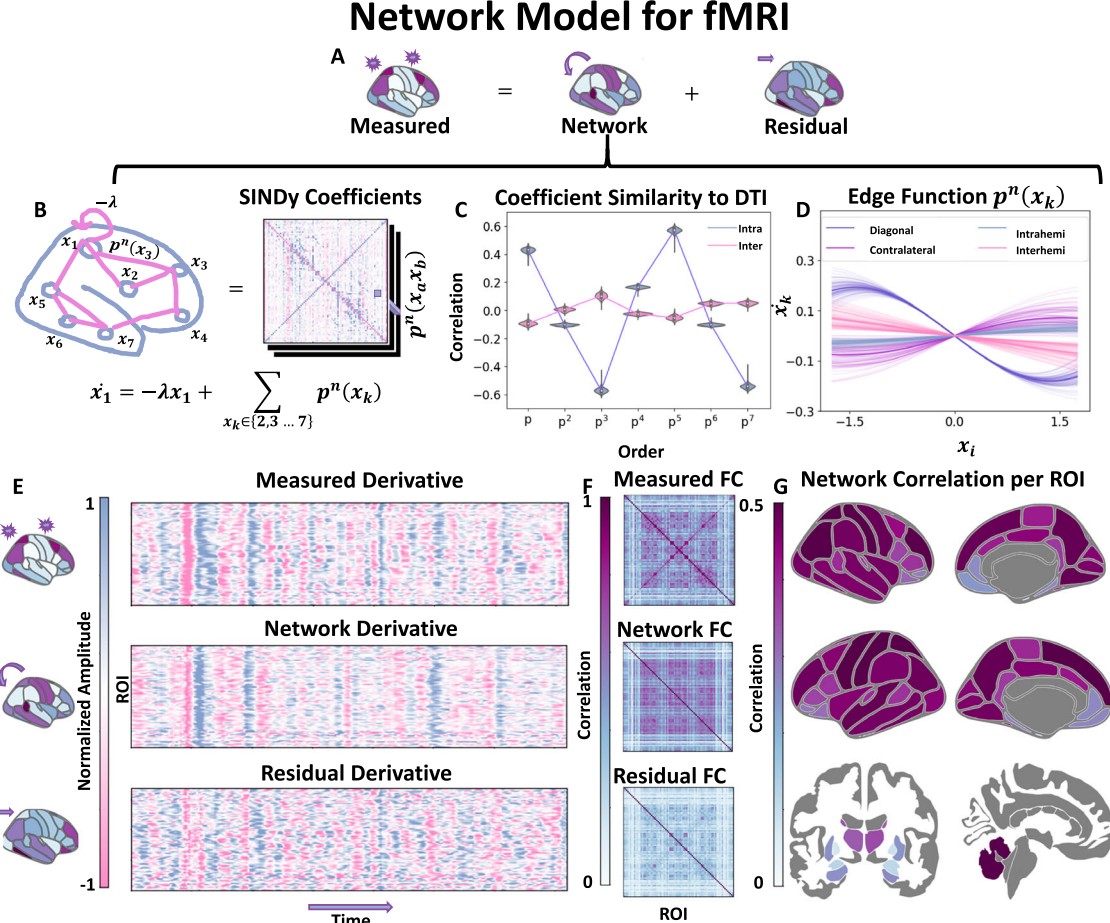

**Fig. 1 | Network model for fMRI. A** A simplified diagram of separating the measured signal into network and unstructured residual components. **B** The network component consists of estimating a Taylor series polynomial between each pair of brain regions estimated from HCP rsfMRI scans ($N = 357$). **C** The correlation of coefficients $\Xi$ for each polynomial order, analyzed separately, with the tractography of interhemispheric and intrahemispheric edges derived from Diffusion Tensor Imaging ($N = 443$). **D** A derivative plot of all the edge functions estimated between pairs of brain regions. **E** An example of separating the measured signal derivative into network and stimulus components, normalized to have the same

amplitude. **F** The resulting average FCs of the measured signal, network, and residual components calculated on unseen HCP individuals ($N = 90$). **G** The correlation between the predicted network and the measured derivative timeseries, computed separately for each brain region calculated on unseen individual HCP individuals ($N = 90$). All the above results are shown for the resting state fMRI ODE, while the task-specific differences are shown in Methods Separating Task. The violin box plot in (**C**) shows the median as a white dot, the interquartile range as a thick bar from the 25th to 75th percentile, and whiskers extending to the most extreme data within $1.5 \times$ IQR of the quartiles. Brain Plots were made using ggseg[75].

previously published BNMs is that our model has negative interhemispheric (non-contralateral) edges, while previously, all positive functions were used for edges to describe the network structure[13,14,17]. The diagonal represents the decay term, and is a hyperparameter in our model, and affects the HRF (see Methods Sparse Identification of Nonlinear Dynamics). Setting values for this parameter, as well as other hyperparameters that control the sparsity of the overall network, are discussed in detail in Methods Optimization and Hyperparameters The identified network ODE depends on the data used for the ridge regression and is unique for the different Tasks and Rest (see Methods Differences between Task Models). In general, they exhibit similar properties as they all consist of sigmoidal edges with the relation shown above and are similar to previously used BNM constructed from *SC*, with the exception of interhemispheric inhibition due to the negative interhemispheric edge functions.

**Dynamics**. The algorithm minimizes the unstructured residual dynamics, capturing the remaining activity through the network model. The separation of the signal into structured and unstructured components in the derivative space is illustrated in Fig. 1E and has been normalized for illustration purposes. At each time point in the measured ROI activity, we can compute the derivative using our ODE model of network activity and subtract the measured derivative to separate the signal. The model captures the large features of the derivative of the fMRI signal, which can be visually observed by the FC network as shown in Fig. 1F. The residual signal has a lower FC between the brain regions (mean 0.118) compared to those seen in the measured FC (mean 0.218) or by the predicted network FC (mean 0.216). We also quantified how similar the predicted network timeseries derivative and the measured timeseries derivative on ROI basis are in Fig. 1G. The median correlation between the network and the measured timeseries is >0.38 across the ROIs. Only a few brain regions known to have poor signal, such as the small structures of the basal ganglia and the temporal and frontal poles, have low correlations (0.2) with our network predictions. The ODE model is also specific to the type of data it is trained on, as in the model trained on Rest fMRI, it is more predictive on Rest dynamics than on Task dynamics (see the Methods Differences between Task Models). The only requirement needed to construct a network model is the amount of data needed to train the dataset (see Supplementary Fig. 3). In this manner, we construct separate network models for the Rest dataset and each of the four respective Task datasets. The separation illustrated here serves as a simplified demonstration because it is not clear if the residuals contain stimulus information, or if the stimulus information is a component of the network or a mix of both. For that, we turn to the following experiment.

**Active cortex model**
In order to isolate the Task component from the measured signal, we consider the following three plausible frameworks describing the relationship between Rest, Task and stimulus activity as shown in Fig. 2A. All frameworks subtract stimulus-unrelated activity from the measured signal, but differ in how this activity is estimated, as follows: (1) The Task Baseline Model, where the unrelated activity is the Task specific Network ODE, (2) The Rest Baseline Model, where the unrelated activity is the Rest Network ODE, (3) The Active Cortex Model (ACM), where the stimulus partially overlaps with the Rest Network ODE and results in the Task Network ODE. In the ACM, the stimulus unrelated activity is defined by the processes present in the Rest Network ODE but not in the Task-specific Network ODE, i.e., $Rest - Task_a$. The term $Task_a$ here does not refer to a single model, but the distinct trained model using a particular Task fMRI dataset (one for each of the four meta Tasks).

To determine which framework best fits the data, the three different signal separation strategies are applied for each trial and

compared with the corresponding RT. Following the stimulus onset, the subsequent next 14.4 seconds are temporally aligned by resampling and separated based on the framework to obtain the stimulus-dependent candidate activity for each trial (see Methods in Behavior Processing). In Fig. 2B, the mean HRF for the ROI with the largest correlation to RT during the WM Task, is plotted for the unseparated measured signal as well as for each of the separation frameworks. Of the three frameworks, only the ACM results in a larger HRF amplitude and a corresponding increase in correlation to RT compared to the unseparated case. In Fig. 2C, the spatial-temporal points that are within 90 percent of the max correlation with RT are plotted for the unseparated and the three different strategies. ACM again is the only tested framework that has larger correlations after separation and shows a significant increase (student $t$ test) compared to the unseparated signal for all of the sub-tasks. This increase in correlation is not due to a decrease in variance in the separated signal, as the ACM actually increases the variance in the signal (see Supplemental Figure 7). While changes shown here are small as they correspond to changes in individual ROIs, in the following section, we show that these changes are much larger when we aggregate all the small changes to each individual ROIs. Moreover, we show in Supplemental Figure 12 that these results generalize to predicting other behavioral variables, namely accuracy and missing trials, where the ACM, unlike the other separation strategies, shows a significant increase with respect to the original signal.

The association between the Rest and Task models is generalized in Fig. 2D, where linear combinations between Rest and Task are considered to model the stimulus-independent activity: $Stim = Meas - (c_1 Rest - c_2 Task)$ (see Methods in Signal Separation In Fig. 2D, the changes to the maximum correlation value for each Task is plotted averaged across the sub-tasks. The horizontal line on the plot represents a scaled version of the Task Baseline Model, and the vertical line a scaled version of the Rest Baseline Model, and the diagonal a scaled version of the ACM. We clearly see in WM, EM and REL that the association between stimulus and the network activity is given by the ACM as seen by the diagonal streak, where the correlation increases due to separation. In the case of GB, the trend seems to be more washed out, but the largest changes to correlation is observed just off the main diagonal.

**Brain to behavior function**
The last section showed that only the linear separation model that isolated the stimulus component of the signal was the ACM. In this section, we examine where these changes occur in the cortex for each specific Task and quantify the overall changes to the brain-behavior relationship by aggregating the spatiotemporal information across all ROIs. In Fig. 3A Left, we can observe that specific ROIs are moderately correlated with each unique Task. The $y$-axis refers to the maximum correlation computed over time for each individual ROI. These areas are very similar to the previously identified areas as shown in Supplemental Figure 6[25-28]. The ACM increases the correlation within a subset of these regions, as can be seen in Fig. 3A Right. The change in correlation is normalized by the initial magnitude of the baseline correlation of that specific region. The regions that show the largest change with respect to RT are more associated with regions that depend on the difficulty of the Task, rather than the visual cortex or motor cortex, which are necessary for Task completion. For example, in Supplementary Fig. 11, we show for WM that these regions are associated with prefrontal regions and align more closely with Task severity than the original areas identified for the completion of the overall Task.

To aggregate all this spatial-temporal information for each trial, we utilized an Elastic Net to map the spatial-temporal brain activity to a single RT. Our results are shown in 3B and C using a 10 times cross-validated scheme on test data. The ACM improves the relationship to

# Active Cortex Model

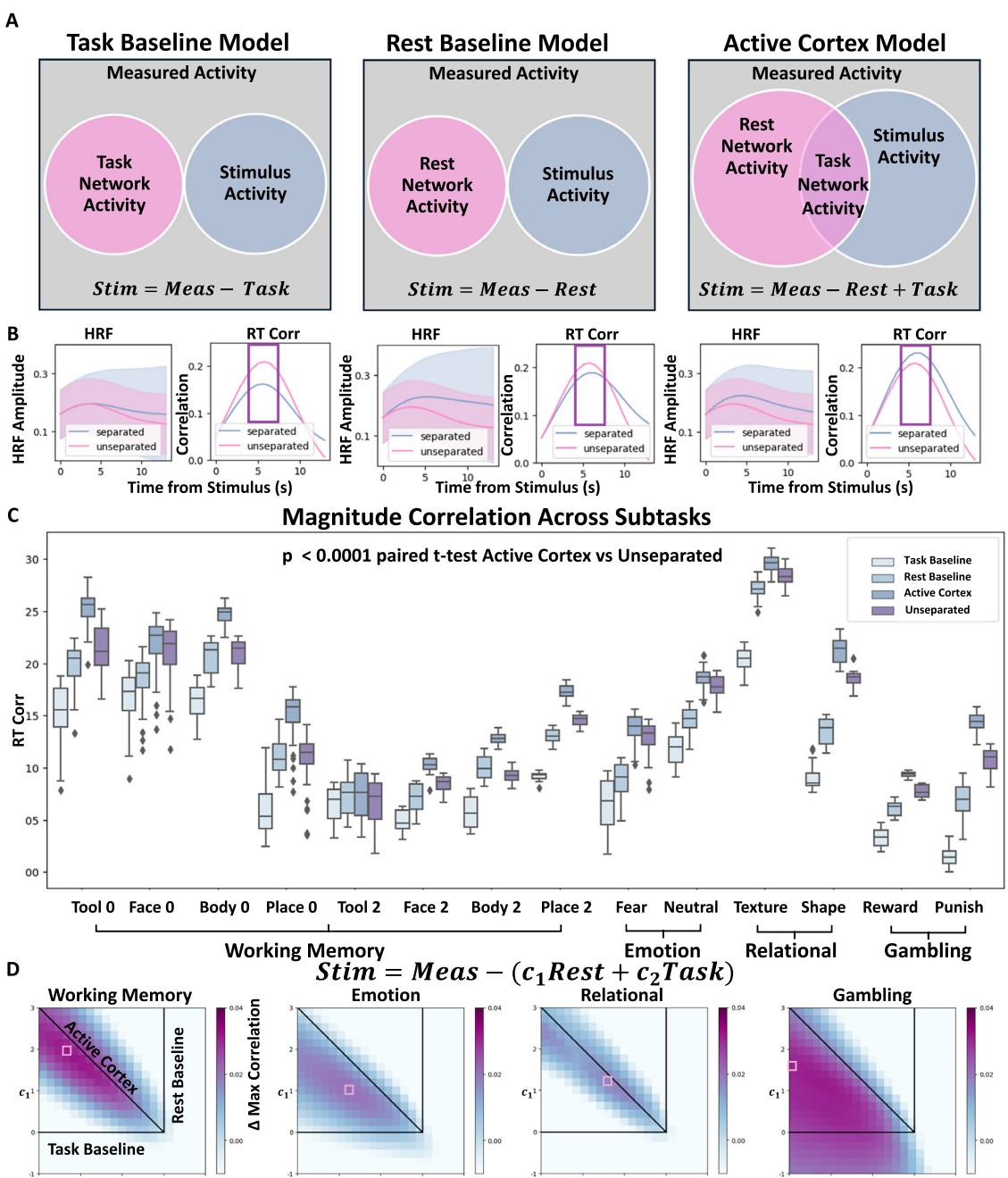

**Fig. 2 | Active cortex model. A** Possible relationship framework between the Stimulus, the measured signal and the fitted Rest and Task network models. Left) The Task Baseline Model, which assumes that the Stimulus activity is the measured signal minus the Task network model. Middle) The Rest Baseline Model, which assumes that the Stimulus activity is the measured signal minus the Rest network model. Right) The Active Cortex Model that assumes that a certain subset of Rest, namely the Task network model, is involved in the processing of the Stimulus Activity. **B** The change in the average HRF of the max-correlated ROI for the WM and the corresponding change in correlation to RT over time for each of the three frameworks ($N = 1032$). **C** Changes in RT correlates are quantified in fourteen subtasks and are significant for all of the sub-tasks ($N > 1000$ for Task correlation and $p$-value represents two-sided paired $t$ test average 12 spatial temporal location correlation). **D** A generalized sweep of the linear combination of Rest network and Task network models plotted against the change in maximum correlation to the signal ($N > 1000$ for each Task correlation). The diagonal represents a scaled version of the ACM, while the vertical and horizontal lines correspond to the Rest and Task network models, respectively. The boxplot boundaries in (**C**) represent the interquartile range (IQR), spanning from the first quartile (Q1) to the third quartile (Q3), with a line indicating the median; whiskers extend to the most extreme data points within 1.5 times the IQR from the quartiles, and outliers beyond this range are plotted individually.

RT significantly after separating the signal into stimulus-dependent and independent components for all fourteen sub-tasks and showing an average increase of 0.09 $R^2$ in all sub-tasks. In Supplementary Fig. 9, we show that these results also hold while accounting for twin relationships since the HCP data, as it is known to contain large amounts of twin data. Thus, the ACM is able to explain nine percent of more variance by subtracting out the processes in the Rest ODE that are not contained in the specific Task ODE.

Fig. 3 | **Brain to behavior function. A** (Left) Multiple brain regions are associated with each of the task experiments that have a moderate correlation with the RT. The Max Corr refers to the maximum correlation across time for each individual ROI ($N > 1000$ subjects for each Task category). The areas with large correlation to RT are similar to previously published Task Activation Maps (See Supplemental Figure 6). **A** (Right) The normalized changes to the correlation based on ACM separation align only with a portion of the original ROIs associated with the Task. For WM, the relationship between task severity and regions showing changes in correlation is examined in Supplemental Figure 12, demonstrating that the areas with increased correlation are closely associated with task difficulty. **B** Results of utilizing an Elastic Net to aggregate the spatial-temporal signal to predict the RT for both the unseparated and the separated using the ACM framework for a sample of the sub-tasks. Each point in (**B**) represents a single trial and corresponds to the sub-task above it. **C** The distribution of using an Elastic Net on ACM separated and original unseparated data ($N = 825$, Train 207 Test repeated 10-fold). Separate or isolated task timeseries perform significantly better in predicting RT in all sub-tasks and result in an average increase of 0.09 $R^2$ ($p < 0.001$ using two-sided paired $t$ tests). The violin plot in (**C**) shows the median as a white dot, the interquartile range as a thick bar from the 25th to 75th percentile, and whiskers extending to the most extreme data within $1.5 \times$ IQR of the quartiles. The brain plots were made using ggseg[75].

## Relationship between Tasks

In the previous sections, one of the four meta Task models was used for the separation step, even though these meta Tasks each contained several sub-tasks. This leaves a few questions open regarding the specificity required for a Task ODE, how the different Task models compare to each other, and whether they are interchangeable in any manner. Unfortunately, there was not enough data to train a model for each of the fourteen sub-tasks, so instead four meta Task models were trained to represent the Task activity ODE (see Methods in Differences between Task Models).

We consider the following multitask relationship shown in Fig. 4A, where two overlapping tasks are shown with a partial overlap in network processes, but also contain their own unique response to stimuli. By removing the shared network component activity from each of these specific Tasks, the unique Tasks should separate apart. This is demonstrated in Fig. 4B, where the FC for each HCP participant's task

# Intertask Relationship

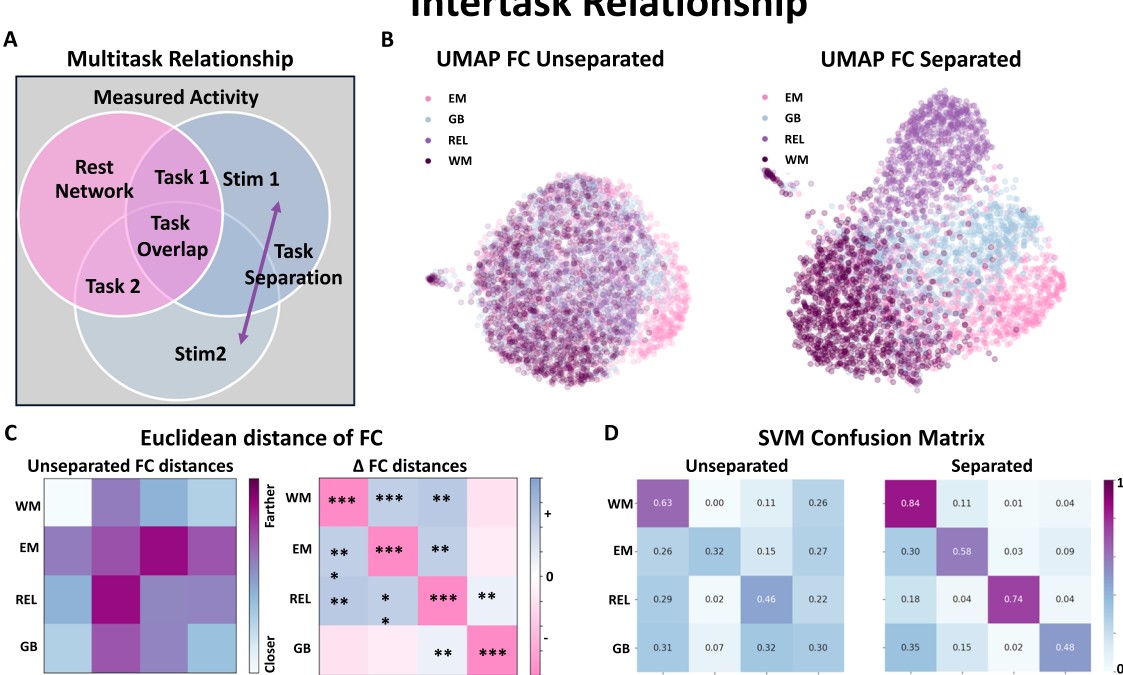

**Fig. 4 | Intertask relationship. A** A possible model of two distinct Tasks illustrates overlapping and non-overlapping neural processes as well as their relation to Rest Network processes. **B** A UMAP representation of the FC matrices in the original unseparated space and the ACM separated space ($N > 1000$ for each Task category). Each point represents a single individual task run. **C** Left: The mean Euclidean distance of each individual Task FC ($N > 1000$ for each Task category) to each cluster center. **C** Right: The change in Euclidean distances in the separated case. **D** The classification rate using an SVM classifier in the 2D UMAP space for the unseparated and separated cases (0.42 for the unseparated and 0.66 for the separated case). The stars represent different significance values of a two-sided student $t$ test: *** < 0.001, ** < 0.01, * < 0.05. WM- Working Memory, EM- Emotional, REL- Relational, GB- Gambling.

activity is calculated using the unseparated and ACM separated signals (see Methods in Analyzing Functional Connectivity Matrices). The differences are visualized in high-dimensional space, using a UMAP 2D projection (see Methods in Analyzing Functional Connectivity Matrices). The ACM separated case yields much more localized Task localization in high-dimensional space, with each of the individual Tasks separating to their own cluster. The mean Euclidean distances are shown in Fig. 4 between the mean cluster centers of each Task FC to each instance of a specific Task FC. After separation, the distances within a single Task category decrease, while the distances between Task categories increase. Using an SVM classifier on the 2D UMAP data, we demonstrate that it is easier to distinguish the FCs of particular tasks after applying ACM separation than before. The confusion matrices shown in Fig. 4D show an accuracy of 0.42 for the unseparated case and 0.66 for the ACM separated case ($p$-value < 0.00001). The differences in UMAP for the other two frameworks, Task Baseline and Rest Baseline, are shown in the Supplementary Fig. 10, showing a decrease in separability compared to the unseparated measured signal.

We also tested if this increase in Task differentiability translates on a per trial instance in Supplementary Fig. 10. Using a Random Forest Classifier, we tested how well we can classify the separated vs unseparated trails with respect to their meta Task categories. Once again, only the ACM framework showed a significant increase in classification accuracy of 15 percent ($p < 0.00001$ two-sided paired $t$ test) compared to the unseparated cases. The other frameworks showed either no change or got worse after separation.

To evaluate whether the separated FCs retained important individual characteristics, such as overall accuracy across the task or the percent of task completion, we tested different frameworks to assess their ability to predict these variables. In Supplementary Fig. 12, we demonstrate that for the WM task, ACM is the only tested strategy that

improves the prediction of both accuracy and task completion. The differences are smaller in magnitude compared to those observed in RT, as the separation effects are averaged across all trials. However, the general trends seen with other behavioral measures remain significant, with ACM consistently outperforming the other frameworks.

## Discussion

In the previous sections, we demonstrated that by utilizing network models to capture fMRI dynamics and by assuming the ACM, we can interpret and separate the fMRI signal. The first discussion section discusses the ACM and its implications. The second discussion section delves into the network model and its implications. Following those sections, we contextualize our findings in the current research framework of whole-brain modeling and speculate on how this approach could develop in the future. We subsequently examine limitations of the approach and explore methods that might overcome these barriers. We then discuss how this could lead to better models in the future.

The results of the ACM modify our understanding of the extensively studied Rest and Task relationship. Early DCM studies characterizing and modeling Rest and Task states represented Task networks by adding a component to the existing Resting State networks while simultaneously solving for external stimulus input[29,30]. BNM models used a fixed structural network model but simulated input stimulus using Gaussian processes[17,31]. Other studies have refined the concept of a fixed network by introducing weighted sub-networks derived from activity flow maps, using functional connectivity (FC) to predict the activation of sub-networks from Resting State functional networks[9,24,32]. Separately, structural sub-networks of the global structural network derived from tractography have also been shown to be predictive for Task activation[10]. These works, along with other works, have also proposed the property of the stimulus quenching,

that is, during Task activity the system approaches its sigmoidal maximum and the dynamics is quenched[7,9].

In our experimental paradigm, the ACM aligns with the idea that specific Resting State sub-networks are elevated to facilitate task performance. The common network processes, primarily captured by the Rest ODE, can only be removed from the signal if the distinct network processes, represented by the Task ODE, remain intact. We demonstrate this across five behavioral variable tests and fourteen sub-tasks spanning four meta-tasks. In addition, we show that functionally similar tasks yield ODEs with more structurally similar coefficients (see Methods section in Differences between Task Models), allowing for greater interchangeability and improved isolation of task-specific processes via ACM separation. This is further supported by our use of metatask models to decode individual sub-tasks, where we only had sufficient data to train meta models, but were able to apply to each of these sub-tasks individually. These findings suggest the existence of a fixed set of sub-networks, with each task selectively engaging specific processes for task execution. Furthermore, we identify sigmoidal relationships between network regions, consistent with previous literature, which could lead to stimulus quenching when these sub-networks are elevated in activity to perform a given task.

Our results also align with prior models of Task using Generalized Linear Models (GLM) that have assumed that Task activation is replicable when observing across multiple stimuli, whereas Rest processes are more randomly distributed[33,34]. This assumption has allowed these models to statistically identify the regions where the Task occurs. This aligns with our approach of training separate models for Rest and Task, where during Task, Rest processes that are not associated with the Task appear to be unstructured processes. GLM models also utilize two Task conditions with varying difficulty levels to isolate primary cortical regions responsible for the Task, rather than all regions necessary for Task execution, such as the visual cortex or motor areas involved in button pressing[34]. In our variant of this experiment, by contrasting regions with higher correlation during 0-back vs. 2-back, we identify regions that match Task severity. These regions closely align with areas that show the greatest normalized increase in behavioral performance due to ACM separation (see Supplementary Fig. 11). This suggests that ACM selectively enhances areas responsible for Task severity and difficulty rather than all regions involved in Task execution.

Another intriguing approach, distinct from the classical GLM paradigm for analyzing task activity, involves designing studies with long intervals between task stimuli to better account for the system's state at task onset[35,36]. These long intervals are used to construct FC preceding stimulus onset and compare its relationship with the accuracy of subsequent trials. In our case, the immediate activity preceding the Task interval is used as an initial condition to estimate a trajectory based on our models. We agree with the conclusions of these previous papers that this initial state is important, as by removing trajectories without accounting for the shared processes between Rest and Task only results in a decrease in association with behavior, as can be seen by our Rest and Task Baseline models. We only observe increases in behavioral predictions by subtracting selective trajectories associated with Rest that are absent in the current Task. Therefore, the initial conditions and our approach, which generalizes to trajectories, suggest that the moment of Task onset plays an important role in predicting Task performance.

The Resting State network ODE identified by the SINDy algorithm exhibits both expected properties and surprising differences from previous studies. Earlier studies represented the fMRI equation as a sigmoidal function with the argument being a product of $SC$ and the activity of each brain region[14,15,17,31,37,38]. The SINDy solution affirms that intrahemispheric connectivity indeed has the prominent features of the $SC$ network. However, interhemispheric connections are different from those present in the $SC$. The predicted interhemispheric

connections are dominated by strong contralateral connections between homologous brain regions, and negative connections between non-contralateral regions. This suggests a segregation vs. integration architecture where regions within a hemisphere are integrated, while the two hemispheres are segregated and pushing each other apart, and at the same time, homologous regions between hemispheres are being coordinated together.

The intrahemispheric connections contribute to the majority of connections in the measured $SC$. Their presence in $EC$ supports the main BNM theory that the derivative is directly proportional to the activity of its connected neighbors, except in the extremes, where, due to biophysical constraints, it is limited. Thus, these processes can be represented by a sigmoidal function with an argument being the product of $SC$ and the activity of the brain regions[17,31]. The formulation is also quite similar to connectome harmonics, where the graph Laplacian is a linear version of the identified $EC$[39]. Interestingly, the first-order coefficients were enough to explain most of the dynamics variance as shown in Fig. 5D, which is in line with the conclusions of previous publications and indicates that the dynamics of the resting state is mostly linear[40–42].

The temporal predictions of the Resting State network model in our work offer a distinct metric in brain dynamics, as only a few previous studies have used ODEs to predict the timeseries of the fMRI spatiotemporal signal. While its hard to evaluate predictions without sufficient precedence, the time series predictions seem remarkably accurate across the entire cortex. The brain regions where the algorithm performs poorly are areas that are known to have poor signals on fMRI, namely the medial orbital frontal, frontal pole, and temporal pole. The model also had poor predictions for the smaller subcortical areas in the basal ganglia, whereas the predictions for the thalamus and cerebellum were fairly accurate. Moreover, from the Supplementary Fig. 2, the inclusion of subcortical regions in the model was the key to increasing the predictability of cortical areas even if some of them had poor signal quality. Individual variance in these predictions indicates the possibility of using multiple models to represent the population, but remarkably, the results seem to generalize pretty well throughout the population, as can be seen with no difference in accuracy between the prediction test and training sets (see Supplementary Fig. 3).

There are a few studies that have published metrics similar to these time series predictions using ODEs, as it is a relatively emerging field. One is a combined EEG/fMRI paper, where the EEG signal is used as noise to simulate the fMRI signal using a BNM. However, this has yielded poor correlations at an average of around 0.2, which is much smaller than the model shown here, which only uses fMRI with mean correlations above 0.38[43]. A more recent paper uses a linear model to predict the fMRI signal from one ROI as a linear combination of other ROIs using white matter $SC$, that is, $X_a = SC*X_{b \neq a}$. This is not a differential equation, but the current activity of one region is related to a linear combination of the current activity in other ROIs. This has produced very accurate predictions with a large correlation mean between 0.6 and 0.7[44]. If we assume that our differential equation is in practice $sigmoid(SC*X)$, then the integral is equivalent to $log(e^{SC*X} + 1)$ and if we assume that the exponent is much larger than 1, then the equation reduces to the linear relationship, which would explain both our results as well as the results of the other paper[44].

The simulation of FC through noise integration is easier to compare with previous work, as it has been extensively simulated and studied[17]. Our results do not have the highest correlation compared to other methods, as some recent studies have correlations using higher parcellations above 0.9, compared to our average correlation of about 0.7[45]. However, as illustrated in Supplementary Fig. 3, the FC metric is the least specific in the analysis of dynamical systems, and we believe that the predictions of the timeseries are much more informative of the performance of the model. Moreover, timeseries predictions allow us to separate the fMRI signal, which was crucial to test the relationship

# Hyperparameter Optimization

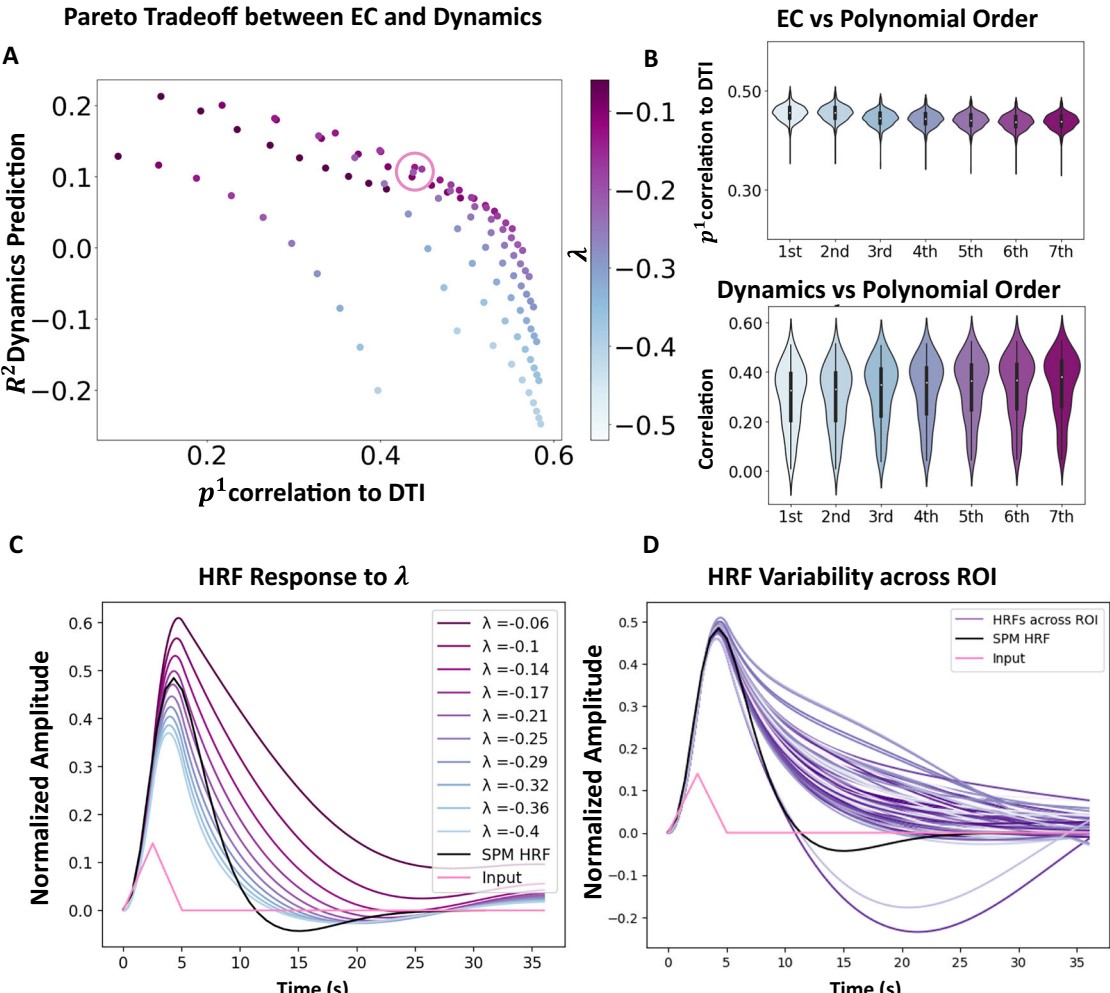

**Fig. 5 | Hyperparameter optimization. A** Pareto curve illustrating the tradeoff between having a model that captures the dynamics (y-axis) vs. the solution containing more biologically plausible network similarity to measured SC (x-axis). The models are varied using different sparsity thresholds, as well as varying the diagonal constraint $\lambda$ that controls the decay rate of the system. **B** The effect of using different orders in the Taylor expansion on SC ($N = 443$) and the dynamics metrics ($N = 90$). The correlation with SC decreases slightly with the inclusion of higher-order terms, but the reproduction of the dynamics improves with the inclusion of higher-order terms. **C** HRF function simulated from the Thalamus at different levels of $\lambda$, where more negative values result in a faster decay rate (Mean of

100 simulations). The canonical SPM HRF is plotted in black, and the input injected into the derivative space is plotted in pink. **D** The HRF function for $\lambda = -0.22$ is plotted with different ROIs, showing that there is a considerable amount of HRF variability depending on the location within the network (Mean of 100 simulations). Cerebellar traces are those with the largest undershoot. These calculations are performed for the Resting-State ODE model, but the Task Models give similar results. The violin plot in (**B**) shows the median as a white dot, the interquartile range as a thick bar from the 25th to 75th percentile, and whiskers extending to the most extreme data within 1.5 × IQR of the quartiles. HRF Hemodynamic Response Function. EC Effective Connectivity. SC Structural Connectivity.

between Rest and Task, and are not reproducible utilizing the past brain simulation setup as the beforementioned paper.

Our predictions of the HRF are also relatively unexplored in previous work. Although a lot of work has been done using the Balloon Windkessel model (BW), less work has been done on HRF predictions due to the input of the BNM or the NFE community[46]. This is because in most of these models, a BW model is used to transform neuronal activity to a hemodynamic response, and hence, by definition, the HRF response of these models is designed to match the measured HRF. Here, the HRF naturally arises while setting a proper value for $\lambda$. The model also predicts the variability of the HRF based on its location in the network. In practice, previous BNMs usually set $\lambda$ to $-1$, but this should be revised in future models as it does not match the decay rate expected from the experimental results, namely $-0.22$[17,31]. It is also interesting to note that $\lambda$ needs to be set as a constraint in the

algorithm, rather than being estimated directly, unlike all other network connections. This might be explained because the single-nodal dynamics is a function of both the decay function and the network activity. Thus, due to the two conflicting sources, we speculate that the SINDy algorithm is unable to differentiate between them. By setting the decay value to a constant, we can estimate the EC as shown in the Supplementary Fig. 4. Although we can predict variation in the simulated HRF, it is hard to compare with experimental predictions, as it is difficult to design Task paradigms to elicit HRF responses unique to each ROI[47]. Furthermore, depending on the Task, the HRF function can vary in a given brain region. Nevertheless, such comparisons should be attempted in future work, as it is a good metric to narrow down the choice of models aimed at reproducing resting-state dynamics.

A DCM paper that applied modeling to enhance RT correlates in the HCP WM dataset is likely the most comparable experimental

paradigm to our methodology on the HCP dataset[48]. This study has also predicted on a single RT level, but they achieved a lower degree of correlation to reaction time using their DCM model (r-squared of 0.21 vs r-squared of 0.5)[48]. This substantial difference might be in part explained due to the inclusion of many more brain regions in our model (only 11 ROIs for the DCM vs our 84 ROIs being modeled), which improved the Elastic Net's estimation of behavioral correlates even for the unseparated case. Compared to other estimates of behavioral correlates using HCP WM data, our results are quite promising, as getting r-squared beyond 0.4 seems challenging[49]. Another recent paper uses network modeling to make inferences about WM using HCP data, but on a different scale by averaging RT metrics across WM trials rather than predicting them on a per trial basis, which makes it hard to compare against[45]. However, we have a better fit to behavioral correlates, suggesting that fitting on a per trial basis might be more appropriate in modeling WM. While we find our results promising in separating Rest from Task, further work needs to be done in fully understanding Task activation, and we hope this work will act as a catalyst and facilitate more research in this direction.

Our approach to model the fMRI equation is similar to the early approaches used to describe ODEs in fMRI, such as DCM[29,46]. In fact, our equations are very similar to those first proposed by Friston et al., except rather than using a linear model, the SINDy framework solves for a general Taylor Expansion to represent the relationship between nodes[29]. DCM impressively solves for both inputs and the ODE structure, while we relax the conditions relying on the law of large numbers and regression to correctly deduce the ODE framework and interpret the non-structured activity to represent stimulus activity[30]. Another difference between the two approaches is that DCM assumes the deconvolution problem of inverting the signal to solve for neural relations, which does not allow for direct predictions on fRMI timeseries data. However, one of the main bottlenecks of DCM is that the method does not scale well and thus has been hampered in characterizing the dynamics of large sets of ROIs. A recent publication that analyzed HCP WM single trials only modeled 11 ROIS[48]. SINDy allows us to solve for the same system but modeling larger amounts of ROIs because it uses regression rather than Bayesian Inference, which can scale appropriately. It can therefore also solve for the complex HRF function and thus make predictions as well as manipulate fMRI on the measured timeseries resolution. Regression based DCM overcomes many of the challenges of traditional DCM and has been able to achieve whole-brain ROI connectivity analysis using a linear model[42,50]. Our methodology, which also applies regression, is a generalization of this method and uses more terms in the Taylor series expansion to account for nonlinear relations. The SINDy spectral package solves for the coefficients in the frequency spectrum similar to those proposed in Frassel et al., 2017. However, since they utilize deconvolution to estimate the neural equation rather than solve for the BOLD equation, their approaches are limited to observing differences in effective connectivity during Task[50] rather than applying the model to directly modify the signal as shown by our timeseries separation.

Meanwhile, the generative modeling community adapted the DCM equations by integrating estimates of SC from white matter tractography, partially to avoid directly solving intricate structural relationships, resulting in a family of models that utilize fixed differential equations, specifically BNMs[15,17]. These network models assumed a network SC derived from white matter connectivity and were popular in generating signals that have FC similar to the resting state FC by integrating candidate network equations with noise over a long period of time. Many of these approaches also use a Balloon Windkessel similar to DCM to transform the simulated neural data to fMRI data[37,38]. However, BNMs have the limitation of simulating (1) their model with noise (2) in another space that could be adjacent to measured data. This makes timeseries predictions rather hard[13], and in fact has been used to simulate summary metrics such as FC, which makes modeling

per-trial Task analysis as shown in this study difficult. They have, however, uncovered properties of Task trials[45] while other research have derived simpler activity flow models from the BNM underlying equations to model Rest Task switching as activations of sub-networks of Resting State functional networks[9,32]. Another popular approach uses NFE that utilizes a wave equation to describe the dynamics, and uses the physical cortical sheet in order to solve for the Dynamics. NFE solves for eigenmodes based on the cortical structure and simulates Resting State dynamics as a superposition of these modes[18]. These can be done in higher resolution than network models in voxel resolution[19]. We have provided the eigenvalue decomposition of our Jacobian, namely the first-order terms of the EC, shown in the Supplementary Fig. 5. They exhibit similar structures observed in NFE, with global eigenmodes with large eigenvalues, and smaller non-bihemispheric modes with smaller values. Apart from the network structure, BNM and NFE are roughly equivalent in terms of their scope. Both are able to show that the simulated signal has dynamic properties, i.e FC, multiple states, which are similar to those extracted from Resting State fMRI, but as they usually simulate in adjacent space which is not representative of measurements, they fail to provide the means to predict/ manipulate the fMRI timeseries directly as shown in this study.

Deep learning approaches have also been used to train dynamical systems to represent the underlying ODE[21,22,51,52]. Here, we specifically refer to a subset of deep learning approaches designed for ODE identification in fMRI, rather than all possible architectures that have been used or could be used to represent fMRI data. A specific architecture, piecewise linear recurrent neural networks, has been theoretically shown to solve any general ODE system from observations and is similar in scope to our regression based approach[21,22]. After applying the algorithm on fMRI datasets, they show they are able to predict the evolution of measured fMRI data[22]. However, most gradient learning-based approaches have issues starting from different initial conditions and converging to a single solution. Deep learning techniques have no convergent limits, unlike the simpler traditional regression-based techniques that this study employs. Moreover, the dynamics are solved in a latent space, which results in different coefficients on every instantiation on the same dataset[13,21]. This makes testing the prediction of ACM challenging as it requires two stable models trained on Rest and Task differently. Thus, this method has been used primarily to model general system statistics, such as the stability of the system in diseases and do not directly manipulate the timeseries, which is required to separate the signal[22]. We anticipate, however, that future deep learning models will advance system identification algorithms, as the field continues to develop rapidly, and those future models are likely to outperform regression-based approaches.

In relation to denoising algorithms such as ICA, PCA, filtering our model increases the signal variance, unlike those transformations (see Supplementary Fig. 7)[53,54]. These increases in signal variance are most similar in our experiment in Supplementary Fig. 7 to adding noise into the signal, but this increase in variance is related to improvement in behavioral correlates in regions corresponding to the Task. In addition, the previously mentioned techniques, as well as other approaches such as FC, do not represent a causal dynamical pathway, unlike ODEs and are in general more difficult to interpret than a causal model. However, these earlier methods are computationally inexpensive compared to our approach. Although the algorithm can execute within minutes, in its current form, it does require a lot more data to train a valid model and is a limitation of the current implementation (see the next section for more information).

There exist limitations in this study and approach. A large amount of fMRI data is needed to train the models (see Supplementary Fig. 4). In HCP terms, about a hundred subjects are needed with all four resting state scans to train a single model. This has limited the scope of the analysis to the HCP dataset or similar large datasets, because HCP

contains enough long recordings that can be used to train Task and Rest network models[55]. Limiting the analysis to HCP datasets tempers our results to healthy young adults who were scanned with the HCP protocol. Moreover, the analysis assumes that the development of the cortex is almost stationary and does not take into account any effects of aging, as well as known gender differences that affect cortical organizations and dynamics[56,57]. Moreover, the data represent awake data and do not take into account any changes due to circadian cycles, as well as longer cycles that have been shown to affect the dynamics of the system[58]. The analysis could be reproduced in the future in similar large imaging studies, such as the UK Biobank, that have task fMRI data[55]. Using priors for the dynamical system can offset the data needed for large amounts of data in future approaches and allow for shorter segments to modify the group dynamical system.

We have only tested the linear separability between the identified Rest and Task network models in this study. For future studies, non-linear separation schemes should also be considered, and might provide a substantial boost in accuracy. However, this is difficult in practice; for example, consider a simpler special case of an optimization problem where equation (7) is rewritten as follows:

$$RT = G\left(\int \dot{X} - Q(\Xi_{Rest} * \theta(X)) + P(\Xi_{Task} * \theta(X))\right) \qquad (1)$$

Solving this optimization is difficult, as the comparison to the RT coefficients requires another function $G$, which we implement as the Brain to Behavior Function. This function is approximated in two manners in this study (1) using correlation, and (2) using an Elastic Net. Since the relationship between Rest and Task are unknown and the Brain to Behavior function is also unknown, it makes it difficult to solve for both of them simultaneously. We have focused our analysis instead to testing a linear relationship between the Rest and Task networks by simply using coefficients for $P$ and $Q$ and sweeping them across a range. In future studies, a more complex technique may be able to address this optimization problem directly and test the nonlinear relationship between Rest and Task.

The linear combination of Rest and Task ODE is not meant to capture the full complexity of neural dynamics, but to serve as a unified framework of dynamics, structure and behavior to examine structural overlap between conditions. The models we construct for Rest and Task are derived by separating their respective epochs into structured and unstructured processes using an ODE-based framework. This structured component represents the dominant network dynamics captured during each condition. Since both models are established through regression, it is methodologically consistent — and scientifically meaningful — to test whether a linear combination of these fitted models can explain measured data.

In this context, the use of linear combinations provides a conservative and interpretable mechanism for evaluating the overlap between Task and Rest dynamics. If Task-related processes are indeed embedded within the structure of Rest, then the linear projection of one model onto another becomes a natural way to quantify and test that inclusion. Our results — including improved behavioral relevance after separating stimulus-dependent components — support this hypothesis.

This study shows that by utilizing ACM, we can improve the isolation of Task both from the measured timeseries and increased separation in the inferred network FC space. Although timeseries and/ or FCs are the main input in imaging-based cognitive and clinical studies, it is not clear how they will directly affect each individual study[59,60]. Future research has to be conducted on how applicable they are, but since they are able to change the primary data analyzed in these studies, we speculate that they will show an improvement on behavioral variables and Task interpretability in most studies. Moreover, we believe that this approach can be improved upon, for

example, by using more ROIs, by constructing priors to analyze smaller datasets, and by using gradient-based deep learning tools to determine more complex equations. However, in all of these contexts, we believe that the most significant impact of this study is the paradigm shift in applying modeling techniques to directly model components of the whole brain activity. This approach enables separating the timeseries from our modeling predictions, allowing more relevant signals to be extracted, for example, Task separation, as shown in this paper.

This work contributes to the field by introducing an ODE-based model with biophysical properties to interpret Rest and Task fMRI signals within a unified, causal framework. While prior approaches have separately modeled aspects of brain structure, dynamics, or behavior, our method uniquely integrates all three. By grounding the framework in regression, we identify the governing dynamical systems for Rest and Task separately, allowing for a methodologically consistent and scientifically meaningful test of whether a linear combination of these models can explain the measured data. We show that assuming Task is a linear subset of Rest reveals distinct signatures in the isolated Task signal, including heightened hemodynamic response in regions of Task engagement, and increase in variance that have a stronger association with a wide range of behavioral measures. This suggests the Active Cortex Model as a core principle of cortical dynamics, where the nervous system at Rest consists of all processes being active, and any Task activity is a subset of these processes. Having established this regression-based framework, future research can build on it by incorporating more complex components to further refine our understanding of brain function.

## Methods

Our methodology is organized into the following sections. The first subsection examines biological data as well as how they were preprocessed for this study (section Biological Data). The following subsection examines the background of how previous BNMs were used to model fMRI and how they can be used to inform about biological constraints for data-driven approaches (section Assumptions of the underlying fMRI ODE). Subsequently, we formulate how to identify an ODE to represent network activity from fMRI data by casting it as a regression problem with constraints and solving it using Sparse Identification of Nonlinear Dynamics (SINDy) (section Sparse Identification of Nonlinear Dynamics). The subsequent subsection examines how to validate the solutions and adjust the hyperparameters to find an optimal solution (sections Metrics to evaluate identified ODE and Optimization and Hyperparameters).

Following those are sections dedicated to the Task separation, starting with how the signal is separated using the network models (section Signal Separation). Subsequently, we show how we preprocess the behavioral data and align them with the fMRI signal (section Behavior Processing). The following sections are dedicated to how we relate the behavioral data with the brain activity (section Brain to Behavior analysis) and to how UMAP was applied to separate the functional connectivity in a hyperdimensional space (section Analyzing Functional Connectivity Matrices). The last section examines the differences in the identified Task ODE models and the identified Rest ODE model (section Differences between Task Models).

### Biological data
The datasets used in this study are from the Human Connectome Project[23]. HCP was chosen because it contains both Rest and Task datasets, as well as the corresponding DTI for each subject. The Desikan Killiany atlas was used as the parcellation scheme, which contains 68 cortical areas and 16 subcortical areas for a total of 84 regions to represent the central nervous system[61]. The Desikan Killiany atlas was utilized, as it is currently the most used atlas in the Brain Network Modeling community and allows our results to be compared to

previous results[13,14,17,31,37,38,41,43,53]. All available ROIs were utilized, and none were preselected. In theory, the methodology can be applied to other atlases with finer parcellations and would probably lead to overall improvements. We justify the inclusion of subcortical areas, even those brain regions that did not have great signal, as they seemed to improve the overall accuracy for the dynamic metrics in the cortex, as shown in the Supplementary Fig. 2.

**Functional imaging data processing.** The HCP fMRI data were downloaded from the minimal preprocessing pipeline, which was registered to the Montreal Neurological Institute (MNI) space[62]. For the Rest dataset, we utilized the smaller Q1 release of 447 participants using all four recordings. Since the Task data are much shorter and had only two recordings per individual, we downloaded as much Task data as was available, and downloaded 1032 subjects for WM, 1013 for REL, 1017 for EM and 1032 for GB. The cohort selected for WM contained $N = 472$ men and $N = 560$ women. From each of these datasets mentioned, we generate a single equation to represent each of the group Task models and the group Rest Model.

In order to make sure test and training datasets were split appropriately while also accounting for the constraints of having enough data to train the network models, we repeated the analysis using two different approaches: (1) a stringent task/ test split by individual while also accounting such that twins are not split between the test/training group for decoding the WM Task (2) a more relaxed approach for the other Tasks where all available data was utilized since the recordings were shorter. We show that stringent and relaxed approaches have similar results for WM (Supplementary Fig. 9), which is expected, as the backbone of the system identification algorithm is regression, which is known to be stable, to be generalizable, and has hard limits of convergence provided that there exists sufficient amounts of data.

The atlas boundaries were calculated in each individual T1 space instead of projecting the data into the MNI space of the group. Thus, we transformed the HCP fMRI data from group space to T1 space using the process described by the HCP Frequently Answered Questions section. In T1 space, we used the high-resolution T1 DK atlas provided for each individual by HCP to compute the Region of Interest (ROI) boundaries. This approach of parcellating in the individual T1 space instead of the group space has been used in our previous publication and is useful for modeling fMRI for Brain Network Modeling for averaged ROI timeseries[43,63].

The post-processing was kept to a minimum, as different preprocessing steps affected the identified ODE and would change the nature of the Taylor series polynomials identified. The only preprocessing steps that were applied was bandpass filtering from 0.008 to 0. Hz and normalizing (z-score) to the averaged ROI timeseries. The filtering step was used to minimize non-neural activity such as breathing and heart rate, which is shown to be around 0.15-0.3 Hz, instead of relying on other techniques such as ICA denoising. This is due to the fact that current ICA denoising has been shown to reduce Task behavioral correlates and was specifically cited by HCP as a reason for not including it in their release for Task fMRI[64]. We also utilized the SINDy algorithm built in the denoising tools as discussed in section "Handling Artifacts, Noise, and Physiological Processes".

**Behavioral data.** The behavioral data attached to the Task data were extracted from the behavioral tables. For the WM Task, these include the RT, the difficulty of the Task 0 vs. 2 back, the type of Task (Face, Place, Body, Tools) and the e-prime timings relative to the fMRI scan for each of the 80 trials in each WM scan. For the GB Task, these included the RT, Punish or Reward trial, and the e-prime timings relative to the fMRI scan for each of the 32 trials in each GB scan. For the REL Task, these included RT, Shape or Texture trial, and the e-prime timings relative to the fMRI scans for each of the 27 trials in

each REL scan. For the EM Task, these included the RT, fear or neutral trial, and the e-prime timings relative to the fMRI scan for each of the 30 trials in each EM scan. Since the EM Task script had a known bug during HCP recording, where only half of the last trial was recorded, we chose to truncate the last task block and thus only had 30 trials per scan.

**Structural diffusion data processing.** We followed a modified version of the Hagmann procedure to extract structural connectivity matrices from DTI[65]. The diffusion data was downloaded from HCP for 443 subjects. We applied a modified form of the MRtrix HCP pipeline to estimate the tractography based on a previous publication that optimized the tractography for the HCP dataset[66,67]. The tractography was aligned in the individual T1 space to be consistent with the fMRI preprocessing. The maximum track length was set at 250 mm, and the SC matrix was estimated by the number of tracks between ROIs based on the DK atlas, the same T1 used for the fMRI parcellation. The final connections were not normalized by volume area or any other method. Rather, the raw number of tracks between ROIs was used to compare with the Taylor series coefficients.

### Constructing a network ODE using fMRI data

**Assumptions of the underlying fMRI ODE.** Resting state fMRI activity has been described by the following general framework using network brain dynamics, where the change in neural activity in single brain region $a$ represented by the vector $\dot{X}_a$ is composed of three processes: (1) function of the structural network SC and the current activity across all other brain regions $X_{b \neq a}$, (2) a decay to baseline $-\lambda X_a$, as well as (3) local activity which looks like noise compared to the structured network activity $\eta Z$.

$$\dot{X}_a = -\lambda X_a + F(SC, X_{b \neq a}) + \eta Z \tag{2}$$

This framework has been extensively studied in the BNM literature, using complex functions to model $F$ such as a linear or a sigmoidal function, as well as using multiple variables to express neural mass activity $X$, such as in the Wilson Cowan model or the Jansen Rit model that uses multiple excitatory and inhibitory populations to represent the dynamics of a single brain region[17,31,68,69]. The activity is then transformed into the fMRI signal through the Balloon Windkessel model[37,46]. We cast this framework slightly differently by approximating a function for each edge $E$, namely:

$$\dot{X}_a = -\lambda X_a + \sum E(X_{b \neq a}) + \eta Z \tag{3}$$

The activity $X$ here represents the fMRI signal directly rather than the underlying neural activity in the BNM. Thus, we directly fit a specific network model for fMRI activity where the variables correspond directly to the measurable activity. From the previous literature, we know that the function $F$ is usually modeled as a sigmoid to simulate biologically realistic responses, since it plateaus for extreme values of $X$ while producing a linear response that is tied to the SC for small values of $X$. Thus, we expect that the estimated function $E$ will also be a sigmoid response function[31,40]. The decay rate $-\lambda$ determines the rate at which the system decays, which for fMRI can be experimentally approximated by observing the decay during the tail end of HRF. This constraint will later be used in the convex optimization problem, which leads to more biologically plausible solutions and is not estimated directly from the data, unlike the edge functions. However, we show that the optimal value for the constraint aligns very closely with empirical measurements of the HRF response.

**Sparse identification of nonlinear dynamics.** The unknown edge functions are approximated using an algorithm known as Sparse Identification of Nonlinear Dynamics (SINDy)[20]. SINDy is an open

source Python package that uses regression to solve for coupled ODEs for a given set of timeseries observations[20]. The algorithm identifies a sparse ODE system by writing the derivative of the state variables $\dot{X}$ as a product of a library of functions $\Theta(X)$ multiplied by the coefficients $\Xi$ plus some unstructured Gaussian noise $\eta Z$:

$$\dot{X} = \Theta(X) * \Xi + \eta Z \qquad (4)$$

The data matrix $X$ contains the activity of n different brain regions over time:

$$X(t) = \begin{bmatrix} x_1(0), & x_1(1) & \dots x_1(T) \\ x_2(0), & x_2(1) & \dots x_2(T) \\ \dots & \dots & \dots \\ x_n(0), & x_n(1) & \dots x_n(T) \end{bmatrix} \qquad (5)$$

The derivative of this matrix $\dot{X}$, can be estimated using numerical differentiation techniques for which SINDy provides many different options. We utilized the spectral method but also experimented with the smoothed finite difference method, but they both result in relatively similar results (see Supplementary Fig. 3). The library matrix $\Theta(X)$ contains functions of $X$, such as the constant, polynomial, or trigonometric terms that are assumed as prior in the analysis depending on the problem. In this case, we wish to fit a Taylor Series Polynomial between each edge. Therefore, we used a polynomial basis with no higher-order interaction terms. This assumes that each pair of nodes interacts with each other independently of other nodes and can be separately approximated by a function. This leads to solutions that have a first-order network structure, where edges are between nodes, rather than meta-edges between two or more nodes to a different node and is the framework that is widely used in the BNM community[14,17,31]. The network activity can then be represented as a sum across all edges. No symmetry condition is imposed, so the edge function between nodes A and B can be different from the function between B and A.

Thus the library function $\Theta(X)$ can be expressed as the following, and is computed for every observation in time of $X(t)$.

$$\Theta(X) = \begin{bmatrix} X(t) & X(t)^2 & X(t)^3 & \dots X(t)^N \end{bmatrix} \qquad (6)$$

Based on the derivative and library terms, we can determine the coefficients $\Xi$ from Eq.(4). Thus, Eq.(4) reduces to a regression. The problem can be reduced to the form of $AX = B$, where $X$ represents the coefficients $\Xi$ that we are solving for, and $A$ is represented by the library polynomial function $\Theta(X)$ and $B$ the observed derivative. The estimated coefficients $\Xi$ represent a Taylor Polynomial between each edge and can be thought of as an approximation of network connectivity using a polynomial basis. We expect that the edges determined by the algorithm are related to the physical wiring between brain regions and, therefore, should be related to other measures of network connectivity, such as through tractography using DTI. Here, we refer to the collection of edge polynomials as Effective Connectivity (EC). Although this term has been used in a slightly different context in the past, here it helps to clarify our methodology. The EC is believed to be sparse in nature as there are few physical connections between the brain regions, as they are restricted by resource constraints, in this case, the amount of volume available inside the skull. This has been shown in simpler systems such as the C. Elegans, but we believe this to be a general property of all nervous systems[70].

The regression problem can be solved using convex optimization tools that optimize the trade-off between the penalties on the reconstruction and the number of coefficients used, which is controlled by the sparsity parameter[20]. Again in the implementation of SINDy, there are many different optimizer choices, but we utilized the $L2$ regression, as $L1$ yielded very sparse solutions that did not fit the dynamics well. The residuals represent activity that is not related to the network ODE,

and is assumed to be unstructured noise $\eta Z$ from Eq. (2) and is minimized by the SINDy algorithm. Note that technically, $L2$ regression is not sparse, and instead, we employ ridge regression to determine the coefficients. While this is a misnomer when using 'Sparse Identification of Nonlinear Dynamics', the SINDy package offers much more versatile options for system identification than traditionally sparse-only solutions.

**Setting the diagonal constraint.** The SINDy implementation also allows us to incorporate prior knowledge of the dynamical system by fixing certain coefficients via constraints. This improves the overall accuracy of the algorithm and constrains the solutions to have realistic network connectivity. The use of constraints is also recommended by the original authors of SINDy[71]. From Eq.(2), the decay term $-\lambda$ is fixed as a constraint to ensure that the resulting effective connectivity (EC) exhibits a network structure that is highly correlated with the measured structural connectivity (SC) obtained from the tractography, as the algorithm cannot determine the network structure without this constraint (see Supplementary Fig. 2). In previous BNMs, the neural dynamics model $\lambda$ is usually set to $-1$[14,31,37]. However, in the context of constructing a dynamical system for fMRI, the decay rate is related to how fast the HRF decays. From the Statistical Parametric Mapping (SPM) provided HRF, we can approximate the decay rate to be $-0.22$[72]. Thus, for our implementation, we varied the $\lambda$ parameter from $-0.06$ to $-0.4$ and at each value, the system is evaluated using the metrics below in order to determine the optimal value of $\lambda$ (see Methods in Optimization and Hyperparameters). The reason $\lambda$ is a constraint rather than a solvable variable is that there are two processes that occur simultaneously from the perspective of a single ROI, the decay to baseline and the input from other regions. Therefore, by fixing the decay term, it improves the algorithm's ability to solve for the network interactions and results in solutions that are biophysically plausible. The results without this constraint lead to more randomized connectivity matrices and are shown in the Supplementary Fig. 2.

**Metrics to evaluate identified ODE.** Since these network coefficients are fitted using $L2$-regression, there exists a trade-off in either matching the dynamics or having sparse network solutions that are closer to white matter connectivity networks as estimated by DTI. To ensure that our model maximizes biophysical plausibility while accurately predicting dynamics, we apply the following three criteria:

1. **Match the network timeseries dynamics exhibited in fMRI:** The model should capture the network dynamics observed through functional connectivity (FC) by ensuring that the predicted timeseries closely matches the measured fMRI signal. Furthermore, the residual signals should be uncorrelated with each other, indicating that the structured network processes have been effectively accounted for in the model.

2. **Match the identified structural network with other methods of inferring the structural network:** The network edges modeled using the Taylor series coefficients should align with the structural connectivity (SC) network derived from other methods, such as white matter tractography. This ensures that the model's dynamics are driven by realistic network processes, rather than by an arbitrary solution to the system's behavior.

3. **Match the Hemodynamic Response Function (HRF):** The impulse response of the model, its HRF, should match the canonical HRF observed in empirical measurements, ensuring that the hemodynamic response of the model aligns with biologically plausible expectations.

To evaluate the identified ODE, we used these three main criteria to determine the plausibility of the system. To implement these three criteria to evaluate a particular ODE, we used the following process.

Firstly, we compared the predicted timeseries with the measured timeseries. As the constructed equation predicts the derivative $\dot{X}$ for every $X$ using Equations 1 and 2, $\dot{X}_a = -\lambda X_a + \sum_n \Xi_n X_{b \neq a}^n$, we can at every time point directly compare the predictions against the measured derivative over time. We tested this using a holdout test dataset, where we predicted the derivative using coefficients determined using 392 subjects, and compared it against the measured derivative in 50 unrelated subjects using the Resting State dataset (see Fig. 1G). Secondly, the coefficients $\Xi$ for each polynomial order were compared to estimates of $SC$ to see if the resulting network of edges matches our estimates of the network using different data modalities. For this, we used the 443 DTI tractography matrices and correlated the tractography matrix with each polynomial order and estimated the mean across all 443 individuals (see Fig. 1C). The third metric that we used is to compare the simulated HRF using the identified ODE with the SPM HRF function. To simulate HRF, 100 random points from the fMRI timeseries close to zero were chosen, and the average HRF trajectory was calculated across all points. The input location was varied across the ROI, but only a single triangular input was used at the singular ROI for each individual simulation.

**Optimization and hyperparameters.** The principal trade-off in using SINDy to characterize fMRI is between reproducing the dynamics and identifying a network that resembles $SC$ obtained from other data modalities. Figure 5A illustrates this tradeoff using a Pareto curve. The points on the lower right have a higher sparsity with fewer network edges. These sparser solutions have a higher correlation with $SC$, while the inclusion of more edges results in a better fit to predict the timeseries shown in the top left corner of the figure. A second hyperparameter $\lambda$ controls the decay rate, while also affecting the performance of the models, and is plotted using different colors in Fig. 5A. They correspond to the HRF trajectories shown in Fig. 5C of a single ROI of the thalamus. The SPM HRF function, plotted in black, decays at an approximate exponential decay rate of $\lambda = -0.22$[72]. However, the corresponding HRF for that constraint decays at a rate slower than $\lambda$ due to the network connections providing input into the ROI. Depending on the location of the ROI and its connections, it leads to variability in the HRF, as demonstrated in Fig. 5C. The model presented in most of the study was chosen to have a decent fit for all three metrics and is circled in pink in 5A. The decay rate was set to $-0.21$, close to the SPM decay rate, and the corresponding sparsity was chosen such that the adjusted r-squared using the sklearn toolbox is larger than 0.1 and the first-order $EC$ correlation to $SC$ is greater than 0.4. To remain consistent, all SINDy models, regardless of Rest and Task, were trained using the same decay rate and sparsity threshold. In addition multiple polynomial orders were tested, as shown in Fig. 5B, up to the maximum order of seven before encountering memory limitations. Inclusion of higher polynomial orders improved the fit to the dynamics without significantly altering the relation between the coefficients and $SC$. However, it should be noted that most dynamics can be captured by the first-order linear approximation, which is consistent with previous work[40,44].

**Handling artifacts, noise, and physiological processes.** In order to handle noise, artifacts and non-neuronal physiological processes, we introduced a few different steps to minimize their effects on the models identified. Since most preprocessing steps affect the dynamics, changes in the preprocessing are transferred directly to the identified coefficients. Instead, to address these nuisance signals, we utilized the in-built methods that exist in the SINDy package that have been specifically developed to address these issues. The SINDy package includes a derivative option, 'Smoothed Finite Difference' to handle noisy data, as well as a trimming function that ignores large derivatives due to artifacts. However, we found that in practice these made little difference (see Supplementary Fig. 3) and with sufficiently large

amounts of fMRI data, the ODE discovered is rather stable and consistent regardless of which subset is used to train the model. Therefore, we went for the fastest option, which was the 'Spectral Derivative.'

Since the dataset used to construct the ODE was a large subset of the entire HCP cohort, to alter and affect coefficients, artifacts and/or noise would have to occur consistently at the same time with reference to current brain activity, which would be very rare. Therefore, our method is robust to noise, which is generally the case for regression-based methods that produce stable, consistent, and convergent solutions.

## Separating task

**Signal separation.** To test which framework best represents the Rest and Task data, we applied the following general procedure to test different linear associations between Rest and Task. We hypothesize that stimulus-independent activity can be constructed by relating the Rest and Task network equations here referred to by their $EC$ coefficients $\Xi_{Rest}$ and $\Xi_{Task}$ and subsequently removed from the unseparated measured signal given by $X_{unsep}$. Since we operate in the derivative space, we integrate and take the initial measurement at Task onset as the initial condition to determine the separated spatial-temporal signal $X_{sep}$. Therefore, we relate the RT behavioral variables to the separated signal, using $c_1$ and $c_2$ as hyperparameters for linear combinations of the Rest and Task SINDy models:

$$\int_{t_{stim}}^{t_{stim}+T} \dot{X}_{unsep}(t) - (c_1 * \Xi_{Rest} + c_2 * \Xi_{Task}) * \theta(X_{unsep}(t))dt = X_{sep}(t) \propto RT$$

(7)

By varying $c_1$ and $c_2$, we test many different relationships between the measured signal and the ODE models. The Rest Baseline Model is given by $c_1 = 1$ and $c_2 = 0$. The Task Baseline Model is given by $c_1 = 0$ and $c_2 = 1$. The Active Cortex Model is given $c_1 = 1$ and $c_2 = -1$. We also swept the parameters $c_1$ from $-1$ to 3 and $c_2$ from $-3$ to 1 as shown in Fig. 2D. Although this association does not cover all possible combinations and is limited in testing linear relationships, it allows for an initial estimate of the relationship between Rest and Task.

Although the coefficients $\Xi_{Rest}$ and $\Xi_{Task}$ are fitted from group-level models, the timeseries $X$ represents an individual trial and contains information about the performance during the trial. The goal is to isolate task-specific activity in individual trials by removing activity unrelated to the task, using group-level models to represent Rest and Task network processes.

**Behavior processing.** An overview of our Task-to-behavior processing is shown in Fig. 6. Each individual scan contains multiple trials that can be divided into categories of sub-tasks. For the WM dataset, we utilized eight sub-tasks, namely, Place 0back, Tools 0back, Faces 0back, Body 0back, Place 2back, Tools 2back, Faces 2back and Body 2back. For the EM dataset, we utilized two sub-tasks, Fear and Neutral. For the REL dataset, we used Shape and Texture. For the GB dataset, we used the Rewards and Punishing labels and ignored the Neutral labels (single-trial labels, not block labels). For each trial, the stimulus onset time was also extracted from the HCP behavioral tables and the RT. The RT is recorded in milliseconds, which is defined as the time it took to complete each trial. We also recorded the accuracy of the trials, as well as recorded which trials were not completed and calculated the percentage of trials completed and the overall accuracy on a per individual basis. For the RT analysis, we only used correct trials. The fMRI signal was aligned with the start of the stimulus onset, and the subsequent 14.4 s (20 consecutive recordings) were resampled and divided by each sub-task category as shown in Fig. 6B.

Overall, we test the separation frameworks on five different behavioral variables: (1) RT prediction on a per-trial basis (2) Task classification on a per-trial basis (3) Divergence of Task FCs on a per

# Task to Behavior Processing

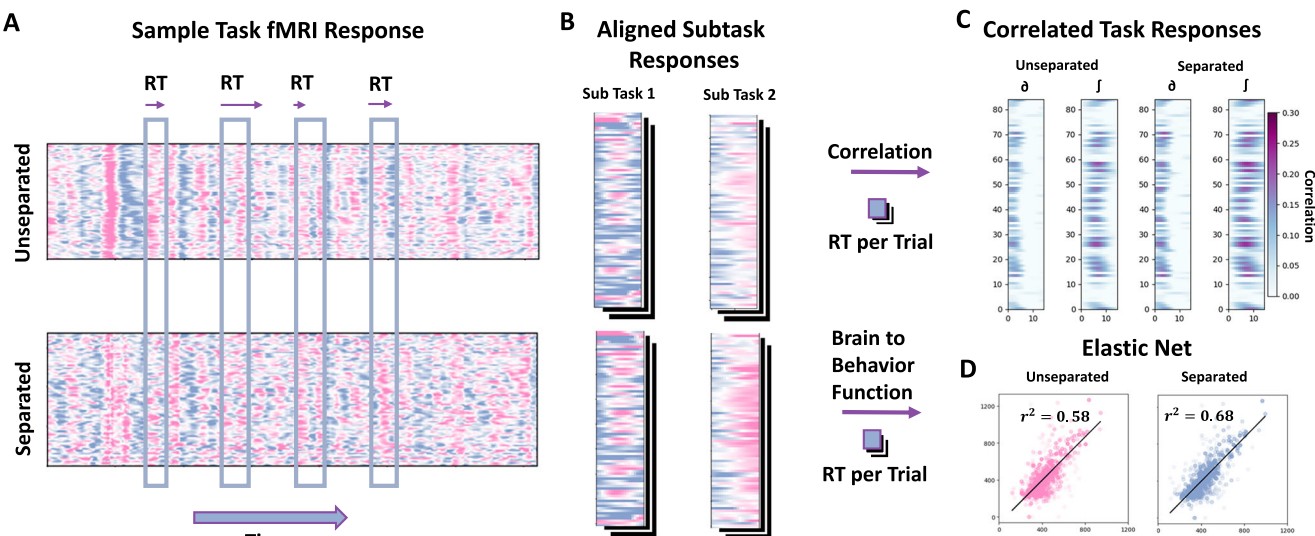

**Fig. 6 | Task to behavior processing. A** An example of a Task fMRI spatial-temporal signal showing both the unseparated and ACM separated signals. The windows represent 14.4 seconds after the task stimulus onset, and for each stimulus, the RT is recorded for how long the person took to respond and complete the task (displayed by differing arrow lengths). **B** The windows are resampled starting from the stimulus to align them on the same time frame and divided by each sub-task category. **C** Each of the spatial-temporal points is correlated with the RT for the Task, illustrating when and where the RT variables have the highest correlation to the measured brain activity. This is performed in the derivative space as well as for the fMRI time series space for both the unseparated and ACM separated signals. **D** To calculate the brain behavior function, the spatial-temporal signals (both the derivative and the original time series) are fed into an Elastic Net to predict the single RT of the resulting brain activity ($N = 825$, Train 207 Test repeated 10-fold).

individual basis 4) Accuracy prediction on a per individual basis (5) Missing trials prediction on a per individual basis.

**Brain to behavior analysis.** The spatial-temporal ROI signal was compared to the corresponding RT for each trial in the following two ways: (1) directly correlating the signal at each spatial-temporal location across all trials (Fig. 6C) and (2) constructing a linear model using an Elastic Net to aggregate the spatial-temporal information (Fig. 6D). For the ROI analysis, we calculated the maximum correlation across all ROIs and defined a subset of all spatial-temporal points that had 90 percent value of the maximum correlations. These spatial-temporal points were defined on the unseparated signal, and the changes to these correlations based on the separation frameworks are plotted in Fig. 2C. To calculate the delta maximum correlation as shown in Fig. 2D, we first calculated the maximum correlation during the task window (computing the correlation for each spatial temporal point across all ROIs for the first 14.4 s after stimulus and taking the maximum) in the unseparated case, as well as for each of the candidate separation schemes determined by the coefficients $c_1$ and $c_2$. To determine the delta, we take the difference between the separated scheme and the unseparated case.

The Elastic Net was trained using spatial-temporal patterns to match a single spatial-temporal trajectory to a single RT. Since the separation is actually performed in the derivative space before integrating (see section above for more details), we utilized both the derivative as well as the integrated signal as a feature vector. The Elastic Net is implemented using the sklearn toolbox and uses both an L1 and L2 penalty. The alpha and L1 ratio was varied and optimized to be alpha = 1, L1 ratio = 0.95. The same values were used for all instantiations of the Elastic Net across all Tasks. We trained the model using a 10-fold cross-validation scheme, namely by dividing the dataset into a training test ratio of 80/20. For the WM dataset, we also applied a more rigorous test, as shown in Supplemental Figure 9, where we repeated the analysis with only Non-Twin participants. The $R^2$ metric that was

reported was determined by applying the square root to the Pearson correlation between the predicted and measured RT and represents the explained variance, not the coefficient of determination.

To classify the single trial trajectories (Supplemental Figure 10, we applied our separation scheme, similar to the RT analysis to generate timeseries from stimulus onset for each trial. We then took only the valid trials and classified them by their Task category (GB, WM, REL, EM) using a Random Forest Classifier (200 Trees).

**Analyzing functional connectivity matrices.** To compute the different Functional Connectivity (FC) matrices, we gathered all trials (14.4 s after stimulus onset) for each corresponding Task (WM, EM, GB, and REL), regardless of their sub-task labeling on an individual level. Then an FC was calculated for each data modality, unseparated vs separated and fed into a UMAP implemented using the sklearn toolbox[73]. The parameters for the UMAP were the following: nearest neighbors set to 100, minimum distance 0.1, with 2 components. The metric to measure the distance was chosen as correlation. The latter was the most important in separating the FCs into distinct categories, as seen in Fig. 4. The SVM algorithm, also implemented using the sklearn toolbox, was used to gauge their separability in the UMAP space. This was implemented using the two UMAP components as inputs and using an RBF kernel to separate it into the four categories. A separate test set consisting of 20 percent of the data was then used to assess the accuracy of SVM.

We also tested if these FCs contained pertinent Task information that were enhanced after the separation such as the accuracy of the participant or the percentage of trials they completed. We used the Elastic Net (alpha = 5, L1 ratio = 0.01) described in the section above to map the FCs to both the accuracy percentage and the completion percentage for WM trials shown in Supplemental Figure 19[74].

**Differences between Task models.** We trained five network ODE models using the same hyperparameters to represent Rest, WM, REL,

# Cross Task Similarity

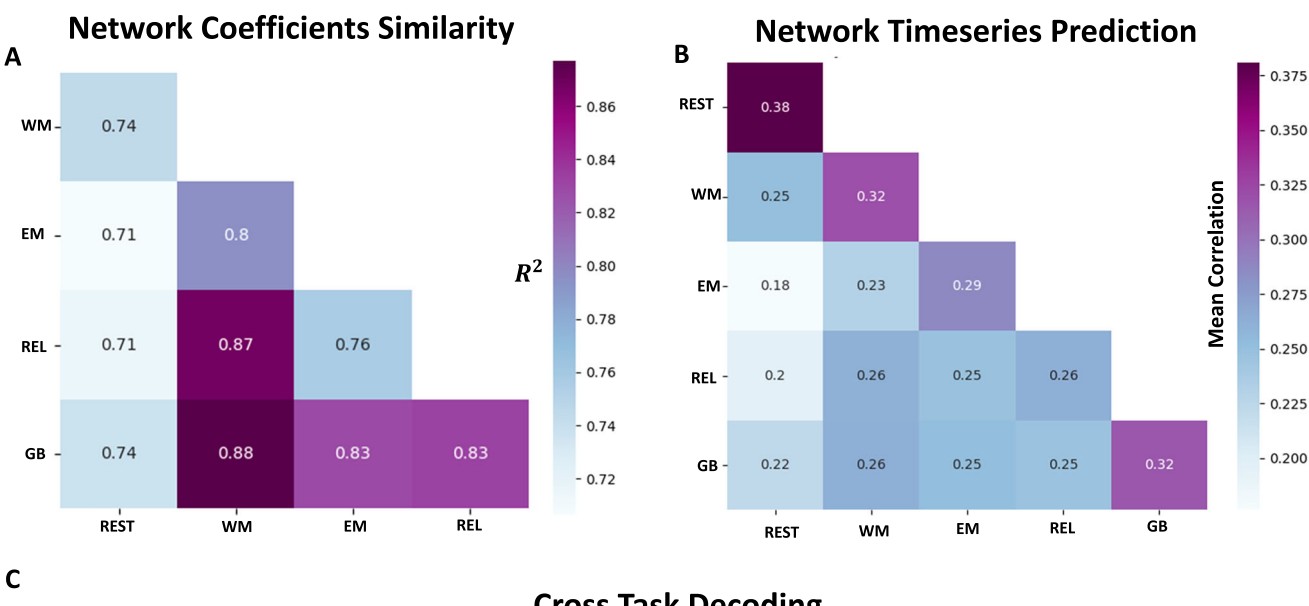

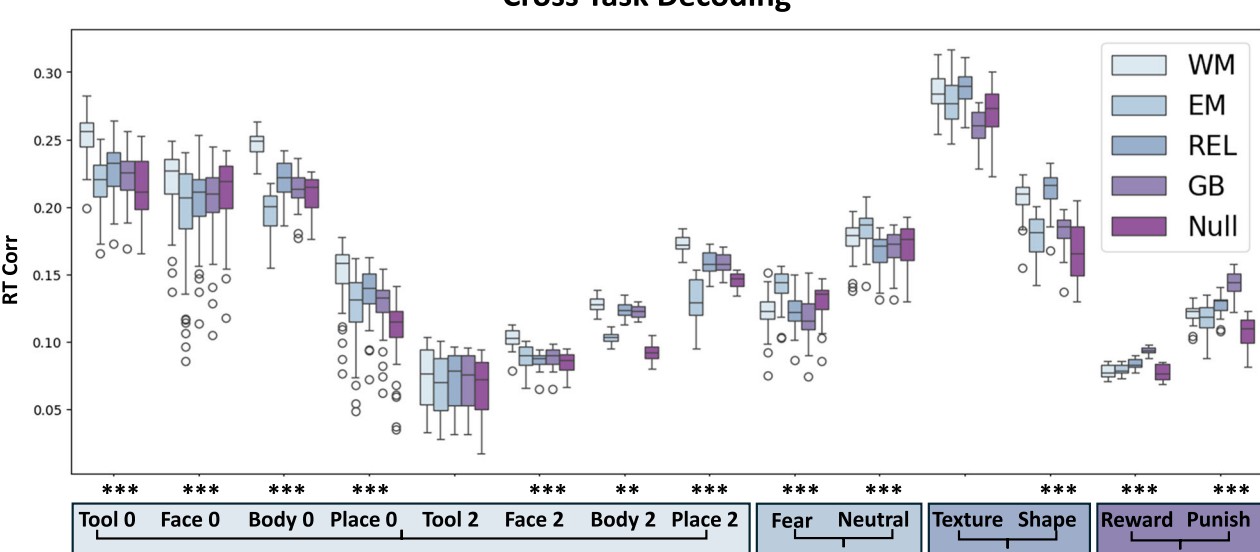

**Fig. 7 | Cross Task similarity. A** The similarity of the *EC* coefficients between each of the respective network models. **B** The quantification of the predicted network dynamics of the model is trained using one data Rest/Task modality and used to predict the other on withheld unseen data timeseries of another Rest/Task modality (*N* = 90). **C** The quantification of using one Task model to decode the signal from another Task and the changes to the resulting behavioral correlates. For all of the sub-tasks, the largest increase in correlation corresponds to decoding using the same Task vs. a different Task, although only for 12 of the 14 these differences are significant compared to other models (*N* > 1000 for Task correlation and *p*-value represents paired *t* test average 12 spatial-temporal location correlation). The stars represent different significance values: *** < 0.001, ** < 0.01, * < 0.05. The boxplots boundaries represent the interquartile range (IQR), spanning from the first quartile (Q1) to the third quartile (Q3), with a line indicating the median; whiskers extend to the most extreme data points within 1.5 times the IQR from the quartiles, and outliers beyond this range are plotted individually.

GB, and EM. The amount of fMRI data needed for the SINDy algorithm to converge is shown in Fig. 3. It roughly equates to around half a million timepoints. Since the Task fMRI dataset is considerably smaller, this represents about all of the REL, EM, and GB datasets and about two-thirds of the WM dataset. This also makes it impossible to train a model for each sub-task, rather just enough data to train one meta Task model containing all sub-tasks within. For WM, we also designed an experiment to test the robustness of the results since there was enough data available. Since HCP contains a high percentage of twin data, we repeated the analysis without containing the twin pairs.

Therefore, we trained a SINDy model using roughly two-thirds of the dataset containing no twins and tested on the other third that had twins. However, the results in Supplemental Figure 9 show that there is no significant difference when including and excluding twins, and demonstrate that our models generalize well among healthy HCP individuals.

A comparison of the different Task and Rest models is shown in Fig. 7. Each of the Task models has *EC* coefficients that are relatively of a similar correlation to *SC*. The correlation between *SC* and the first-order REL *EC* is 0.46, for EM *EC* is 0.41, for GB *EC* is 0.44, for WM *EC* is

0.46, and for Rest *EC* is 0.44. The *EC* are also fairly similar to each other, as shown in Fig. 7A where the $R^2$ is larger than 0.7. Interestingly, the Task models *EC* are more similar to each other than the *EC* of Rest. In addition, each of the models is specific to the dynamics they were trained on as the accuracy decreases approximately 0.1 − 0.2 when predicting across different Task/Rest types, as shown in Fig. 7B. This difference is not due to testing on an unobserved timeseries, as the accuracy is unchanged when using withheld examples of the same dataset shown in the Supplementary Fig. 8. The differences in the Task models affects their ability to separate the stimulus-dependent signal which can be observed when using an alternate Task model to decode. Figure 7C shows that using an alternate Task model results in an overall decrease in correlation to behavioral variables. However, these differences are only significant in 12 of the 14 sub-tasks, suggesting an overlap between certain Task processes.

### Reporting summary

Further information on research design is available in the Nature Portfolio Reporting Summary linked to this article.

### Data availability

The HCP data are available under restricted access for data protection, access can be obtained by creating an account and requesting access. The raw HCP data are protected and are not available due to data privacy laws. The processed HCP data are available in our Dryad repository: https://doi.org/10.5061/dryad.bzkh189p9. The source data for each of the figures, as well as many of the look up tables and ROI ordering, are provided in the GitHub folder.

### Code availability

The code that reproduces the resulting SINDy analysis is provided in a Jupyter Notebook as well as the code to analyze the WM SINDy analysis.

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

## Acknowledgements

This study acknowledges support by EU Horizon Europe program Horizon EBRAINS2.0 (101147319), Virtual Brain Twin (101137289), EBRAINS-PREP 101079717, AISN 101057655, EBRAIN-Health 101058516, EIC grant PHRASE 101058240, by the Digital Europe Programme TEF-Health (101100700), Shaiped (101195135), CoordinaTEF (101168074) German

Research Foundation SFB 1436 (project ID 425899996); SFB 1315 (project ID 327654276); SFB 936 (project ID 178316478; SFB-TRR 295 (project ID 424778381); SPP Computational Connectomics RI 2073/6-1, RI 2073/10-2, RI 2073/9-1; DFG Clinical Research Group BECAUSE-Y 504745852, Berlin University Alliance OpenMake, the Virtual Research Environment at the Charité Berlin and EBRAINS Health Data Cloud and the Berlin Institute of Health and Foundation Charité.

## Author contributions

A.K.- Preprocessing, Analysis, Code, Figures, Writing of Manuscript. A.K., E.G., P.B., K.D., K.G., S.P., S.K., and P.R.- Discussion, Revision, Comments on Manuscript. S.P., S.K., and P.R.- Advising.

## Funding

## Competing interests

The authors declare no competing interests.
