## [Transparent Peer Review file · Nature Communications]

Using an ordinary differential equation model to separate Rest and Task signals in fMRI

Corresponding Author: Dr Amrit Kashyap

Version 0:

Reviewer comments:

Reviewer #1

(Remarks to the Author)

This work describes a new analytical approach, based on ordinary differential equations (ODE) that aims to separate task-specific activity from underlying ongoing activity in fMRI data. Using this method on rest and task data from the Human Connectome Project, the authors attempt to demonstrate whether the Active Cortex Model—namely a model that “posits that the cortex is always active, and that Rest State encompasses all processes, while certain subsets of processes get elevated to perform specific task computations.” is a reasonable model of brain function. Empirically, the authors make two key observations: 1) task-specific ODEs are a subset of rest-specific ODEs, and 2) separation of these two activity components results in a 9% increase in accuracy when trying to predict trial-by-trial response times. Based on these observations, the authors conclude that indeed the Active Cortex Model is a valid model.

The manuscript is well written, and the research question this work tries to address (how to separate task specific activity from all other ongoing activity) is indeed an important one for the field. That said, I have some major concerns regarding the interpretation of results, the missing contextualization of the findings relative to relevant existing literature on the topic, and the strength of claims regarding future impact. I describe these in detail below.

MAJOR ISSUES

=====

Limited description of existing interpretation frameworks for intrinsic/rest activity

In the opening paragraph the authors state: “Resting-state, is hypothesized to exist in order to maintain the neural computational hardware rather than solving a particular problem [1]...” In my opinion, this is only partly correct. Ongoing resting-state activity is involved in many different functions. For example, Newbolt et al. (2020) recently demonstrated how activity pulses linked to neural plasticity play a role in resting-state fMRI. Pezzulo et al. (2021) has proposed that spontaneous activity is a manifestation of top-down dynamics that optimize generative models for future interactions with the environment. Gonzalez-Castillo et al. (2021) has proposed that spontaneous activity can be linked to spontaneous thought patterns and conscious experience occurring during rest. Finally, Smith et al. (2022) and Liu et al. (2023) have demonstrated that interoceptive processes are also an important aspect of spontaneous activity. In other words, spontaneous activity likely has many different roles, and not just that of “maintaining the neural computational hardware”.

Several previous studies have looked at this problem of directly determining the relationship between rest and task components in fMRI, both from a behavioral and modeling perspective. None of these are cited or discussed.

From a behavioral perspective, several groups have shown how the state of the brain at the moment of stimulus presentation can help predict behavioral outcomes (see (Ekman et al., 2012; Sadaghiani et al., 2015; Thompson et al., 2013) to name a few). Here, the authors state that once the rest component is removed, prediction of response time (the metric used here as a proxy of behavior) improves. How would the author reconcile their findings with these prior reports suggesting that the state of the system at the moment the stimulus onset matters? In other words, why if the state of the system at stimulus onset

matters, how would removing such information (e.g., what the authors call the rest component) lead to a better prediction of behavior?

From a modeling perspective, I believe it is also key to frame the current work in relation to that of Ito et al. (2020), Cole et al. (2021), He (2013), and Lynch et al. (2018). All these previous reports have looked precisely at the same challenge that the authors try to address here. Moreover, some of the prior approaches are much less computationally expensive. What it is that the ODE framework provides that these other approaches do not, and therefore justifies the need for a much more complex approach? Also, how would the ODE framework accommodate neural quenching during task (as proposed in He and Ito et al.)?

Finally, on page 5 the authors state: "Thus, we conclude that the primary organization of the Rest and Task can be approximated by the Active Cortex Model, aptly named as all processes are active during Rest and Task, but a certain subset are elevated to perform the particular Task which confirms this hypothesis that has also been stated previously in literature [15] [1]." I do not think that these two references put forward exactly this view. First, Smith et al. conclude that "the full repertoire of functional networks utilized by the brain in action is continuously and dynamically "active" even when at "rest."". They do not say anything regarding whether task performance results in elevated activity in certain sets of regions, or in suppressing of others. In other words, they only claim that the same subsystems are present in both rest and task data. Similarly, Laumann et al. main claim is that "...spontaneous BOLD activity may be more closely aligned with off-line plasticity and homeostatic processes than on-line fluctuations in cognitive content". Moreover, when it comes to task, they tend to agree with the quenching theory: "It appears likely that 'BOLD fluctuation quenching' occurs in all parts of the cerebral cortex recruited by any task, although the magnitude of the effect may be modest.". I stress this because this statement relates to the main claim of the manuscript, and I do not think these two prior papers provide additional evidence in support of the claim, as the statement suggests.

Response Time as the gold standard for model accuracy

The metric used by the authors to demonstrate the quality of separation between task and rest activity is predictability of trial-by-trial response time. Unless I missed it, I believe the author provide no reasoning for this decision, and why other metrics (e.g., response accuracy, missing trials, etc.) were not considered. Moreover, while response time is perhaps a logical choice, it is not free of potential confounds. For example, could it be that prediction of response time is improving only because the separated data contains a stronger signature of the response process associated with each trial (e.g., the button press action) due to a removal of other task-related processes of interest (e.g., stimuli perception, interpretation, decision making)? Would that be of interest or a problem?

Also, could it be that the isolated task signal leads to a better prediction because it has been "better" denoised (e.g., it's less contaminated by motion/physiological global effects)? I mention this last option because looking at figures 3.E and 8.A it looks like prior to signal separation the data contains bands of fluctuations that co-occur in time across the whole brain, and that those bands are not present in the task-isolated signals. Very similar profiles are shown by Powers et al.(2018) when discussing how physiological and motion artifacts appear in data that has not been denoised properly.

What did we learn about working memory, gambling, relational and emotional processes?

One outcome of the proposed work is the better isolation of stimulus-dependent signals, which should lead to a better understanding of how the brain reconfigures to accomplish a given task. Moreover, the authors mention: "...we found rather that Task would seem as a subset of Rest processes that can be to some degree linearly separated from one another." But what are these processes and the regions that sustain them? Which of them are common across the four tasks? Which ones are different? How any of these findings relate to prior extensive literature on these same tasks? The authors make several statements throughout the manuscript suggesting that this ODE framework brings novel insights (e.g., "While more research needs to be performed to verify the relationship between Rest and Task, this work is a significant advancement in utilizing modeling in a practical manner to isolate the stimulus component of the signal." or "...namely the Active Cortex Model and can be potentially applied to improve all Task fMRI studies, influencing the already published 100,000 Task studies."), yet it is difficult to assess the accuracy of these substantial impact claims when there is no discussion of what new knowledge we gained regarding how the brain performs these four tasks when analyzing the data here using the ODE framework.

MINOR ISSUES:
=====

Quite often the concept of "network" seems to be used as a synonym of "resting-state" (e.g., "In order to model fMRI transient dynamics directly, it would require an ODE framework that for any measurable time point of fMRI activity can estimate the network component of the fMRI timeseries.") I don't think the concept of networks is unique to resting-state, as networks are also present during task states, and communication between regions is key for accomplishing task demands. As such, I would suggest not using network and resting-state almost as synonyms in the text.

Page 4: "we apply the following three criteria to ensure that they are biophysically plausible: 1) The loss function namely how much the ODE matches the derivative of the fMRI signal 2) The degree of the Taylor series coefficients Ξ_n are related to the SC network as measured through Tractography such as in BNM 3) the Hemodynamic Response function(HRF) of the system and comparing it to the canonical HRF response." This sentence could be written more clearly. For example, how

was the loss function chosen to be physiologically plausible?

Whenever a functional connectivity matrix is shown please provide some information regarding how the ROIs are sorted. Is it randomly? Is it by hemisphere? By network?

Figure 4.B. Please add shaded regions depicting the uncertainty around the means reported as solid traces.

“The above reasoning is consistent with how GLM models are currently run, in the sense that processes that are unique to the Task activate during the presentation of stimulus, and other Rest processes that always exist would not show up as statistically significant [40].” Many current task studies no longer contrast task and rest, but two different task conditions that differ in terms of the processes being targeted by the study. This is because rest is not simply everything except the stimuli response. Please amend accordingly. Also, reference [40], which originally introduced the AFNI software back in 1996, might not be the best reference to support this claim.

Figure 7D. Two ROIs seem to have very different HRFs, particularly they seem to be the only ones showing a post-undershoot. What are these and why are they the only ones showing this property?

Prior work to consider:

=====

Cole, M.W., Ito, T., Cocuzza, C., Sanchez-Romero, R., 2021. The Functional Relevance of Task-State Functional Connectivity. *J. Neurosci.* 41, 2684–2702. <https://doi.org/10.1523/jneurosci.1713-20.2021>

Ekman, M., Derrfuss, J., Tittgemeyer, M., Fiebach, C.J., 2012. Predicting errors from reconfiguration patterns in human brain networks. *Proceedings of the National Academy of Sciences* 109. <https://doi.org/10.1073/pnas.1207523109>

Gonzalez-Castillo, J., Kam, J.W.Y., Hoy, C.W., Bandettini, P.A., 2021. How to Interpret Resting-State fMRI: Ask Your Participants. *J Neurosci* 41, 1130–1141. <https://doi.org/10.1523/jneurosci.1786-20.2020>

He, B.J., 2013. Spontaneous and Task-Evoked Brain Activity Negatively Interact. *J. Neurosci.* 33, 4672–4682. <https://doi.org/10.1523/jneurosci.2922-12.2013>

Ito, T., Brincat, S.L., Siegel, M., Mill, R.D., He, B.J., Miller, E.K., Rotstein, H.G., Cole, M.W., 2020. Task-evoked activity quenches neural correlations and variability across cortical areas. *PLoS Comput. Biol.* 16, e1007983. <https://doi.org/10.1371/journal.pcbi.1007983>

Liu, Z., Wang, X., Bazan, A.C.S., Cao, J., 2023. Advances in Resting-State Functional MRI 87–105. <https://doi.org/10.1016/b978-0-323-91688-2.00015-1>

Lynch, L.K., Lu, K., Wen, H., Zhang, Y., Saykin, A.J., Liu, Z., 2018. Task-evoked functional connectivity does not explain functional connectivity differences between rest and task conditions. *Hum. Brain Mapp.* 39, 4939–4948. <https://doi.org/10.1002/hbm.24335>

Newbold, D.J., Laumann, T.O., Hoyt, C.R., Hampton, J.M., Montez, D.F., Raut, R.V., Ortega, M., Mitra, A., Nielsen, A.N., Miller, D.B., Adeyemo, B., Nguyen, A.L., Scheidter, K.M., Tanenbaum, A.B., Van, A.N., Marek, S., Schlaggar, B.L., Carter, A.R., Greene, D.J., Gordon, E.M., Raichle, M.E., Petersen, S.E., Snyder, A.Z., Dosenbach, N.U.F., 2020. Plasticity and Spontaneous Activity Pulses in Disused Human Brain Circuits. *Neuron* 107, 580-589.e6. <https://doi.org/10.1016/j.neuron.2020.05.007>

Pezzulo, G., Zorzi, M., Corbetta, M., 2021. The secret life of predictive brains: what’s spontaneous activity for? *Trends Cogn Sci* 25, 730–743. <https://doi.org/10.1016/j.tics.2021.05.007>

Power, J.D., Plitt, M., Gotts, S.J., Kundu, P., Voon, V., Bandettini, P.A., Martin, A., 2018. Ridding fMRI data of motion-related influences: Removal of signals with distinct spatial and physical bases in multiecho data. *Proceedings of the National Academy of Sciences* 115, 201720985. <https://doi.org/10.1073/pnas.1720985115>

Sadaghiani, S., Poline, J.-B., Kleinschmidt, A., D’Esposito, M., 2015. Ongoing dynamics in large-scale functional connectivity predict perception. *Proceedings of the National Academy of Sciences* 112. <https://doi.org/10.1073/pnas.1420687112>

Smith, S.D., Nadeau, C., Sorokopud-Jones, M., Kornelsen, J., 2022. The Relationship Between Functional Connectivity and Interoceptive Sensibility. *Brain Connect.* 12, 417–431. <https://doi.org/10.1089/brain.2020.0777>

Thompson, G.J., Magnuson, M.E., Merritt, M.D., Schwarb, H., Pan, W.-J.J., McKinley, A., Tripp, L.D., Schumacher, E.H., Keilholz, S.D., 2013. Short-time windows of correlation between large-scale functional brain networks predict vigilance intraindividually and interindividually. *Human brain mapping* 34, 3280–98. <https://doi.org/10.1002/hbm.22140>

(Remarks on code availability)

Reviewer #2

(Remarks to the Author)

This study presents a novel approach to separating resting-state and task-specific fMRI signals using an ODE-based framework. The research introduces the Active Cortex Model and applies SINDy to identifying network interactions. While the paper has notable strengths, including its innovative model and application to Human Connectome Project data, several critical issues need to be addressed for publication:

- 1) The relationship between task and rest is described as linear, yet no robust justification is provided. It would be beneficial to explore non-linear alternatives, or at least discuss why a linear approach was chosen and its limitations.
- 2) The authors use SINDy to identify network dynamics from fMRI data but do not sufficiently describe how noise and artifacts from fMRI data were handled. fMRI signals can be influenced by a range of physiological factors; a more detailed discussion of preprocessing steps (e.g., denoising methods) is required.
- 3) While the paper reports a 9% increase in explanatory power across subtasks, it is unclear how this improvement compares to baseline models or other approaches in the literature. Adding comparisons with more traditional methods or competing models (e.g., DCM) would clarify the model's advantages and limitations.
- 4) The validation of the model is restricted to the data from the Human Connectome Project. For broader generalization, the authors should test the model on independent datasets or, at the very least, discuss the limitations of only using HCP data for training and testing.
- 5) One major concern is the interpretability of the results. While the authors claim to improve behavioral predictions using the model, there is limited discussion of the practical utility of these predictions. It is not evident how much this 9% increase in R^2 translates into meaningful behavioral insights. Additionally, more explanation is needed on how this model could be applied to clinical or cognitive neuroscience research.
- 6) The authors claimed in section 3.1.1 that there are 447 subjects with resting-state fMRI, and then 1032 subjects for working memory, 1013 for relational, 1017 for emotion, and 1032 for gambling. Does this mean the comparisons between resting-state fMRI and task-related fMRI are using different subjects? Some clarification is needed here.
- 7) Some literature on applying ODE to brain networks or brain graphs should be cited and discussed.

While this study addresses an important question in neuroimaging and introduces innovative modeling techniques, significant revisions are needed before the paper is ready for publication. The clarity of the methods and rationale, as well as additional validation and broader discussion of the model's implications, will greatly enhance the manuscript's contribution to the field.

(Remarks on code availability)

It would be great if the authors could provide sample input data and let the users test-run the code.

Reviewer #3

(Remarks to the Author)

This paper proposes a network Ordinary Differential Equation (ODE) to analyze fMRI data from both resting and task conditions. The model incorporates higher-order terms of the signal to capture its dynamics. By subtracting simulated resting signals from the task signals, the R2 is able to explain 9% of the observation variance compared to the unseparated data.

Among all the results, Fig. 5 (Brain to Behavior Function) directly supports the claims of the paper. By separating the resting signal, the model achieves a higher R2 score in the Working Memory Tool, Emotion Fear, Relational Texture, and Gambling Reward tasks. Additionally, the UMAP shown in Fig. 2 illustrates that, in the separated model, the functional connectivity of distinct tasks is more clearly differentiated.

Below are some suggestions for the manuscript:

1. In the methodology, the paper fits a high-order polynomial plus Gaussian noise to the derivative of the signal. It requires further explanation of why this method was chosen over others. For instance, why not use a neural network with a comparable number of unknown parameters to replace the estimated function $E(X_{b \neq a})$? It would be beneficial for the author to provide more evidence that the chosen method is superior to existing alternatives. A numerical experiment comparing the pros and cons of different methods would be helpful.
2. Still in the methodology, when separating the resting signal from the task signal, the author tests different combinations of coefficients c_1 and c_2 in Eq. 6. It would be more formal if the problem were framed as an optimization problem.
3. Regarding the results, although Fig. 5 shows higher R2 scores in the resting-task separated cases, it remains unclear why we can assume the separated signal represents the resting signal rather than noise. It is possible that the coefficient c_1 in

Eq. 6 controls the magnitude of the noise, and after the signal separation in Eq. 6, the remaining signal may be noiseless. The author should provide additional evidence to clarify this point.

(Remarks on code availability)

The code generate results of the paper. It would be better for the author to add additional comments to the code and remove some unnecessary long output.

Reviewer #4

(Remarks to the Author)

This paper develops a new network ODE model for analyzing fMRI data during rest and task. The authors use ridge regression with a polynomial dictionary to construct network ODEs. The main finding and claim of the paper is that task-specific brain activity is a subset of the broader rest state network. By isolating task-related activity from rest-related processes, the model enhances the prediction of trial reaction times by 9%. This is formulated into a theory called the Active Cortex Model, which proposes that all cortical processes are active during rest, with specific subsets becoming more prominent during task execution.

I generally like the paper and enjoyed reading it. What I liked the most was the rigor in the computational modeling. Unlike most other works in this area that choose a model structure fairly arbitrarily and fit at most a couple of parameters, this work rigorously uses data to inform almost the complete structure and parameters of their ODE model.

What I also liked was the idea of subtracting rest model from task model. It is quite impressive that this can robustly and statistically significantly improve RT predictions.

This brings me to a question/concern. Equation (6) shows the "separation" model used in the paper, and Figure 4D shows the performance of this model for different values of c_1 and c_2 . It is fairly clear from Figure 4D that increasing both c_1 and c_2 while keeping their ratio the same (such along the $c_2 = -c_1$ line) makes the performance better and better. This means the term \dot{X} in equation (6) is better to be removed (because multiplying both c_1 and c_2 by, say 2, is the same as multiplying \dot{X} by 1/2 and so on). This actually makes perfect sense. If you remove \dot{X} from equation (6), and set $c_1 = 1$ and $c_2 = -1$, the term inside the integral becomes (approximately) $\dot{X}_{\text{task}} - \dot{X}_{\text{rest}}$. After integration, this becomes $X_{\text{task}} - X_{\text{rest}}$, which is simply removing task data from rest data. Please clarify (with new results if needed) whether this simple model actually outperforms the ACM or not.

Other concerns and comments:

I'm a bit confused about what data was used to train each model. Is one model fit to data of all subjects, or one model per subject? Section 3.2.4 seems to imply that one model was fit for all subjects combined, but if so, then how can that group-level model predict RTs of individuals? Please clarify.

The paper claims to have used SINDy, but that's problematic for two reasons. First, the authors have not really used SINDy, they have used simple ridge regression. The whole selling point of SINDy is sparsity. Here the authors explicitly say that sparsity promotion using L1 regularization didn't work, so they used L2 regularization. Why then call it SINDy? Second, SINDy is actually not a good modelling technique for brain dynamics. It is hardly a good modeling technique for any real-world system. Note that sparsity is a property of the coordinate system in which a dynamical system is represented, not an invariant property of any system. Take any sparse model, do a random linear (or nonlinear) change of coordinates, and with probability 1 the resulting representation of the same system isn't sparse anymore. So I'm not surprised at all that SINDy hasn't worked here, and I'm confused why the authors still want to call their approach SINDy.

In Figure 4D, what is "Delta Max Correlation" on the color axis? I can't find a proper definition of it anywhere.

Figure 2 top left: I don't understand the Venn diagram. The equation below the Venn is also a bit confusing the way it is written, but later when you read equation (6)

Figure 2 bottom right: It's not entirely clear to me that different task point clouds are more separate in the right panel than the left panel. Especially with a 3D plot, who knows, maybe if you rotate the left one there is a view angle that looks better. I think you need to come up with a quantitative measure (look at the unsupervised clustering literature) and compare it between the 2 panels, instead of just relying on eyeballing.

Figure 7: are these results on rest or task data?

The notation in Equation (6) is a bit loose. Please update the writing of the equation to clarify what's integrated over (dt), the limits of the integral, and enclose the integrand in brackets to clarify what is exactly integrated.

(Remarks on code availability)

Version 1:

Reviewer comments:

Reviewer #1

(Remarks to the Author)

I would like to thank the authors for taking the time to answer my previously raised concerns in detail. As of now, the only previous concern that remains unresolved is that of interpretability and novel insights.

To ameliorate this concern, the authors have performed an additional experiment contrasting ODE results for Working Memory under two different levels of difficulty (0-back vs. 2-back) and shown that such a contrast leads to more specific activations patterns as compared to (2-back vs. rest). Yet, that is what happens on a regular GLM framework; which brings me back to my original comment/question: what is it that we can learn about how the brain responds to a given task that other simpler methods cannot uncover? I believe that clearly exemplifying the power of this method for generating new knowledge beyond that of what more traditional approaches (which are much less computationally expensive and require much less data) can do would be critical for its successful adoption by the community.

I have two minor new concerns. First, the manuscript contains numerous typos, repetitions, unformatted latex, etc. (see a few examples below), which not only clouds interpretation but makes one feel like the writing was quite rushed. Second, likely due to the number of issues raised by reviewers, the manuscript has grown not only in size, but also in complexity and technicality; perhaps making it less accessible to a broader audience. I think that any efforts to amend this could be clearly beneficial for the quality of the text and its impact.

Examples of typos, repetitions, etc.

! appears instead of < when reporting p-values

the only the ACM framework (repeated "the")

Sometimes the authors refer to Working Memory as WM, sometimes is fully spelled.

using model F using complex functions (repeated "using")

open source open source Python (repeated "open source")

Our results also align with prior models of Task using Generalized Linear Models (GLM) have assumed that (missing tat before have)

"It has limited the scope" to "this has limited the scope"

Sometimes you use Metatasks, sometimes meta-task. The inconsistent use of hypens applies to other terms.

$\$c_1$, $\$c_2$ not rendered properly.

(Remarks on code availability)

Reviewer #2

(Remarks to the Author)

The authors have addressed all my concerns, and I am satisfied with the current version for acceptance.

(Remarks on code availability)

Reviewer #3

(Remarks to the Author)

The authors addressed my questions point by point. Below are my opinions regarding their responses and the paper:

1. The authors provide a justification for using a high-order polynomial model compared to alternative methods. They argue that the since the field is still relatively new, no existing approach fully replicates their results. They also discuss existing methods such as DCM, asserting that their approach offers superior scalability and correlation accuracy. Additionally, they reference deep learning-based techniques and highlight current limitations in convergence and interpretability. While the response addresses my concerns, it lacks enough empirical and theoretical proofs for demonstration of why their chosen model outperforms other possible approaches.

2. The authors acknowledge that Eq. 6 can be reframed as an optimization problem, allowing for a more formal approach to solving for the coefficients c_1 and c_2 . The primary challenge lies in incorporating the Brain-to-Behavior function G , which maps neural signals to reaction time (RT). They have expanded the discussion in Section 4.2 (Limitations), recognizing that future studies could explore nonlinear separation schemes and optimization techniques to refine this approach. From my perspective, the work still requires a more rigorous equation formulation and a more robust solver to make sure their choice of the method is proper and to make sure the result is convincing.

3. Regarding whether the separated signal genuinely represents resting activity or is merely noise, the authors argue that their method does not perform traditional denoising, as evidenced by an increase in both Functional Connectivity (FC) and signal variance after separation. To validate that the transformation is not simply removing noise, they compare their

separated signal to a matched random signal with the same mean and variance as the Active Cortex Model (ACM). The results indicate that the separated signal preserves structured network properties, whereas subtracting a purely random signal does not. Furthermore, the authors compare their approach to ICA, PCA, and filtering techniques, which typically reduce variance by removing structured components. In contrast, their method increases variance while still improving behavioral correlates, reinforcing that the separation process is not merely noise removal. However, a remaining issue is the lack of statistical tests to rigorously validate their argument.

(Remarks on code availability)

The authors provide a jupyter notebook to visualize their results appearing in the article. They also provide a file for data preprocessing. A readme is needed to understand how to use them to replicate the results.

Reviewer #4

(Remarks to the Author)

All my comments are addressed. Thanks to the authors for the careful revision.

(Remarks on code availability)

Reviewer #1 (Remarks to the Author):

This work describes a new analytical approach, based on ordinary differential equations (ODE that aims to separate task-specific activity from underlying ongoing activity in fMRI data. Using this method on rest and task data from the Human Connectome Project, the authors attempt to demonstrate whether the Active Cortex Model—namely a model that “posits that the cortex is always active, and that Rest State encompasses all processes, while certain subsets of processes get elevated to perform specific task computations.” is a reasonable model of brain function. Empirically, the authors make two key observations: 1) task-specific ODEs are a subset of rest-specific ODEs, and 2) separation of these two activity components results in a 9% increase in accuracy when trying to predict trial-by-trial response times. Based on these observations, the authors conclude that indeed the Active Cortex Model is a valid model.

The manuscript is well written, and the research question this work tries to address (how to separate task specific activity from all other ongoing activity) is indeed an important one for the field. That said, I have some major concerns regarding the interpretation of results, the missing contextualization of the findings relative to relevant existing literature on the topic, and the strength of claims regarding future impact. I describe these in detail below.

We thank the reviewer for their detailed review of our manuscript. Below are a point by point response and the changes detailed in the manuscript that we hope will address the reviewers concerns.

MAJOR ISSUES

=====

Limited description of existing interpretation frameworks for intrinsic/rest activity

In the opening paragraph the authors state: “Resting-state, is hypothesized to exist in order to maintain the neural computational hardware rather than solving a particular problem [1]...” In my opinion, this is only partly correct. Ongoing resting-state activity is involved in many different functions. For example, Newbolt et al.

(2020) recently demonstrated how activity pulses linked to neural plasticity play a role in resting-state fMRI. Pezzulo et al. (2021) has proposed that spontaneous activity is a manifestation of top-down dynamics that optimize generative models for future interactions with the environment. Gonzalez-Castillo et al. (2021) has proposed that spontaneous activity can be linked to spontaneous thought patterns and conscious experience occurring during rest. Finally, Smith et al. (2022) and Liu et al. (2023) have demonstrated that interoceptive processes are also an important aspect of spontaneous activity. In other words, spontaneous activity likely has many different roles, and not just that of “maintaining the neural computational hardware”.

We thank the reviewer for providing relevant context to the discussion of resting state dynamics. We have revised the first paragraph in the introduction to reflect the complexity of the origins of resting state:

One of the main successes of fMRI is measuring noninvasively how the cortex responds to particular stimuli and performs cognitive tasks, i.e., Task fMRI. Over 100,000 Task studies have already been performed and analyzed as of 2023 to understand how different components of the nervous system interact to perform certain behavioral Tasks and answered a diverse set of cognitive and clinical questions (Acar et al., 2023). However, a key challenge in these studies has been characterizing the changes in activity triggered by task onset relative to the large nonstationary baseline cortical activity (Wobst et al., 2001, Berkovich-Ohana et al., 2016). This baseline activity, also known as Resting State fMRI has been studied in absence of stimulus and portions of the activity have been linked to numerous different sources such as unobserved spontaneous thought processes or the introspective processes (Gonzalez-Castillo et al., 2021, Smith et al., 2022). However, considering the substantial energy demands of activating large neural regions and the persistent rhythmic nature of these resting-state processes during Task activation, studies have suggested that these processes may rather play a role in homeostatic functions, such as priming plasticity, maintaining functional interactions between regions, and optimize generative models for future interactions with the environment (Laumann et al., 2021, Pezzulo et al., 2021). To explain Task activation, studies have proposed that certain Resting State sub-networks are functionally activated with a higher weight during Task activation, while others have also shown that sub-networks of the white matter structural connectivity are predictive of Task performance (Cole et al., 2021, Yan et al., 2021). However, there is uncertainty of what occurs to the other Rest processes within Task as many of these Resting State networks are also shared with the Task networks (Smith 2009). Mechanistically, these quasi-rhythmic processes observed in Rest are thought to originate from intrinsic cortical loops that exist in the white matter networks that are stimulated by spontaneous activity and thought to persist during Task processes (Thompson et al., 2014, Kashyap et al., 2019). Determining

this exact relationship between Rest and Task, would not only further our fundamental understanding of how the nervous system operates, but would also have practical consequences in improving our ability to analyze and decode Task fMRI activity. In the following study, we extend our current generative Ordinary Differential Equations (ODE) models for fMRI activity, and model the components of the Rest signal during the Task activation (Ritter et al., 2013). Our results support the notion that Task processes are a subset of Rest processes and demonstrate a practical approach to isolate Task-specific signals with stronger correlations to behavioral measures across various tasks.

Several previous studies have looked at this problem of directly determining the relationship between rest and task components in fMRI, both from a behavioral and modeling perspective. None of these are cited or discussed.

From a behavioral perspective, several groups have shown how the state of the brain at the moment of stimulus presentation can help predict behavioral outcomes (see (Ekman et al., 2012; Sadaghiani et al., 2015; Thompson et al., 2013) to name a few). Here, the authors state that once the rest component is removed, prediction of response time (the metric used here as a proxy of behavior) improves. How would the author reconcile their findings with these prior reports suggesting that the state of the system at the moment the stimulus onset matters? In other words, why if the state of the system at stimulus onset matters, how would removing such information (e.g., what the authors call the rest component) lead to a better prediction of behavior?

We thank the reviewer for bringing our attention to the notion of the initial state of the system and how it has been used in the past to predict Task responses. We see our method as an extension of these previous methods. The Ekman and Sadaghiani papers use long intervals between stimuli in order to construct FC as a measure for the state of the system prior Task onset and compare its relationship with the accuracy of subsequent trials. In our case, the immediate activity preceding the Task interval is used as an initial condition to predict subsequent trajectories based on our models. However, our results show that by subtracting the baseline activity by using either the Rest models or Task models we lose information, resulting in a decrease to behavioral variables (See Task or Rest Baseline Model in Figure 4 and 6). Only when specific trajectories are removed, that are present in Rest and not contained in the Task, results in better prediction of behavior. Therefore, we agree that the information present at stimulus onset plays a crucial role in predicting the subsequent trial, and removing trajectories without considering the common processes in Rest and the specific Task diminishes the prediction of behavior.

We also acknowledge that there is also a difference between using the dynamical functional connectivity vs the initial conditions although these processes might be related. The functional connectivity represents a measurement of the underlying ODE, and the ODE can change over time, whereas the initial conditions are the immediate activity prior to Task onset and can come from either a stationary or non-stationary ODE. Therefore, it could be that their findings relate to changes in performance to the instantaneous ODE rather than to the activity before Task. In Sadaghiani et al., 2015 the authors suggest that the changes in Task performance are due to "moment-to-moment dynamic changes" which could be interpreted as the derivative prior to Task onset. However, to fully disentangle these two sources of variance initial condition vs the state of the ODE at stimulus onset, future studies would be necessary to replicate the long task interval experiment with ODEs. These studies would have to create an instantaneous individual Rest model representing the state of the ODE at Task onset using the long intervals between stimulus recordings and test whether it improves decoding vs using a stationary individual Rest model. While we believe that both would account for a portion of the variance associated with behavioral performance, we speculate that the Rest model is mostly stationary, as we have observed that it generalizes robustly across the HCP population, and the initial conditions not the instantaneous state of the ODE would account for the larger slice of variance.

We have added the following paragraph to the discussion section:

Another intriguing approach, distinct from the classical GLM paradigm for analyzing task activity, involves designing studies with long intervals between task stimuli to better account for the system's state at task onset (Ekman et al., 2012, Sadaghiani et al., 2015). These long intervals are used to construct FC preceding stimulus onset and compare its relationship with the accuracy of subsequent trials. In our case, the immediate activity preceding the Task interval is used as an initial condition to estimate a trajectory based on our models. We agree with the conclusions of these previous papers that this initial state is important, as by removing trajectories without accounting for the shared processes between Rest and Task only results in a decrease in association with behavior as can be seen by our Rest and Task Baseline models. We only observe increases to behavioral predictions by subtracting selective trajectories associated with Rest that are absent in the current Task. Therefore, the initial conditions and our approach which generalizes to trajectories suggest that the moment of Task onset plays an important role in predicting Task performance.

From a modeling perspective, I believe it is also key to frame the current work in relation to that of Ito et al. (2020), Cole et al. (2021), He (2013), and Lynch et al. (2018). All these previous reports have looked precisely at the same challenge that the authors try to address here. Moreover, some of the prior approaches are much

less computationally expensive. What it is that the ODE framework provides that these other approaches do not, and therefore justifies the need for a much more complex approach? Also, how would the ODE framework accommodate neural quenching during task (as proposed in He and Ito et al.)?

We thank the reviewer for introducing these other works especially Ito et al. and Cole et al. which are relevant to the discussion. The equation that Cole et al. (2016) uses to validate the activity flow map is identical to BNM equations (See Sanz Leon et al. 2015, or Ritter et al. 2013 or Breakspear 2017) as well as the equation we identify via regression. All of them show a sigmoidal activation function and a diagonal decay (corresponding to the HRF) and network weights that align with the tractography:

(Equation From Cole et al 2016):

$$\tau_i \frac{dx_i}{dt} = -x_i + f_i \left(\sum_{j=1}^n w_{ji} x_j + bias_i \right) \quad i = [1..n]$$

The function f is modeled as a sigmoidal function. This equation is used as the starting point to derive the simpler activity flow in Cole et al., 2016 to model Task Switching. The equation uses weighted activity of different functional networks are used to construct Task activation:

(Equation From Cole et al 2016):

$$P_j = \sum_{i \neq j \in V} A_i F_{ij}$$

Our approaches are closely related since we identify a very similar equation directly from Resting State Data using regression. The edge functions in Figure 3 show that regression results in sigmoidal activation functions to represent network activity with the edge weights related to the structural connectivity (we have below shown the comparison between Figure 6 of Ito et al and our own edge functions). The regression also identifies that the decay rate rather than set to a linear slope of -1 in previous literature, has a slope related to time constant λ and the activation function for the decay rate is also identified to be sigmoidal in nature. We also show that we can approximate the correct time constant λ to match the measured decay term in observed HRFs. Therefore, since Cole et al., and Ito et al., use a very similar framework to derive their activity flow, their results should hold in our case as well, including the stimulus quenching which would

naturally occur due to the presence of the sigmoidal functions. Moreover, the notion of weighted subnetworks occurring in Task which has been also shown using a structural white matter study (Yan et al., 2021) in congruent with the Active Cortex Model, which assumes that Task processes are a subset of Rest processes. Therefore, in our view, our approaches are aligned, and the results from both studies reinforce each other's claims, as we derive these equations directly from the data through regression, which aligns with previously established findings in the literature.

Caption: The light blue interhemispheric connections exhibit the same sigmoid structure as those reported in Cole et al. and Ito et al. (from Figure 3D). The x-axis represents the input from a brain region, while the y-axis represents its subsequent derivative. The sigmoidal activation function then gives the equivalent to the "output" of the region that influences which causes the change in the derivative of its neighbors. Given that both studies assume similar equations to model brain dynamics, the results from one study are expected to hold for the other.

Moreover, the ODE framework described in this manuscript allows us to go beyond these previous models that have been described. This methodology extends and builds upon previous modeling work in the following ways:

1. Validation of the equations previously assumed for fMRI dynamics by applying the well-established and computationally efficient method of ridge regression.
2. A practical approach to applying the ODE theory, in the form of direct manipulations of fMRI time series, validated by observing improvements in behavioral predictions.
3. The dynamics of both Rest and Task states are determined by regression, allowing for one of the strongest conclusions so far about their relationship to be drawn.

The major drawback of previous ODEs is that they are simulated in their own space rather than constructed to fit fMRI activity (both traditional BNM as well as the models used by

Cole and Ito et al). This has limited to the analysis of steady state dynamics, through the use of FC and observing properties in Rest and Task states rather than establishing a rigorous relationship between them.

Moreover, it avoids pitfalls of falsely assuming properties that might not be necessarily correct. For example, our approach has revealed different associations between the interactions of hemispheres that has previously been assumed. Moreover, it also the steady state decay rate with a fixed time constant that might not be appropriate for modeling fMRI. Therefore, we see our approach as a natural extension of Cole et al, Sanz Leon et al, that allows the scientific community to answer a broader range of questions than before and test our assumptions of our beliefs of the underlying ODE that governs fMRI dynamics.

We thank the reviewer in bringing up these relevant studies and have incorporated them into our discussion in a number of different:

In the discussion section Active Cortex Model:

Other works have extended BNM to task paradigms by constructing activity flow maps using FCs to predict activations of sub-networks from the Resting State functional networks (Cole et al., 2016, Ito et al., 2020, Cole et al., 2021). Separately, structural sub-networks of the global network have also been shown to be predictive for Task activation (Yan et al., 2021). Our results are consistent with the notion of sub-network activations during Task which would yield the relationship seen in the ACM, namely that Task processes are a subset of Rest processes. These works have, along with other works, have also proposed the notion of stimulus quenching during Task activation. Specifically, during Task activity the system approaches its sigmoidal maximum which lead the dynamics to be quenched (Cole et al., 2021, Laumann et al., 2021). Our results largely agree with these prior works, as the identified equations have a similar sigmoidal relationship and thus would lead to the similar dynamics as proposed by the stimulus quenching hypothesis.

In the discussion section comparison to other models:

They have however, uncovered properties of Task trials (Schirner et al., 2023) while others have derived simpler activity flow models from the BNM underlying equations to model Rest Task switching as activations of sub-networks of Resting State functional networks (Cole et al., 2021 Ito et al., 2020).

Finally, on page 5 the authors state: “Thus, we conclude that the primary organization of the Rest and Task can be approximated by the Active Cortex Model, aptly named as all processes are active during Rest and Task, but a certain subset are elevated to perform the particular Task which confirms this hypothesis that has

also been stated previously in literature [15] [1].” I do not think that these two references put forward exactly this view. First, Smith et al. conclude that “the full repertoire of functional networks utilized by the brain in action is continuously and dynamically “active” even when at “rest.””. They do not say anything regarding whether task performance results in elevated activity in certain sets of regions, or in suppressing of others. In other words, they only claim that the same subsystems are present in both rest and task data. Similarly, Laumann et al. main claim is that “...spontaneous BOLD activity may be more closely aligned with off-line plasticity and homeostatic processes than on-line fluctuations in cognitive content”. Moreover, when it comes to task, they tend to agree with the quenching theory: “It appears likely that ‘BOLD fluctuation quenching’ occurs in all parts of the cerebral cortex recruited by any task, although the magnitude of the effect may be modest.”. I stress this because this statement relates to the main claim of the manuscript, and I do not think these two prior papers provide additional evidence in support of the claim, as the statement suggests.

We thank the reviewer for their insightful discussion in contextualizing the ACM model. After reviewing through the previous literature on Rest-Task activation we believe that the results that we show using ACM most strongly supports the conclusions of Cole et al., 2016 and 2021 where the activity flow maps represent Task as a weighted subnetwork of Rest functional networks. Moreover the work by Yan et al, 2021 also show that subcomponents of the white matter structural networks are predictive of Task responses.

Therefore, we have revised the following sentences in the introduction section on Active Cortex accordingly:

To explain Task activation, studies have proposed that certain Resting State sub-networks are functionally activated with a higher weight during Task activation, while others have also shown that sub-networks of the white matter structural connectivity are predictive of Task performance (Cole et al., 2021, Yan et al., 2021).

This supports the conclusion of previous literature that have modeled Task activity as weighted components of functional and structural subnetworks of Rest processes (Cole et al., 2021, Yan et al., 2021).

Furthermore we have rewritten the discussion section pertaining to the ACM:

The results of the Active Cortex model modify our understanding of the extensively studied Rest and Task relationship. Early DCM studies characterizing and modeling Rest and Task states represented Task networks by adding a component to the existing Resting State networks while simultaneously solving for external stimulus input (Friston

2003 Friston et al., 2014). BNM models used a fixed structural network model but simulated input stimulus using Gaussian processes (Sanz-Leon et al., 2015, Breakspear 2017). Other studies have refined the concept of a fixed network by introducing weighted sub-networks derived from activity flow maps, using functional connectivity (FC) to predict the activation of sub-networks from Resting State functional networks (Cole et al., 2016, Ito et al., 2020, Cole et al., 2021). Separately, structural sub-networks of the global structural network derived from tractography have also been shown be predictive for Task activation (Yan et al., 2021). These works have also along with other works, have proposed the property of the stimulus quenching, that is, during Task activity the system approaches its sigmoidal maximum and the dynamics is quenched (Cole et al., 2021, Laumann et al., 2021).

In our experimental paradigm, the ACM aligns with the idea that specific Resting State sub-networks are elevated to facilitate task performance. The common network processes, primarily captured by the Rest ODE, can only be removed from the signal if the distinct network processes, represented by the Task ODE, remain intact. We demonstrate this across five behavioral variable tests and fourteen subtasks spanning four metatasks. Additionally, we show that functionally similar tasks yield ODEs with more structurally similar coefficients (see Figure 9), allowing for greater interchangeability and improved isolation of task-specific processes via ACM separation. This is further supported by our use of meta-task models to decode individual subtasks, where we only had sufficient data to train meta models but were able to apply to each of these subtasks individually. These findings suggest the existence of a fixed set of sub-networks, with each task selectively engaging specific processes for task execution. Furthermore, we identify sigmoidal relationships between network regions, consistent with previous literature, which could lead to stimulus quenching when these sub-networks are elevated in activity to perform a given task.

Response Time as the gold standard for model accuracy

The metric used by the authors to demonstrate the quality of separation between task and rest activity is predictability of trial-by-trial response time. Unless I missed it, I believe the author provide no reasoning for this decision, and why other metrics (e.g., response accuracy, missing trials, etc.) were not considered. Moreover, while response time is perhaps a logical choice, it is not free of potential confounds. For example, could it be that prediction of response time is improving only because the separated data contains a stronger signature of the response process associated with each trial (e.g., the button press action) due to a removal of other

task-related processes of interest (e.g., stimuli perception, interpretation, decision making)? Would that be of interest or a problem?

We agree with the reviewer using Reaction Time as a gold standard has its own list of potential confounds. We have included in our revision predictions across the different frameworks for accuracy as well as missing trials as the reviewer has suggested (See Figure below or Supplemental Figure 21). The accuracy and trial completion percentages are both calculated across each individual across the trials. We estimated the FC for each separation strategy (Unseparated, ACM, Rest Baseline, Task Baseline) and mapped the resulting dynamic FCs to these continuous time predictions using our Elastic Net (Fong et al., 2019). These were the same FCs that showed separation in the UMAP space in Figure 6. Our results shows that while we observe the effects of ACM in predicting missing trial or accuracy, they are much less pronounced to our other results using RT.

Fig. 21 Predicting Missing Trials and Accuracy for WM trials across different frameworks. The percentage of missing trials and accuracy was calculated for each individual across both WM runs. Functional connectivity (FC) was computed for all the different separation strategies, as visualized using the UMAP in Figure 6. The FCs were then mapped to the percentage of missing trials and accuracy using an Elastic Net. The results of a 10-fold cross-validation scheme are shown above for each framework strategy. The ACM performed significantly better at predicting both missing trials and accuracy, although the difference is smaller in magnitude compared to the differences observed in RT.

In our revision, we have also extended these results to another behavioral variable namely the single trial classification of each Task category using a 10-fold cross validation

scheme in conjunction with a Random Forest Classifier (Figure below or Supplemental Figure 19C). This shows a much larger 15 percent classification difference between the unseparated and separated (ACM) datasets ($p < 0.00001$).

In total now we have shown that the ACM separation strategy improves at least 5 different behavioral predictions: 1) RT prediction on a per trial basis 2) Task classification on a per trial basis 3) Divergence of Task FCs on a per individual basis 4) Accuracy prediction on a per individual basis 5) Missing Trials prediction on a per individual basis.

After completing all these experiments, we speculate that for this particular experimental paradigm that Reaction Time is probably the most appropriate as it provides 1) a continuous behavioral variable that we can map changes from our continuous timeseries predictions 2) It is quite ubiquitous across Tasks, allowing for comparison of results across Tasks.

Supplemental Figure 19C - The separability was also tested using the individual trials from each category, rather than the FC. A random forest classifier using 10-fold cross validation testing split was used to train on each type of dataset vs separation framework. The ACM once again is the only tested framework that improves from the baseline accuracy by 15 percent while other frameworks have no benefit.

In the revision, in addition to adding the single trial classification to Supplementary Figure 19 and the accuracy and mistrial classification in Supplementary Figure 21, we have added the following sentences in the method discussing using our behavioral variables used in the study:

Overall, we test the separation frameworks on five different behavioral variables: 1) RT prediction on a per trial basis 2) Task classification on a per trial basis 3) Divergence of Task FCs on a per individual basis 4) Accuracy prediction on a per individual basis 5) Missing Trials prediction on a per individual basis.

To classify the single trial trajectories (Supplemental Figure 19C), we applied our separation scheme similar to the RT analysis to generate timeseries from stimulus onset

for each trial. We then took only the valid trials and classified them by their Task category (GB, WM, REL, EM) using a Random Forrest Classifier (200 Trees).

We also tested if these FCs contained pertinent Task information after the separation such as the accuracy of the participant or the percentage of trials they completed. We used the Elastic Net (alpha = 5, L1 ratio = 0.01) described in the section above to map the FCs to the both the accuracy percentage and the completion percentage for WM trials shown in Supplemental Figure 21.

We also changed the following sections in the Results based on our new behavioral results.

We also tested if this increase in Task differentiability translates on a per trial instance in Supplementary Figure 20. Using a Random Forrest Classifier, we tested how well we can classify the separated vs unseparated trails with respect to their meta Task categories. Once again, the only the ACM framework showed a significant increase in classification accuracy of 15 percent ($p < 0.00001$) compared to the unseparated cases. The other frameworks showed either no change or got worse after separation.

To evaluate whether the separated FCs retained important individual characteristics, such as overall accuracy across the task or the percent of task completion, we tested different frameworks to assess their ability to predict these variables. In Supplementary Figure 21, we demonstrate that for the Working Memory (WM) task, ACM is the only tested strategy that improves the prediction of both accuracy and task completion. The differences are smaller in magnitude compared to those observed in RT, as the separation effects are averaged across all trials. However, the general trends seen with other behavioral measures remain significant, with ACM consistently outperforming the other frameworks.

Also, could it be that the isolated task signal leads to a better prediction because it has been “better” denoised (e.g., it’s less contaminated by motion/physiological global effects)? I mention this last option because looking at figures 3.E and 8.A it looks like prior to signal separation the data contains bands of fluctuations that co-occur in time across the whole brain, and that those bands are not present in the task-isolated signals. Very similar profiles are shown by Powers et al.(2018) when discussing how physiological and motion artifacts appear in data that has not been denoised properly.

We thank the reviewer for bringing to our attention the interpretation of our algorithm with relation to denoising algorithms. As shown in the first figure below (right), the variance of the signal *increases* after separation, which would not be expected if the algorithm were simply performing a denoising of the timeseries. The covariance, or the FC matrices also show an increase in variance,

which we have visualized with our UMAP in Figure 6. From the figure, the transformation is most analogous to subtracting a random signal with the same mean and variance as the ACM which increases the amount of noise in the signal. This was performed to match our separation transformations, but this is equivalent to adding noise, as the noise is drawn from a Gaussian process symmetric around the origin. Additionally, this increased variance aligns with an improvement in behavioral correlation to RT, as demonstrated in the first figure below (left). Another perspective that may clarify the separation process is the observed changes in functional connectivity, as illustrated in Figure 3 EF (second figure below). The network signal captures the significant resting-state variations it was trained on. Subtracting the network signal from the measured signal decreases functional connectivity by removing the components representing structured processes, leaving the unstructured components intact. Consequently, this transformation eliminates structured processes, increases signal variance, and which we observe aligns more closely with behavioral responses. The ACM highlights the importance of selectively removing unrelated structured processes, as only this targeted approach that we tested that enhances behavioral correlates.

Supplemental Figure 7- The Separation of the signal using various models. M. stands for Measured. The signal separation using both task and rest ODE, the Active Cortex Model (Rest - Task), is the only separation that yields significantly larger behavioral correlates ($p < \$0.05$) than the original Measured Signal. The matched signal represents subtracting the measured signal using a random signal with the same mean and variance as the Active Cortex Model. The figure on the left shows that the variance increases significantly ($p < 0.05$) for the Active Cortex Model after subtraction compared to the original and matched model.

Figure 3 EF- An example of separating the measured signal derivative into network and stimulus components normalized to have the same amplitude. F) The network structure is preserved in our model which has similar FC to the measured FC (~0.7 correlation) The residual signal which would be the result of our separation scheme has an increased variance and decreased relationship to the signal in other brain areas.

We have the following sentences in the discussion with our models relation to ‘denoising’ algorithms:

In relation to denoising algorithms such as ICA, PCA, filtering our model increases the signal variance unlike those transformations (see Supplementary Figure 16) (Hutchison et al., 2013, Allen et al., 2014). These increases in signal variance are most similar to adding noise into the signal, but this increase in variance is related to improvement in behavioral correlates in regions corresponding to the Task.

What did we learn about working memory, gambling, relational and emotional processes?

One outcome of the proposed work is the better isolation of stimulus-dependent signals, which should lead to a better understanding of how the brain reconfigures to accomplish a given task. Moreover, the authors mention: “...we found rather that Task would seem as a subset of Rest processes that can be to some degree linearly separated from one another.” But what are these processes and the regions that sustain them? Which of them are common across the four tasks? Which ones are different? How any of these findings relate to prior extensive literature on these

same tasks? The authors make several statements throughout the manuscript suggesting that this ODE framework brings novel insights (e.g., “While more research needs to be performed to verify the relationship between Rest and Task, this work is a significant advancement in utilizing modeling in a practical manner to isolate the stimulus component of the signal.” or “...namely the Active Cortex Model and can be potentially applied to improve all Task fMRI studies, influencing the already published 100,000 Task studies”), yet it is difficult to assess the accuracy of these substantial impact claims when there is no discussion of what new knowledge we gained regarding how the brain performs these four tasks when analyzing the data here using the ODE framework.

We thank the reviewer for bringing up this important issue in the interpretability of the different identified ODEs. In terms of which processes are common and which are different we have written the following sentence in the Discussion:

The common network processes, primarily captured by the Rest ODE, can only be removed from the signal if the distinct network processes, represented by the Task ODE, remain intact.

Additionally, to gain more insight during Task have also done an experiment contrasting the different ODE Task models shown in Figure 9, in terms of their shared dynamics, shared inferred network structure, and applying one Task model to decode another. We added to the Discussion section summarizing our results:

Additionally, we show that functionally similar tasks yield ODEs with more structurally similar coefficients (see Figure 9), allowing for greater interchangeability and improved isolation of task-specific processes via ACM separation. This is further supported by our use of meta-task models to decode individual subtasks, where we only had sufficient data to train meta models but were able to apply to each of these subtasks individually. These findings suggest the existence of a fixed set of sub-networks, with each task selectively engaging specific processes for task execution.

To gain deeper insight into which neural processes are affected, we have incorporated the reviewer's suggestions regarding the use of GLM-based task contrasts to differentiate regions involved in core task processing from those necessary for task execution (e.g., motor cortex for button pressing) (Monti et al., 2011). In our revision, we include an additional analysis and Supplementary Figure 20, where we compare the task severity paradigm with brain regions that exhibit improved behavioral correlation following ACM separation in the working memory (WM) task. Specifically, we contrast the easier 0-back tasks with the more demanding 2-back tasks. The 2-back tasks show greater trial-by-trial variability, leading to lower correlations with reaction time (RT) measures compared to 0-back tasks.

The regions showing the largest task contrast effects are prefrontal and parietal regions, which are well known for their role in working memory. In contrast, regions common to both tasks, such as the visual and motor cortex, exhibit little difference in task severity effects. Notably, these task severity-related regions also show the greatest normalized improvement in correlation following ACM separation ($r^2 = 0.79$). This improvement is more aligned with the task contrast network than with either the 0-back ($r^2 = 0.75$, $p < 0.003$) or 2-back networks ($r^2 = 0.76$, $p < 0.002$). Thus, our approach enhances signal resolution specifically in regions involved in higher-order cognitive processing, rather than in areas associated with motor execution, where ACM separation does not significantly improve RT correlations.

Task Activation Maps Relationships

Fig. 20 Top Left: The activation Map from ACM based on the max correlation of a ROI across time for 0 back working memory. Top Right: The activation Map from the unseparated data based on the max correlation of a ROI across time for 0 back working memory. Bottom Left: The normalized change in correlation between the ACM (separated) and the unseparated cases showing a subset of regions that were originally identified in the Task increasing in correlation. Bottom Right: The difference between the activation maps based on correlation to RT for the zeroback and twoback. The zeroback trials are more correlated with the RT, and the difference between them removes associations between regions that are necessary for the Task such as the visual cortex but do not reflect the Task severity. The change in separated vs unseparated cases match the difference between tasks suggesting that the increases in correlation are in regions that are specific to the particular difficulty of the Task, not just regions that activate for Task completion such as the visual cortex to see the input or the motor cortex for button pushing. Significance Test: Norm Δ RT Corr to 0 vs 2back = 0.79 R^2 Norm Δ RT Corr to ACM RT Corr = 0.75, Norm Δ RT Corr to Unseparated RT Corr = 0.76. P-value difference between 0.79 > 0.75 is 0.002 and 0.79 > 0.76 is 0.003.

We changed the following in the Brain to Behavior Results section:

In this section, the overall changes to RT are quantified by aggregating the spatial temporal information, since the correlation at each individual region increases separately. In Figure 5A Left, we can observe that specific ROIs are moderately correlated with each unique Task. The y-axis refers to the maximum correlation computed over time for each individual ROI. These areas are very similar to previous identified areas as shown in

Supplemental Figure 15 (Barch et al., 2013, Curtis et al., 2019, Tozzi et al., 2020, Assem et al., 2020). The ACM increases the correlation with a subset of these regions as can be seen in Figure 5A Right. These identified regions that show a change with respect to RT, for example in WM, are more associated with memory areas in the prefrontal, rather than the visual cortex, which is recruited for the overall Task. In Supplemental Figure 20, we show for WM that these regions are significantly closer to the severity of the Task rather than all the regions needed to perform the Task.

In the discussion section pertaining to the Active Cortex Model we have also briefly discussed the isolation of correlation increases to regions that are different for Task severity:

Prior models of Task using Generalized Linear Models (GLM) have assumed that Task structure is replicable, whereas Rest processes are more randomly distributed (Cox, 1996; Monti et al., 2011). This assumption has allowed these models to statistically identify the regions where the Task occurs. This aligns with our approach of training separate models for Rest and Task, where during Task, Rest processes that are not associated with the Task appear to be unstructured processes. GLM models also utilize two Task conditions with varying difficulty levels to isolate primary cortical regions responsible for the Task, rather than all regions necessary for Task execution, such as the visual cortex or motor areas involved in button pressing (Monti et al., 2011). In our variant of this experiment, by contrasting regions with higher correlation during 0-back vs. 2-back, we identify regions that match Task severity. These regions closely align with areas that show the greatest normalized increase to behavioral performance due to ACM separation (see Supplementary Figure 18). This suggests that ACM selectively enhances areas responsible for Task severity and difficulty rather than all regions involved in Task execution.

In terms of impact, we believe our work significantly advances the understanding of task subnetworks as functional subsets of resting-state subnetworks. Beyond validating this hypothesis, we demonstrate the ability to isolate task-specific signals across 14 subtasks and five behavioral measures. Since removing unrelated resting-state activity has long been a fundamental goal in task-based analyses, our approach represents a paradigm shift in how task data is analyzed. While the specifics of the separation process may evolve, we are confident based on both prior literature and our results that the Active Cortex Model accurately describes the relationship between Rest and Task processes and will play a crucial role in future research aimed at analyzing, isolating, and interpreting Task activity.

MINOR ISSUES:

=====

Quite often the concept of “network” seems to be used as a synonym of “resting-state” (e.g., “In order to model fMRI transient dynamics directly, it would require an ODE framework that for any measurable time point of fMRI activity can estimate the network component of the fMRI timeseries.”) I don’t think the concept of networks is unique to resting-state, as networks are also present during task states, and communication between regions is key for accomplishing task demands. As such, I would suggest not using network and resting-state almost as synonyms in the text.

We thank the reviewer for bringing this unclear phrasing to our attention. The term “network” refers to both the “network activity” from our model, which can represent either the Resting State network activity or one of the Task network activities. It specifically refers to the structured component of network activity that is present in both states (Figure 3). We have added these sentences to clarify and define its usage in the text and have also reviewed all 132 instances of the term “network” to ensure consistency throughout the manuscript. In addition, we define network in the introduction as:

The network activity is defined here as the component of the signal that can be explained by the activity of neighboring nodes. It can be constructed separately for Rest and Task datasets to capture the processes specific to each dataset.

Page 4: “we apply the following three criteria to ensure that they are biophysically plausible: 1) The loss function namely how much the ODE matches the derivative of the fMRI signal 2) The degree of the Taylor series coefficients Ξ_n are related to the SC network as measured through Tractography such as in BNM 3) the Hemodynamic Response function (HRF) of the system and comparing it to the canonical HRF response.” This sentence could be written more clearly. For example, how was the loss function chosen to be physiologically plausible?

We thank the reviewer and have clarified our writing according to this issue as follows:

To constrain the possible ODE solutions and ensure our model maximizes biophysical plausibility while accurately predicting dynamics, we apply the following three criteria:

1. Match the network timeseries dynamics exhibited in fMRI: The model should capture the network dynamics observed through functional connectivity (FC) by ensuring that the predicted timeseries closely matches the measured fMRI signal. Furthermore,

the residual signals should be uncorrelated with each other, indicating that the structured network processes have been effectively accounted for in the model.

2. Match the identified structural network with other methods of inferring the structural network: The network edges modeled using the Taylor series coefficients should align with the structural connectivity (SC) network derived from other methods, such as white matter tractography. This ensures that the model's dynamics are driven by realistic network processes, rather than by an arbitrary solution to the system's behavior.

3. Match the Hemodynamic Response Function (HRF): The impulse response of the model, its HRF, should match the canonical HRF observed in empirical measurements, ensuring that the hemodynamic response of the model aligns with biologically plausible expectations.

These criteria collectively guide the model towards biophysically plausible and accurate predictions.

Whenever a functional connectivity matrix is shown please provide some information regarding how the ROIs are sorted. Is it randomly? Is it by hemisphere? By network?

The FC matrices have the ROI sorted based on Cabral et al 2011. We have included the ROI order in the supplemental materials.

Table 2 Brain Regions by Hemisphere in the order that they appear in FC and SC matrices

No. Right	No. Left	Regions
1	84	Cerebellum
2	83	Thalamus
3	82	Caudate
4	81	Putamen
5	80	Pallidum
6	79	Hippocampus
7	78	Amgydala
8	77	Accumbens
9	76	Insula
10	75	Entorhinal
11	74	Posthippocampal
12	73	Temporal Pole
13	72	Frontal Pole
14	71	Fusiform
15	70	Transverse Temporal
16	69	Lateral Occipital
17	68	Superior Parietal
18	67	Inferior Temporal
19	66	Inferior Parietal
20	65	Supramarginal
21	64	Baudamus
22	63	Middle Temporal
23	62	Superior Temporal
24	61	Postcentral
25	60	Precentral
26	59	Caudal Middle Frontal
27	58	Pars Opercularis
28	57	Pars Triangularis
29	56	Rostral Middle Frontal
30	55	Pars Orbitalis
31	54	Lateral Orbitofrontal
32	53	Caudal Anterior Cingulate
33	52	Rostral Anterior Cingulate
34	51	Superior Frontal
35	50	Medial Orbitofrontal
36	49	Lingual
37	48	Precentral
38	47	Cuneus
39	46	Paracentral
40	45	Isthmus Cingulate
41	44	Precentral
42	43	Posterior Cingulate

Figure 4.B. Please add shaded regions depicting the uncertainty around the means reported as solid traces.

We have added the standard deviation for the HRF responses as well as highlighted time slices (determined to be the timepoints that are above 90 percent of the maximum correlation in the unseparated case) that are used to test significance in Figure 4C for the correlation analysis for Figure 4D.

“The above reasoning is consistent with how GLM models are currently run, in the sense that processes that are unique to the Task activate during the presentation of stimulus, and other Rest processes that always exist would not show up as statistically significant [40].” Many current task studies no longer contrast task and rest, but two different task conditions that differ in terms of the processes being targeted by the study. This is because rest is not simply everything except the stimuli response. Please amend accordingly. Also, reference [40], which originally introduced the AFNI software back in 1996, might not be the best reference to support this claim.

We thank the reviewer for bringing to attention the notion of two task conditions. We directly address the relationship between contrasting two different task conditions and ACM separation in supplemental Figure 20. It follows the detailed response to the reviewers comment section **“What did we learn about working memory, gambling, relational and emotional processes?”**.

In addition, we have updated the references to later Task review papers that are more reflective to current Task studies and modified the discussion accordingly:

Prior models of Task using Generalized Linear Models (GLM) have assumed that Task structure is replicable, whereas Rest processes are more randomly distributed (Cox, 1996; Monti et al., 2011). This assumption has allowed these models to statistically identify the regions where the Task occurs. This aligns with our approach of training separate models for Rest and Task, where during Task, Rest processes that are not associated with the Task appear to be unstructured processes. GLM models also utilize two Task conditions with varying difficulty levels to isolate primary cortical regions responsible for the Task, rather than all regions necessary for Task execution, such as the visual cortex or motor areas involved in button pressing (Monti et al., 2011). In our

variant of this experiment, by contrasting regions with higher correlation during 0-back vs. 2-back, we identify regions that match Task severity. These regions closely align with areas that show the greatest normalized increase to behavioral performance due to ACM separation (see Supplementary Figure 18). This suggests that ACM selectively enhances areas responsible for Task severity and difficulty rather than all regions involved in Task execution.

Figure 7D. Two ROIs seem to have very different HRFs, particularly they seem to be the only ones showing a post-undershoot. What are these and why are they the only ones showing this property?

The two Cerebellum ROIs exhibit a larger undershoot compared to the other ROIs. Unlike the other ROIs, they produce negative output to their neighbors, and when activated, they inhibit neighboring ROIs, which in turn suppresses their activity further, resulting in a more pronounced undershoot after the initial impulse. However, other ROIs also exhibit an undershoot, though it is less pronounced than the Cerebellar ROI. These are particularly for more negative lambda values which cause a larger undershoot, as shown in Figure 7C.

Prior work to consider:

(Thank you)

=====

Cole, M.W., Ito, T., Cocuzza, C., Sanchez-Romero, R., 2021. The Functional Relevance of Task-State Functional Connectivity. J. Neurosci. 41, 2684–2702. <https://doi.org/10.1523/jneurosci.1713-20.2021>

Ekman, M., Derrfuss, J., Tittgemeyer, M., Fiebach, C.J., 2012. Predicting errors from reconfiguration patterns in human brain networks. Proceedings of the National Academy of Sciences 109. <https://doi.org/10.1073/pnas.1207523109>

Gonzalez-Castillo, J., Kam, J.W.Y., Hoy, C.W., Bandettini, P.A., 2021. How to Interpret Resting-State fMRI: Ask Your Participants. J Neurosci 41, 1130–1141. <https://doi.org/10.1523/jneurosci.1786-20.2020>

He, B.J., 2013. Spontaneous and Task-Evoked Brain Activity Negatively Interact. J. Neurosci. 33, 4672–4682. <https://doi.org/10.1523/jneurosci.2922-12.2013>

Ito, T., Brincat, S.L., Siegel, M., Mill, R.D., He, B.J., Miller, E.K., Rotstein, H.G., Cole, M.W., 2020. Task-evoked activity quenches neural correlations and variability across cortical areas. *PLoS Comput. Biol.* 16, e1007983. <https://doi.org/10.1371/journal.pcbi.1007983>

Liu, Z., Wang, X., Bazan, A.C.S., Cao, J., 2023. Advances in Resting-State Functional MRI 87–105. <https://doi.org/10.1016/b978-0-323-91688-2.00015-1>

Lynch, L.K., Lu, K., Wen, H., Zhang, Y., Saykin, A.J., Liu, Z., 2018. Task-evoked functional connectivity does not explain functional connectivity differences between rest and task conditions. *Hum. Brain Mapp.* 39, 4939–4948. <https://doi.org/10.1002/hbm.24335>

Newbold, D.J., Laumann, T.O., Hoyt, C.R., Hampton, J.M., Montez, D.F., Raut, R.V., Ortega, M., Mitra, A., Nielsen, A.N., Miller, D.B., Adeyemo, B., Nguyen, A.L., Scheidter, K.M., Tanenbaum, A.B., Van, A.N., Marek, S., Schlaggar, B.L., Carter, A.R., Greene, D.J., Gordon, E.M., Raichle, M.E., Petersen, S.E., Snyder, A.Z., Dosenbach, N.U.F., 2020. Plasticity and Spontaneous Activity Pulses in Disused Human Brain Circuits. *Neuron* 107, 580-589.e6. <https://doi.org/10.1016/j.neuron.2020.05.007>

Pezzulo, G., Zorzi, M., Corbetta, M., 2021. The secret life of predictive brains: what's spontaneous activity for? *Trends Cogn Sci* 25, 730–743. <https://doi.org/10.1016/j.tics.2021.05.007>

Power, J.D., Plitt, M., Gotts, S.J., Kundu, P., Voon, V., Bandettini, P.A., Martin, A., 2018. Ridding fMRI data of motion-related influences: Removal of signals with distinct spatial and physical bases in multiecho data. *Proceedings of the National Academy of Sciences* 115, 201720985. <https://doi.org/10.1073/pnas.1720985115>

Sadaghiani, S., Poline, J.-B., Kleinschmidt, A., D'Esposito, M., 2015. Ongoing dynamics in large-scale functional connectivity predict perception. *Proceedings of the National Academy of Sciences* 112. <https://doi.org/10.1073/pnas.1420687112>

Smith, S.D., Nadeau, C., Sorokopud-Jones, M., Kornelsen, J., 2022. The Relationship Between Functional Connectivity and Interoceptive Sensibility. *Brain Connect.* 12, 417–431. <https://doi.org/10.1089/brain.2020.0777>

Thompson, G.J., Magnuson, M.E., Merritt, M.D., Schwarb, H., Pan, W.-J.J., McKinley, A., Tripp, L.D., Schumacher, E.H., Keilholz, S.D., 2013. Short-time windows of correlation between large-scale functional brain networks predict

vigilance intraindividually and interindividually. Human brain mapping 34, 3280–98. <https://doi.org/10.1002/hbm.22140>

Reviewer #2 (Remarks to the Author):

This study presents a novel approach to separating resting-state and task-specific fMRI signals using an ODE-based framework. The research introduces the Active Cortex Model and applies SINDy to identifying network interactions. While the paper has notable strengths, including its innovative model and application to Human Connectome Project data, several critical issues need to be addressed for publication:

We thank the reviewer for their thoughtful evaluation of our manuscript and their consideration for its potential future publication. Below, we provide a point-by-point response addressing the critical issues raised.

1) The relationship between task and rest is described as linear, yet no robust justification is provided. It would be beneficial to explore non-linear alternatives, or at least discuss why a linear approach was chosen and its limitations.

We agree with the reviewer that the relationship between Rest and Task could be nonlinear, which is indeed an intriguing question for future research. However, a general optimization was not performed to determine their relationship unlike the other parameters in our model, due to the unknown Brain-to-Behavior function. For instance, below we consider a simplified case where the optimization problem focuses solely on Eq. 6 to directly determine c_1 and c_2 :

$$RT = G\left(\int \dot{X} - Q(\Xi_{Rest} * \theta(X)) + P(\Xi_{Task} * \theta(X))\right)$$

Here, instead of the simple c_1 and c_2 , we consider slightly more general functions P and Q . In the final step of comparing the left and right sides, we only derive a proportion, as the exact function relating brain activity to behavioral data remains unknown. In our approach, this function was approximated using either an Elastic Net or using correlation, making it challenging to simultaneously solve for the unknown P and Q functions along with the unknown Brain-to-Behavior function G . Consequently, we prioritized exploring the linear relationship by fixing the Brain-to-Behavior function as a simple correlation and systematically varying the c_1 and c_2 values. Moreover, even at this simple linear level, we were able to demonstrate that Task processes can be represented as a linear subset of Rest, enabling the isolation of Task-specific activity within the

measured whole-brain activity. Thus, while this approach is a limitation, it does not alter the main results of our study.

We have added this to the limitation section of the Discussion Section under limitations:

We have only tested the linear separability between the identified Rest and Task network models in this study. For future studies, non-linear separation schemes should also be considered and might provide a substantial boost in accuracy. However, this is difficult in practice; for example, consider a simpler special case an optimization problem of equation Eq 6 rewritten as follows:

$$RT = G\left(\int \dot{X} - Q(\Xi_{Rest} * \theta(X)) + P(\Xi_{Task} * \theta(X))\right)$$

Solving this optimization is difficult, as the comparison to the RT coefficients needs another function G , which we implement as the Brain to Behavior Function. This function is approximated in two manners 1) using correlation and 2) using an Elastic Net. Since the relationship between Rest and Task are unknown as well as the Brain to Behavior function, it makes it difficult to solve for both the relationship as well as the Brain to Behavior functions directly from the data. We have limited our analysis instead of testing a linear relationship between the Rest and Task networks by simply using coefficients for P and Q and sweeping them across a range. In future studies, a more complex technique may be able to address this optimization problem directly and test the nonlinear relationship between Rest and Task.

2) The authors use SINDy to identify network dynamics from fMRI data but do not sufficiently describe how noise and artifacts from fMRI data were handled. fMRI signals can be influenced by a range of physiological factors; a more detailed discussion of preprocessing steps (e.g., denoising methods) is required.

We thank the reviewer for bringing this to our attention, and have elaborated below on how noise and artifacts were handled. Our main strategy to deal with these signals, is by utilizing SINDy's package of built-in functions to denoise the data in addition to HCP denoising preprocessing steps (the minimally processed pipeline). The SINDy denoising functions are tailored to the SINDy algorithm, as simple transformations such as ICA, PCA or even global signal regression would change the equations that are determined from the SINDy algorithm. SINDy has built-in functions to smooth the derivative to reduce noise and reject large outliers via trimming and excluding them from the regression. Moreover, SINDy is inherently robust as it uses regression to fit the coefficients. Only noise and motion that is consistent across all HCP scans in reference to the timeseries would affect the coefficients which would be exceedingly rare. This is one of the main strengths of using regression instead of other approaches. In fact, the trimming and smoothing of the derivative had little effect on the resulting coefficients, so we used faster methods such as computing the spectral derivative in order to solve for our ODE (Supplementary Figure 11).

We have added a new Methods section 3.2.6 called “Handling Artifacts, Noise, and Physiological Processes”:

In order to handle noise, artifacts and non-neuronal physiological processes, we introduced a few different steps to minimize their effects on the models identified. Since most preprocessing steps affect the dynamics, changes in this processing are transferred directly to the identified coefficients. Instead to address these nuisance signals, we utilized the in-built methods that exist in the SINDy package that have been specifically developed for these issues. The SINDy package includes a derivative option 'Smoothed Finite Difference' to handle noisy data as well as a trimming function that ignores large derivatives due to artifacts. We found however, that in practice these made little difference (see Supplementary Figure 11) and with sufficiently large amounts of fMRI data, the ODE discovered is rather stable and consistent regardless of which subset is used to train the model. Therefore, we went for the fastest option which was 'Spectral Derivative'.

Since the dataset used to construct the ODE was a large subset of the entire HCP cohort, for artifacts and noise to translate into meaningful changes in the coefficients, they would have to consistently occur at the same time with reference to the current brain activity. Therefore, our method is robust to noise, which is generally the case for such regression based methods that yield stable, consistent and convergent solutions.

3) While the paper reports a 9% increase in explanatory power across subtasks, it is unclear how this improvement compares to baseline models or other approaches in the literature. Adding comparisons with more traditional methods or competing models (e.g., DCM) would clarify the model's advantages and limitations.

We agree with the reviewer that adding comparisons would strengthen and contextualize our approach. However, as this field is still relatively new, no existing methods can fully replicate our results for direct comparison, although certain aspects of our findings can be evaluated against existing approaches.

Our method was chosen as it makes no assumptions, while predicting the major components of an fMRI model: the timeseries, the HRF response, the structural connectivity, the functional connectivity, and interpreting the signal to behavior variables on a single trial resolution while being able to make claims on the general structure of Rest and Task, the Active Cortex Model. While these approaches could potentially be extended to predict additional elements of an fMRI model, their implementation would require further development and warrant a separate publication.

The only DCM method with which we can numerically compare our main results (Figures 4-6) has a correlation accuracy of 0.46 for the 2-back WM task (Cai et al., 2021). In contrast, our correlations for the 2-back WM task exceed 0.7 (Figure 5). It is worth noting that the y-axis in our

paper represents r-squared values, so the correlation is derived by taking the square root of these values. A key limitation of the DCM model is its difficulty in modeling a large number of ROIs; even recent studies, such as Cai et al., 2021, are restricted to only 11 ROIs, whereas our method successfully models over 84 ROIs. Regression based DCM does not have this problem, but that is equivalent to our method using a single linear component to model the edge functions, which we show in Figure 7 to be worse in terms of fitting to the data than those models that include higher polynomial terms.

While it is difficult to assess our 9 percent increase in terms of previous or even perhaps future publication with better models, our method stands out as a novel approach that allows us to prove ACM. We would like to emphasize this more as the significance of the work rather than a fixed increase in explainability.

In our revision we have added clarity in our text in addition to the table, rewritten the Discussion Section 4.2, to help contextualize our work in relation to DCM as well as other models

Our approach to model the fMRI equation is most similar to the early approaches used to describe ODEs in fMRI such as DCM (Friston 2000, Friston et al., 2003). In fact, our equations are very similar to those first proposed by Friston et al., except rather than using a linear model, the SINDy framework solves for a general Taylor Expansion to represent the relationship between nodes (Friston et al., 2003). DCM impressively solves for both inputs and the ODE structure, while we relax the conditions relying on the law of large numbers and regression to correctly deduce the ODE framework and interpret the non-structured activity to represent stimulus activity (Friston et al., 2014). Another difference between the two approaches is that DCM assumes the deconvolution problem of inverting the signal to solve for neural relations, which does not allow for direct predictions on fMRI timeseries data. However, one of the main bottlenecks of DCM is that the method does not scale well and thus has been hampered in characterizing the dynamics of large sets of ROIs. A recent publication that analyzed HCP WM single trials only modeled 11 ROIs (Cai et al., 2021). SINDy allows us to solve for the same system but modeling larger amounts of ROIs because it uses regression rather than Bayesian Inference which can scale appropriately. It can therefore also solve for the complex HRF function and thus make predictions as well as manipulate fMRI on the measured timeseries resolution. Regression based DCM overcomes many of the challenges of traditional DCM and has been able to achieve whole brain ROI connectivity analysis using a linear model (Frassle et al., 2017, Frassle et al., 2021). Our methodology, which also applies regression, is a generalization of this method and uses more terms in the Taylor series expansion to account for nonlinear relations. The SINDy spectral package solves for the coefficients in the frequency spectrum similar to those proposed in Frassel et al., 2017. However, since they utilize deconvolution to estimate the neural equation rather than the BOLD equation, their approaches are limited to observing differences in effective connectivity during Task (Frassle et al., 2021) rather than applying the model to directly modify the signal.

Meanwhile, the generative modeling community adapted the DCM equations by integrating estimates of *SC* from white matter tractography, partially to avoid directly solving intricate structural relationships, resulting in a family of models that utilize fixed differential equations, specifically BNMs (Jirsa et al., 2009, Breakspear 2017). These network models assumed a network *SC* derived from white matter connectivity and were popular in generating signals that have *FC* similar to the resting state *FC* by integrating candidate network equations with noise over a long period of time. Many of these approaches also use a Balloon Windkessel similar to DCM to transform the simulated neural data to fMRI data (Cabral et al., 2012, Cabral et al., 2011). However, BNMs have the limitation of simulating 1) their model with noise 2) in another space that could be adjacent to measured data. This makes timeseries predictions rather hard (Kashyap et al., 2019) and in fact has been used to simulate summary metrics such as *FC* and modeling per trial Task analysis difficult. They have, however, uncovered properties of Task trials (Schirner et al., 2023) while others have derived simpler activity flow models from the BNM underlying equations to model Rest Task switching as activations of sub-networks of Resting State functional networks (Cole et al., 2021, Ito et al., 2020). Another popular approach uses NFE that utilize a wave equation to describe the dynamics, and uses the physical cortical sheet in order to solve for the Dynamics. NFE solves for eigenmodes based on the cortical structure and simulates Resting State dynamics as a superposition of these modes (Robinson 2016). These can be done in higher resolution than network models in voxel resolution (Pang et al., 2023). We have provided the eigenvalue decomposition of our Jacobian, namely the first-order terms of the *EC*, shown in the Supplementary Figure 14. They exhibit similar structures observed in NFE, with global eigenmodes with large eigenvalues, and smaller non bihemispheric modes with smaller values. A recent paper also showed that the NFM simulations are similar to those of BNM, although the performance of NFM is slightly better than that of BNM (Pang et al., 2023). Therefore, we interpret that BNM and NFE are related, as it has been demonstrated that *SC* is predominantly organized according to the principle that geodesically closer areas are more strongly connected, leading to similar network models where adjacent regions exhibit strong connectivity (Margulies et al., 2016, Ponce-Alvarez et al., 2023). Apart from the network structure, BNM and NFE are roughly equivalent in terms of their scope. Both are able to show that the simulated signal has dynamic properties, i.e *FC*, multiple states, which are similar to those extracted from Resting State fMRI, but as they usually simulate in adjacent space which is not representative of measurements, they fail to provide the means to predict/ manipulate the fMRI timeseries directly.

Deep learning approaches have also been used to train dynamical systems to represent the underlying ODE (Durstewitz et al., 2017, Koppe et al., 2019, Kashyap et al., 2023, Champion et al., 2019). Here, we specifically refer to a subset of deep learning approaches designed for ODE identification in fMRI, rather than all possible architectures that have been used or could be used to represent fMRI data. A specific architecture, piecewise linear recurrent neural networks, has been theoretically shown to solve any general ODE system from observations (Durstewitz et al., 2017, Koppe et al., 2019). After applying the algorithm on fMRI datasets they show they are able to predict the evolution of measured fMRI data (Koppe et al., 2019). However, most gradient

learning-based approaches have issues starting from different initial conditions and converging to a single solution observations which the authors themselves acknowledge (Durstewitz et al., 2017, Koppe et al., 2019). Deep learning techniques have no convergent limits unlike the simpler traditional regression based techniques that this study employs. Moreover, the dynamics are solved in a latent space, and thus is not only different training on different datasets such as Rest and Task, but also on every instantiation on the same dataset (Kashyap et al., 2019, Durstewitz et al., 2017). This makes testing the prediction of the difference of two models employed in the ACM impossible and the interpretation of the coefficients in relation to *SC* difficult, as different latent spaces lead to different connectivity networks. They have thus used the model to infer system statistics such as the stability of the system in diseases and do not directly manipulate the timeseries to infer relationships to behavioral variables (Koppe et al., 2019). We anticipate, however, that future deep learning models will advance system identification algorithms, as the field continues to develop rapidly, and those future models will massively outperform regression based approaches.

4) The validation of the model is restricted to the data from the Human Connectome Project. For broader generalization, the authors should test the model on independent datasets or, at the very least, discuss the limitations of only using HCP data for training and testing.

The reviewer is correct that showing generalization of the algorithm and results would strengthen the results of this study as well as justify the model more broadly. However, in this work as we are chiefly interested in relating Rest and Task, there are few datasets that have such a rich amount of Rest and Task studies, as well as DTI from a large number of healthy individuals as this HCP study. A minimum of 100 hours of fMRI are needed to train either the Rest or Task network models although using priors in future methods could be used to compensate for smaller datasets (see Supplementary Figure 4). A dataset that would fit these criteria other than HCP would probably be the UK biobank and we welcome the field to independently verify our results. We have thus qualified our claims about only using HCP dataset as well as cited the UK biobank as a potential for future research:

A large amount of fMRI data is needed to train the models, which is a limitation (see Supplementary Figure 11). In HCP terms about a hundred subjects are needed with all four resting state scans to train a single model which is much larger than other approaches. It has limited the scope of this analysis to the HCP dataset or similar large datasets, because HCP contains enough long recordings that can be used to train the Task and Rest network models (Miller et al., 2016). Limiting the analysis to HCP datasets tempers our results to healthy young adults who were scanned with the HCP protocol. Moreover, the analysis assumes that the development of the cortex is almost stationary and does not take into account any effects of aging, as well as known gender differences that affect cortical organizations and dynamics (Ritchie et al., 2018, Cole et al., 2020). Moreover, the data represent awake data and do not take into account any changes due to circadian cycles as well as longer cycles that have been shown to affect the dynamics of the system (Karoly et al., 2021). The analysis could be reproduced in the future in similar large imaging studies such

as the UK Biobank that have task fMRI data (Miller et al., 2016). Using priors for the dynamical system can offset the data needed for large amounts of data in future approaches and allow for shorter segments to modify the group dynamical system.

5) One major concern is the interpretability of the results. While the authors claim to improve behavioral predictions using the model, there is limited discussion of the practical utility of these predictions. It is not evident how much this 9% increase in R^2 translates into meaningful behavioral insights. Additionally, more explanation is needed on how this model could be applied to clinical or cognitive neuroscience research.

We agree with the reviewer's concerns about the interpretability of the results. Most of the focus of this paper is to establish that ACM is a viable model for understanding the relationship between Rest and Task and we see that as a meaningful behavioral insight. Separating Rest from Task was one of the first hurdles in understanding neural data and a long-standing unresolved problem in fMRI. For that we have provided these sets of evidence: 1) the Task isolated timeseries is more related to measured behavioral variables such as reaction time or the prediction of missing data from the separated timeseries 2) the separation of Task isolated FCs visualized via UMAP. Since both the timeseries and FCs are extensively used in the analysis of Task specific data in both clinical and cognitive neuroscience (Barch et al., 2013, Greene et al., 2018), we expect that they would change the outcome of any such studies. But quantifying that improvement is not limited to this specific method, as a similar approach in generating ODE from fMRI data in the future will most likely outperform this method in separating and modeling the fMRI dynamics. Thus, the focus is on establishing that ACM is a viable method and that this kind of separation is even possible in the first place, and it answers the fundamental question between Rest and Task. This can be used in future research to account for and remove unrelated Task processes to aid interpretation.

For the revision, we changed the top half of Figure 5 and added Supplementary Figure 20 in order to show that the changes in ROIs that account for increase in explainability of behavioral responses, correspond to brain regions associated with the difficulty of the Task, i.e prefrontal areas rather than regions such as the visual cortex that are necessary for Task completion. We hope this clarifies where these changes occur and moreover elaborate which regions it would be impacted for cognitive studies namely, the areas that are specifically related to the severity of the Task.

In addition, we have now included a section relating to future work addressing how this could affect future cognitive and clinical neuroscience studies.

This study shows that by utilizing ACM we can improve the isolation of Task both from the measured timeseries as well as increased separation in the inferred network FC space. Although timeseries and / or FCs are the main input in imaging-based cognitive and clinical studies, it is not clear how they will directly affect each individual study (Greene et al., 2018, Fox et al., 2010).

Future research has to be conducted on how applicable they are, but since they are able to change the primary data analyzed in these studies, we speculate that they will show an improvement on behavioral variables and Task interpretability in most studies. Moreover, we believe that this approach can be improved upon, for example using more ROIs, by constructing priors to analyze smaller datasets, by using gradient-based deep learning tools to determine more complex equations. However, in all of these contexts, we believe that the most significant impact of this study is the paradigm shift in applying modeling techniques to directly model components of the whole brain activity and manipulating the timeseries via ACM to gain more information from the Task studies. This would aid in translating the extensive work and the progress that the modeling community has achieved in representing fMRI dynamics to the wider neuroscience community.

Changes in Figure 5 and Supplementary Figure 20:

Fig. 5 A Left) Multiple brain regions are associated with each of the Task experiments that have moderate correlation with the RT. The Max Corr refers to the maximum correlation across time for each individual ROI across all trials from all individuals. The areas with large correlation are similar to previously published Task Activation Maps (See Supplemental Figure 15) A Right) The normalized changes to the correlation in these areas align with a portion of the original ROIs associated with the Task. For WM, we have examined the relation between the severity of the Task and the areas that change in correlation in Supplemental Figure 20 B) Results of utilizing an Elastic Net to aggregate

Task Activation Maps Relationships

Fig. 20 Top Left: The activation Map from ACM based on the max correlation of a ROI across time for 0 back working memory. Top Right: The activation Map from the unseparated data based on the max correlation of a ROI across time for 0 back working memory. Bottom Left: The normalized change in correlation between the ACM (separated) and the unseparated cases showing a subset of regions that were originally identified in the Task increasing in correlation. Bottom Right: The difference between the activation maps based on correlation to RT for the zeroback and twoback. The zeroback trials are more correlated with the RT, and the difference between them removes associations between regions that are necessary for the Task such as the visual cortex but do not reflect the Task severity. The change in separated vs unseparated cases match the difference between tasks suggesting that the increases in correlation are in regions that are specific to the particular difficulty of the Task, not just regions that activate for Task completion such as the visual cortex to see the input or the motor cortex for button pushing. Significance Test: Norm Δ RT Corr to 0 vs 2back = 0.79 R^2 Norm Δ RT Corr to ACM RT Corr = 0.75, Norm Δ RT Corr to Unseparated RT Corr = 0.76. P-value difference between 0.79 > 0.75 is 0.002 and 0.79 > 0.76 is 0.003.

6) The authors claimed in section 3.1.1 that there are 447 subjects with resting-state fMRI, and then 1032 subjects for working memory, 1013 for relational, 1017 for emotion, and 1032 for gambling. Does this mean the comparisons between resting-state fMRI and task-related fMRI are using different subjects? Some clarification is needed here.

We thank the reviewer for bringing up this unclear point in our methodology. We constructed different models for Rest and Task with different subjects. From each of these datasets mentioned, we generate a single equation to represent each of the group Task models and group Rest Model. We then test to see if this group Task model is a subset of the group Rest Model.

In terms of not mixing training and testing subjects, our most stringent analysis was performed in Supplementary Figure 18, we trained the Rest and Working Memory Task models on subsets of the population containing twins and then tested it on unrelated non twin population. We showed that the results are similar to repeating the analysis without the stringent restrictions, suggesting that the algorithm generalizes and converges to a single set of ODE equations for the particular task across the HCP population. For the smaller Tasks datasets that were roughly half or less than half as long as the Working Memory this generalization was essential, as we needed to utilize all of the data to converge to a single set of ODE equations for the particular task.

We thank the reviewer for bringing this to our attention and have clarified in the methods section 3.1.1 as follows:

From each of these datasets mentioned, we generate a single equation to represent each of the group Task models and group Rest Model. We then test to see if this group Task model is a subset of the group Rest Model.

In order to make sure test and training datasets were split appropriately while also accounting for the constraints of having enough data to train the network models, we repeated the analysis using two different approaches: 1) a stringent task/ test split by individual while also accounting such that twins are not in the test/training group for constructing a Working Memory model 2) a more relaxed approach for the other Tasks where all available data was utilized since the recordings were shorter. We show that stringent and relaxed approaches have similar results for WM (Supplemental Figure 18), which is expected, as the backbone of the system identification algorithm is regression, which is known to be stable, to be generalizable, and has hard limits of convergence provided that there exists sufficient amounts of data.

7) Some literature on applying ODE to brain networks or brain graphs should be cited and discussed.

We thank the reviewer for bringing this up as and have included more citations on previous models of applying ODEs to brain networks. These include the Ito and Cole et al papers that utilize functional connectivity in conjunction with ODEs to model the Task and Rest states as well as those from Durstewitz that use neural networks to identify ODEs. We also included an additional 30 citations in our revision although not all of them are on applying ODE to brain networks

We are not sure if this addresses the reviewer concern of citing ‘brain graphs’ as we are unsure what this is exactly referring to but would be interested in citing those as well.

While this study addresses an important question in neuroimaging and introduces innovative modeling techniques, significant revisions are needed before the paper is ready for publication. The clarity of the methods and rationale, as well as additional validation and broader discussion of the model's implications, will greatly enhance the manuscript's contribution to the field.

We thank the reviewer for their thorough review of our manuscript, and have attempted to clarify the methods, rationale and implications of the model's implications. For additional validation of the ACM, we have included in our revision an analysis of classifying trajectories with and without ACM (Supplemental Figure 19C). We also included another Figure (Supplemental Figure 21) on classifying two additional behavioral variables missing trials and accuracy based on ACM separation. Thus, we have demonstrated across a total of 14 subtasks and 5 behavioral variables that ACM allows for Task isolation. We have also shown in the revision that the changes in RT represent areas that are related to Task severity (Figure 5 and Supplemental Figure 20). In addition, we have rewritten the introduction and discussion sections relating to other work as well as expanded on the future direction and limitations section.

Thus, we hope to focus our scope of this manuscript to justify the ACM approach and clarify the limitations that can be addressed using future research.

Reviewer #2 (Remarks on code availability):

It would be great if the authors could provide sample input data and let the users test-run the code.

HCP does not allow us to share modified portions of its data. However, we have included the scripts that were used to generate the data from HCP.

References:

Cai, W., Ryali, S., Pasumarthy, R., Talasila, V., Menon, V.: Dynamic causal brain circuits during working memory and their functional controllability. *Nature Communications* 12(1), 3314 (2021) <https://doi.org/10.1038/s41467-021-23509-x>. Number: 1 Publisher: Nature Publishing Group. Accessed 2023-06-20

Cole, M.W., Ito, T., Cocuzza, C., Sanchez-Romero, R., 2021. The Functional Relevance of Task-State Functional Connectivity. *J. Neurosci.* 41, 2684–2702. <https://doi.org/10.1523/jneurosci.1713-20.2021>

Ito, T., Brincat, S.L., Siegel, M., Mill, R.D., He, B.J., Miller, E.K., Rotstein, H.G., Cole, M.W., 2020. Task-evoked activity quenches neural correlations and variability across cortical areas. *PLoS Comput. Biol.* 16, e1007983. <https://doi.org/10.1371/journal.pcbi.1007983>

Liu, Z., Wang, X., Bazan, A.C.S., Cao, J., 2023. Advances in Resting-State Functional MRI 87–105. <https://doi.org/10.1016/b978-0-323-91688-2.00015-1>

Barch, D.M., Burgess, G.C., Harms, M.P., Petersen, S.E., Schlaggar, B.L., Corbetta, M., Glasser, M.F., Curtiss, S., Dixit, S., Feldt, C., Nolan, D., Bryant, E., Hartley, T., Footer, O., Bjork, J.M., Poldrack, R., Smith, S., Johansen-Berg, H., Snyder, A.Z., Van Essen, D.C.: Function in the human connectome: Task-fMRI and individual differences in behavior. *NeuroImage* 80, 169–189 (2013) <https://doi.org/10.1016/j.neuroimage.2013.05.033> . Accessed 2023-06-20

Greene, A.S., Gao, S., Scheinost, D., Constable, R.T.: Task-induced brain state manipulation improves prediction of individual traits. *Nature Communications* 9(1), 2807 (2018) <https://doi.org/10.1038/s41467-018-04920-3> . Publisher: Nature Publishing Group. Accessed 2024-12-24

Durstewitz, D.: A state space approach for piecewise-linear recurrent neural networks for identifying computational dynamics from neural measurements. *PLOS Computational Biology* 13(6), 1005542 (2017) <https://doi.org/10.1371/journal.pcbi.1005542> . Publisher: Public Library of Science. Accessed 2024-12-04

Koppe, G., Toutounji, H., Kirsch, P., Lis, S., Durstewitz, D.: Identifying nonlinear dynamical systems via generative recurrent neural networks with applications to fMRI. *PLOS Computational Biology* 15(8), 1007263 (2019) <https://doi.org/10.1371/journal.pcbi.1007263> . Publisher: Public Library of Science.

Reviewer #3 (Remarks to the Author):

This paper proposes a network Ordinary Differential Equation (ODE) to analyze fMRI data from both resting and task conditions. The model incorporates higher-order terms of the signal to capture its dynamics. By subtracting simulated resting signals from the task signals, the R2 is able to explain 9% of the observation variance compared to the unseparated data.

Among all the results, Fig. 5 (Brain to Behavior Function) directly supports the claims of the paper. By separating the resting signal, the model achieves a higher R2 score in the Working Memory Task, Emotion Fear, Relational Texture, and Gambling Reward tasks. Additionally, the UMAP shown in Fig. 2 illustrates that, in the separated model, the functional connectivity of distinct tasks is more clearly differentiated.

We thank the reviewer for their thorough review of our manuscript and their insightful comments into our work. We would like to emphasize in our decoding process that it contains two separately trained models (one for Rest and one for Task) and only the combination of them both is able to increase the variance and the Rest and Task baseline models in isolation are not able to decode the data.

To summarize our revisions:

We have included for additional validation of the ACM, an analysis of classifying trajectories during each trial and shown improvement only using ACM (Supplemental Figure 19C). We also included another Figure (Supplemental Figure 21) on classifying two additional behavioral variables missing trials and accuracy based on ACM separation. Thus, we have demonstrated across a total of 14 subtasks and 5 behavioral variables that ACM allows for Task isolation. We have also shown in the revision that the changes in RT represent areas that are related to Task severity to clarify where and how these changes might localize in the cortex (Figure 5 and Supplemental Figure 20). In addition, we have rewritten the discussion sections relating to other work as well as expanded on the future direction and limitations section.

Thus, we hope to focus our scope of this manuscript to things that are shown by this approach and clarify the limitations that can be addressed using future research.

In addition, we hope our comments below clarify and address the reviewers specific concerns about our manuscript in our point by point response:

Below are some suggestions for the manuscript:

1. In the methodology, the paper fits a high-order polynomial plus Gaussian noise to the derivative of the signal. It requires further explanation of why this method was chosen over others. For instance, why not use a neural network with a comparable number of unknown parameters to replace the estimated function $E(X_{b \neq a})$? It would be beneficial for the author to provide more evidence that the chosen method is superior to existing alternatives. A numerical experiment comparing the pros and cons of different methods would be helpful.

The reviewer is correct in suggesting that there exist alternative system identification tools could be used to solve for an ODE to fit the neural data. However, as this field is still relatively new, no existing methods can fully replicate our results for direct comparison, although certain aspects of our findings can be evaluated against existing approaches.

Our method was chosen as it makes no assumptions, while predicting the major components of an fMRI model: the timeseries, the HRF response, the structural connectivity, the functional connectivity, and interpreting the signal to behavior variables on a single trial resolution while

being able to make claims on the general structure of Rest and Task, the Active Cortex Model. While other approaches could potentially be extended to predict additional elements of an fMRI model, their implementation would require further development and warrant a separate publication.

The only method with which we can numerically compare our main results (Figures 4-6) is the DCM model, which demonstrates a correlation accuracy of 0.46 for the 2-back WM task (Cai et al., 2021). In contrast, our correlations for the 2-back WM task exceed 0.7 (Figure 5). It is worth noting that the y-axis in our paper represents r-squared values, so the correlation is derived by taking the square root of these values. A key limitation of the DCM model is its difficulty in modeling a large number of ROIs; even recent studies, such as Cai et al., 2021, are restricted to only 11 ROIs, whereas our method successfully models over 84 ROIs.

The reviewer is correct in suggesting that deep learning algorithms can be employed to infer the underlying ODE. In Champion et al. 2019, the original authors of SINDy introduced a deep implementation of SINDy using an autoencoder to derive an ODE. We further extended this approach specifically to fMRI data in Kashyap et al. 2021. In parallel, the Durstewitz group has published several studies that infer ODEs from fMRI data using a distinctive architecture known as piecewise linear RNNs (Durstewitz et al., 2018; Koppe et al., 2019). However, these approaches, including our own, encounter challenges with the convergence of their coefficients. This limitation is also mentioned by Durstewitz and their collaborators. This issue is notably absent in regression-based methods, which benefit from strict upper bounds on variance of the coefficients for a given amount of data (See Supplemental Figure 11). Moreover, all these deep learning approaches model dynamics in a latent space, which varies with each instantiation of the trained deep network. While this can be advantageous for reducing noise, it poses significant challenges for interpretation. Specifically, comparing two trained models for Rest and Task conditions becomes difficult, as each resides in its own unique latent space. This makes testing hypotheses like the Active Cortex Model exceedingly challenging as it requires the subtraction of the two models. Therefore, the Durstewitz group has primarily utilized their approach to assess the overall stability of the identified ODE. They achieve this by calculating the spread of eigenvalues across multiple instantiations of the trained models rather than interpreting timeseries trajectories. Furthermore, while many of these methods require hours or even days to computationally train, the regression approach used in this study completes in minutes and provides consistent, reproducible solutions. While we believe that in the future that an appropriate deep learning algorithm can be used to identify an ODE for the fMRI data, there does not seem to be an established method that we have found published that accomplishes this task and would require further development and warrant a separate publication.

In our revision we have added context for the alternative deep learning methods in the discussion section:

Deep learning approaches have also been used to train dynamical systems to represent the underlying ODE (Durstewitz et al., 2017, Koppe et al., 2019, Kashyap et al., 2023, Champion et al., 2019). Here, we specifically refer to a subset of deep learning approaches designed for ODE identification in fMRI, rather than all possible architectures that have been used or could be used to represent fMRI data. A specific architecture, piecewise linear recurrent neural networks, has been theoretically shown to solve any general ODE system from observations (Durstewitz et al., 2017, Koppe et al., 2019). After applying the algorithm on fMRI datasets they show they are able to predict the evolution of measured fMRI data (Koppe et al., 2019). However, most gradient learning-based approaches have issues starting from different initial conditions and converging to a single solution observations which the authors themselves acknowledge (Durstewitz et al., 2017, Koppe et al., 2019). Deep learning techniques have no convergent limits unlike the simpler traditional regression based techniques that this study employs. Moreover, the dynamics are solved in a latent space, and thus is not only different training on different datasets such as Rest and Task, but also on every instantiation on the same dataset (Kashyap et al., 2019, Durstewitz et al., 2017). This makes testing the prediction of the difference of two models employed in the ACM impossible and the interpretation of the coefficients in relation to *SC* difficult, as different latent spaces lead to different connectivity networks. They have thus used the model to infer system statistics such as the stability of the system in diseases and do not directly manipulate the timeseries to infer relationships to behavioral variables (Koppe et al., 2019). We anticipate, however, that future deep learning models will advance system identification algorithms, as the field continues to develop rapidly, and those future models will massively outperform regression based approaches.

As well as for non deep learning methods in the Discussion Section 4.2 paragraph 2:

Our approach to model the fMRI equation is most similar to the early approaches used to describe ODEs in fMRI such as DCM (Friston 2000, Friston et al., 2003). In fact, our equations are very similar to those first proposed by Friston et al., except rather than using a linear model, the SINDy framework solves for a general Taylor Expansion to represent the relationship between nodes (Friston et al., 2003). DCM impressively solves for both inputs and the ODE structure, while we relax the conditions relying on the law of large numbers and regression to correctly deduce the ODE framework and interpret the non-structured activity to represent stimulus activity (Friston et al., 2014). Another difference between the two approaches is that DCM assumes the deconvolution problem of inverting the signal to solve for neural relations, which does not allow for direct predictions on fMRI timeseries data. However, one of the main bottlenecks of DCM is that the method does not scale well and thus has been hampered in characterizing the dynamics of large sets of ROIs. A recent publication that analyzed HCP WM single trials only modeled 11 ROIS (Cai et al., 2021). SINDy allows us to solve for the same system but modeling larger amounts of ROIs because it uses regression rather than Bayesian Inference which can scale appropriately. It can therefore also solve for the complex HRF function and thus make predictions as well as manipulate fMRI on the measured timeseries resolution. Regression based DCM overcomes many of the challenges of traditional DCM and has been able to achieve whole brain ROI connectivity analysis

using a linear model (Frassle et al., 2017 , Frassle et al., 2021). Our methodology, which also applies regression, is a generalization of this method and uses more terms in the Taylor series expansion to account for nonlinear relations. The SINDy spectral package solves for the coefficients in the frequency spectrum similar to those proposed in Frassel et al., 2017. However, since they utilize deconvolution to estimate the neural equation rather than the BOLD equation, their approaches are limited to observing differences in effective connectivity during Task (Frassle et al., 2021) rather than applying the model to directly modify the signal.

Meanwhile, the generative modeling community adapted the DCM equations by integrating estimates of SC from white matter tractography, partially to avoid directly solving intricate structural relationships, resulting in a family of models that utilize fixed differential equations, specifically BNMs (Jirsa et al., 2009, Breakspear 2017). These network models assumed a network SC derived from white matter connectivity and were popular in generating signals that have FC similar to the resting state FC by integrating candidate network equations with noise over a long period of time. Many of these approaches also use a Balloon Windkessel similar to DCM to transform the simulated neural data to fMRI data (Cabral et al., 2012, Cabral et al., 2011). However, BNMs have the limitation of simulating 1) their model with noise 2) in another space that could be adjacent to measured data. This makes timeseries predictions rather hard (Kashyap et al., 2019) and in fact has been used to simulate summary metrics such as FC and modeling per trial Task analysis difficult. They have, however, uncovered properties of Task trials (Schirner et al., 2023) while others have derived simpler activity flow models from the BNM underlying equations to model Rest Task switching as activations of sub-networks of Resting State functional networks (Cole et al., 2021, Ito et al., 2020). Another popular approach uses NFE that utilize a wave equation to describe the dynamics, and uses the physical cortical sheet in order to solve for the Dynamics. NFE solves for eigenmodes based on the cortical structure and simulates Resting State dynamics as a superposition of these modes (Robinson 2016). These can be done in higher resolution than network models in voxel resolution (Pang et al., 2023). We have provided the eigenvalue decomposition of our Jacobian, namely the first-order terms of the EC , shown in the Supplementary Figure 14. They exhibit similar structures observed in NFE, with global eigenmodes with large eigenvalues, and smaller non bihemispheric modes with smaller values. A recent paper also showed that the NFM simulations are similar to those of BNM, although the performance of NFM is slightly better than that of BNM (Pang et al., 2023). Therefore, we interpret that BNM and NFE are related, as it has been demonstrated that SC is predominantly organized according to the principle that geodesically closer areas are more strongly connected, leading to similar network models where adjacent regions exhibit strong connectivity (Margulies et al., 2016, Ponce-Alvarez et al., 2023). Apart from the network structure, BNM and NFE are roughly equivalent in terms of their scope. Both are able to show that the simulated signal has dynamic properties, i.e FC , multiple states, which are similar to those extracted from Resting State fMRI, but as they usually simulate in adjacent space which is not representative of measurements, they fail to provide the means to predict/ manipulate the fMRI timeseries directly.

2. Still in the methodology, when separating the resting signal from the task signal, the author tests different combinations of coefficients c1 and c2 in Eq. 6. It would be more formal if the problem were framed as an optimization problem.

The reviewer brings up an interesting point as indeed Eq 6 can be reframed as an optimization problem. In general, we can rewrite Eq 6 as the following optimization problem:

$$RT = G\left(\int \dot{X} - Q(\Xi_{Rest} * \theta(X)) + P(\Xi_{Task} * \theta(X))\right)$$

This approach not only solves for c1 and c2 but extends to any independent relationship between Rest and Task, represented by functions P and Q . However, the main challenge lies in comparing the resulting time series with the RT which is represented by the function G representing the Brain to Behavior function. To address this, we need to aggregate the spatiotemporal signal to align it with the RT. In this study, we modeled this Brain-to-Behavior function using either correlation or the Elastic Net. This adds another layer of complexity, as the Brain-to-Behavior function is not fixed, turning it into a challenging optimization problem. Instead, we explore different values for c1 and c2, using correlation to evaluate their relationship with RT.

We have added this to the Discussion Limitation section 4.2 since we are not sure there is a straight forward manner to solve this problem, and might require an in-depth analysis in a future publication:

We have only tested the linear separability between the identified Rest and Task network models in this study. For future studies, non-linear separation schemes should also be considered and might provide a substantial boost in accuracy. However, this is difficult in practice; for example, consider a simpler special case an optimization problem of equation 6 rewritten as follows:

$$RT = G\left(\int \dot{X} - Q(\Xi_{Rest} * \theta(X)) + P(\Xi_{Task} * \theta(X))\right)$$

Solving this optimization is difficult, as the comparison to the RT coefficients needs another function G , which we implement as the Brain to Behavior Function. This function is approximated in two manners 1) using correlation and 2) using an Elastic Net. Since the relationship between Rest and Task are unknown as well as the Brain to Behavior function, it makes it difficult to solve for both the relationship as well as the Brain to Behavior functions directly from the data. We have limited our analysis instead of testing a linear relationship between the Rest and Task networks by simply using coefficients for P and Q and sweeping them across a range. In future studies, a more

complex technique may be able to address this optimization problem directly and test the nonlinear relationship between Rest and Task.

3. Regarding the results, although Fig. 5 shows higher R2 scores in the resting-task separated cases, it remains unclear why we can assume the separated signal represents the resting signal rather than noise. It is possible that the coefficient c_1 in Eq. 6 controls the magnitude of the noise, and after the signal separation in Eq. 6, the remaining signal may be noiseless. The author should provide additional evidence to clarify this point.

We thank the reviewer for bringing to our attention the interpretation of our algorithm with relation to denoising algorithms. As shown in the first figure below (right), the variance of the signal *increases* after separation, which would not be expected if the algorithm were simply performing a denoising of the timeseries. The covariance, or the FC matrices also show an increase in variance, which we have visualized with our UMAP in Figure 6. From the figure, the transformation is most analogous to subtracting a random signal with the same mean and variance as the ACM which increases the amount of noise in the signal. This was performed to match our separation transformations, but this is equivalent to adding noise, as the noise is drawn from a Gaussian process symmetric around the origin. Additionally, this increased variance aligns with an improvement in behavioral correlation to RT, as demonstrated in the first figure below (left). Another perspective that may clarify the separation process is the observed changes in functional connectivity, as illustrated in Figure 3 EF (second figure below). The network signal captures the significant resting-state variations it was trained on. Subtracting the network signal from the measured signal decreases functional connectivity by removing the components representing structured processes, leaving the unstructured components intact. Consequently, this transformation eliminates structured processes, increases signal variance, and which we observe aligns more closely with behavioral responses. The ACM highlights the importance of selectively removing unrelated structured processes, as only this targeted approach that we tested that enhances behavioral correlates.

Supplemental Figure 16- The Separation of the signal using various models. *M.* stands for Measured. The signal separation using both task and rest ODE, the Active Cortex Model (Rest - Task), is the only separation that yields significantly larger behavioral correlates ($p < \$0.05\$$) than the original Measured Signal. The matched signal represents subtracting the measured signal using a random signal with the same mean and variance as the Active Cortex Model. The figure on the left shows that the variance increases significantly ($p < 0.05$) for the Active Cortex Model after subtraction compared to the original and matched model.

Figure 3 EF- An example of separating the measured signal derivative into network and stimulus components normalized to have the same amplitude. F) The network structure is preserved in our model which has similar FC to the measured FC (~0.7 correlation) The residual signal which would be the result of our separation scheme has an increased variance and decreased relationship to the signal in other brain areas.

We have added this discussion about its relation to noise to the discussion section:

In relation to denoising algorithms such as ICA, PCA, filtering our model increases the signal variance unlike those transformations (see Supplementary Figure 16) (Hutchison

et al., 2013, Allen et al., 2014). These increases in signal variance are most similar to adding noise into the signal, but this increase in variance is related to improvement in behavioral correlates in regions corresponding to the Task. In addition, the previously mentioned techniques, as well as other approaches such as *FC*, do not represent a causal dynamical pathway unlike ODEs. However, these earlier methods are computationally inexpensive and are useful to calculate, when comparing to our approach. While the algorithm can execute within minutes, in its current form it does require a lot more data to train a valid model (see the next section for more information).

Reviewer #3 (Remarks on code availability):

The code generates results of the paper. It would be better for the author to add additional comments to the code and remove some unnecessary long output.

We have taken the reviewers suggestions into considerations and have reformatted our code such that it is more readable.

References:

Durstewitz, D.: A state space approach for piecewise-linear recurrent neural networks for identifying computational dynamics from neural measurements. *PLOS Computational Biology* 13(6), 1005542 (2017) <https://doi.org/10.1371/journal.pcbi.1005542> . Publisher: Public Library of Science. Accessed 2024-12-04

Koppe, G., Toutounji, H., Kirsch, P., Lis, S., Durstewitz, D.: Identifying nonlinear dynamical systems via generative recurrent neural networks with applications to fMRI. *PLOS Computational Biology* 15(8), 1007263 (2019) <https://doi.org/10.1371/journal.pcbi.1007263> . Publisher: Public Library of Science.

Champion, K., Lusch, B., Kutz, J.N., Brunton, S.L.: Data-driven discovery of coordinates and governing equations. *Proceedings of the National Academy of Sciences* 116(45), 22445–22451 <https://doi.org/10.1073/pnas.1906995116> . Publisher: Proceedings of the National Academy of Sciences. (2019)

Kashyap, A., Plis, S., Ritter, P., Keilholz, S.: A deep learning approach to estimating initial conditions of Brain Network Models in reference to measured fMRI data. *Frontiers in Neuroscience* 17, 1159914 (2023) <https://doi.org/10.3389/fnins.2023.1159914>

Cai, W., Ryali, S., Pasumarthy, R., Talasila, V., Menon, V.: Dynamic causal brain circuits during working memory and their functional controllability. *Nature Communications* 12(1), 3314 (2021) <https://doi.org/10.1038/s41467-021-23509-x> . Number: 1 Publisher: Nature Publishing Group. Accessed 2023-06-20

Reviewer #4 (Remarks to the Author):

This paper develops a new network ODE model for analyzing fMRI data during rest and task. The authors use ridge regression with a polynomial dictionary to construct network ODEs. The main finding and claim of the paper is that task-specific brain activity is a subset of the broader rest state network. By isolating task-related activity from rest-related processes, the model enhances the prediction of trial reaction times by 9%. This is formulated into a theory called the Active Cortex Model, which proposes that all cortical processes are active during rest, with specific subsets becoming more prominent during task execution.

I generally like the paper and enjoyed reading it. What I liked the most was the rigor in the computational modeling. Unlike most other works in this area that choose a model structure fairly arbitrarily and fit at most a couple of parameters, this work rigorously uses data to inform almost the complete structure and parameters of their ODE model.

What I also liked was the idea of subtracting rest model from task model. It is quite impressive that this can robustly and statistically significantly improve RT predictions.

We are grateful for the reviewer's appreciation for our work. We aim to address their concerns in the comments below.

We would also like to highlight that in addition to our point by point response below to the reviewers concerns, we have included for additional validation of the ACM, an analysis of classifying trajectories during each trial and shown improvement only using ACM (Supplemental Figure 19C). We also included another Figure (Supplemental Figure 21) on classifying two additional behavioral variables missing trials and accuracy based on ACM separation. Thus, we have demonstrated across a of total 14 subtasks and 5 behavioral variables that ACM allows for Task isolation. We have also shown in the revision that the changes in RT represent areas that are related to Task severity to clarify where and how these changes might localize in the cortex (Figure 5 and Supplemental Figure 20). In addition, we have rewritten the discussion sections relating to other work as well as expanded on the future direction and limitations section.

This brings me to a question/concern. Equation (6) shows the "separation" model

used in the paper, and Figure 4D shows the performance of this model for different values of c_1 and c_2 . It is fairly clear from Figure 4D that increasing both c_1 and c_2 while keeping their ratio the same (such as along the $c_2 = -c_1$ line) makes the performance better and better. This means the term \dot{X} in equation (6) is better to be removed (because multiplying both c_1 and c_2 by, say 2, is the same as multiplying \dot{X} by 1/2 and so on). This actually makes perfect sense. If you remove \dot{X} from equation (6), and set $c_1 = 1$ and $c_2 = -1$, the term inside the integral becomes (approximately) $\dot{X}_{\text{task}} - \dot{X}_{\text{rest}}$. After integration, this becomes $X_{\text{task}} - X_{\text{rest}}$, which is simply removing task data from rest data. Please clarify (with new results if needed) whether this simple model actually outperforms the ACM or not.

This is a very interesting observation by the reviewer. The integral resulting in Task – Rest does make a lot of sense when removing the measured timeseries out of the equation especially in conjunction that increasing c_1 and c_2 while keeping their ratio consistent does improve performance to a certain limit. However, this yields a much smaller correlation with the behavioral prediction as shown below (the proposed model is labeled as separated).

Changes to HRF and Max ROI correlation for Task - Rest

Changes to Correlations across significant ROIs for Task - Rest

While c_1 and c_2 can be increased, there is a limit to this, and there exists maximums for all Tasks that are not at infinity. For WM its roughly at $(2, -2)$, for relational its closer to $(1, -1)$. The other two are slightly off the ACM c_2, c_1 line. We have in our revision marked these maximums on Figure 4D in order to clarify that there exists a certain maximum.

D

The term \dot{X} in equation 6 represents the measured signal that we are trying to separate (the Venn diagram in Figure 4 or 2 might be really helpful in understanding the relation). Our understanding is that we modify the measured signal by removing task independent

processes that can be approximated as the difference between the Rest and Task ODE network activity (The Venn Diagram portion of Rest not contained in Task). However, the inclusion of the measured signal is crucial as it contains the information of the actual trial.

Other concerns and comments:

I'm a bit confused about what data was used to train each model. Is one model fit to data of all subjects, or one model per subject? Section 3.2.4 seems to imply that one model was fit for all subjects combined, but if so, then how can that group-level model predict RTs of individuals? Please clarify.

We thank the reviewer for bringing up this point as this might not have been clear. Yes, the Rest and Task models are trained using all the subjects (exception to this is the twin study which is trained using the twin subset and tested on the non-twin subjects). They represent the general processes that exist under the Task/Rest conditions and are represented by the coefficients which are fitted at the group level. However, in order to predict interpret individual fMRI on a per trial basis, we manipulate the measured timeseries using equation 6 by removing the group level non-Task processes (revised one as requested). The terms circled in red are from group level fitting, while those in blue represent the single individual trial run.

$$\int_{t_{stim}}^{t_{stim}+T} \dot{X}_{unsep}(t) - (c_1 * \Xi_{Rest} + c_2 * \Xi_{Task}) * \theta(X_{unsep}(t)) dt = X_{sep}(t) \propto RT \quad (6)$$

We believe that if we had individual coefficients for Rest and Task by fitting to individual Rest and Task fMRI, it would achieve better results, but since we do not have enough data using our current approach, we are limited to using group models to represent these processes. In the future, we would like to modify our approach to incorporate priors in order to account for individual variability as well.

We have added the following sentences in the Method Section 3.3.1 for additional clarity:

Although the coefficients Ξ_{Rest} and Ξ_{Task} are fitted from group level models, the timeseries X represents an individual trial (see Figure 8 below for more details) and contains the information of the performance during the trial. The goal is to isolate task-specific activity in individual trials by removing activity unrelated to the task, using group-level models to represent Rest and Task network processes.

The paper claims to have used SINDy, but that's problematic for two reasons. First, the authors have not really used SINDy, they have used simple ridge regression. The whole selling point of SINDy is sparsity. Here the authors explicitly say that sparsity promotion using L1 regularization didn't work, so they used L2 regularization. Why then call it SINDy? Second, SINDy is actually not a good modelling technique for brain dynamics. It is hardly a good modeling technique for any real-world system. Note that sparsity is a property of the coordinate system in which a dynamical system is represented, not an invariant property of any system. Take any sparse model, do a random linear (or nonlinear) change of coordinates, and with probability 1 the resulting representation of the same system isn't sparse anymore. So I'm not surprised at all that SINDy hasn't worked here, and I'm confused why the authors still want to call their approach SINDy.

The reviewer is correct that our method is not technically computing sparsity since we are utilizing L2 regularization. However, the SINDy package offers a wide range of tools in terms of appropriate denoising algorithms with respect to ODE identification, providing a flexible framework that can construct variety of different functions to use as candidate library functions, and providing different optimizers that allow for the incorporation of constraints. We agree that these algorithms are not all necessarily 'sparse' in a traditional sense but these options allows for a versatile number of different tools that have abetted our system identification. This methodology could have been implemented without the package, but since it has been rigorously tested and used, we opted to use these tools. However, using the SINDy package properly is not as simple as SINDy advertises, and requires a sophisticated understanding on how to represent the dynamics with appropriate library of equations and how to set up the constraints properly to get a solution that is biologically feasible.

In terms of the sparse coordinate system, the reviewer brings up a very interesting point. We agree that after an arbitrary coordinate transformation the system is not necessarily sparse. However, we believe there exists a real physical representation of this dynamical system in some form in the brain which is limited by resource constraints (total wiring in a constrained volume), and it must be sparse in terms of network connections. This has been shown to be true in *C. Elegans* but we suspect its true for any nervous system, and we use this notion to justify looking for a sparse network to represent the dynamics (Varshney et al., 2011).

We have added this to clarify in our methodology, the following sentences in the bottom of section 3.2.2:

Note that technically L2 regression is not sparse, and instead we employ ridge regression to determine the coefficients. While this is a misnomer when using 'Sparse Identification of Nonlinear Dynamics', the SINDy package offers much more versatile options for system identification than traditionally sparse-only solutions.

And the following sentences in the paragraph above that:

The EC is believed to be sparse in nature as there are few physical connections between the brain regions, as they are restricted by the amount of volume available inside of the skull. This has been shown in simpler systems such as the C. Elegans, but we believe this to be a general property of all nervous systems.

In Figure 4D, what is "Delta Max Correlation" on the color axis? I can't find a proper definition of it anywhere.

The 'delta max correlation' is calculated by first computing the correlation between the measured timeseries and the RT for across the task window for the unseparated data (from the starting of the stimulus to 14.4 seconds after, at a resolution of 0.72 TR for each ROI. The result is a 2D matrix representing correlations across space and time with the behavioral variable). This is the baseline correlation of that particular task at each spatial temporal point after stimulus onset and the maximum is the particular point in time and at a certain ROI that results in the highest correlation to RT. Then, we calculate the same value after the separation strategy for the same spatial/temporal window and display the change (delta) in max correlation in Figure 4D. The ROIs that these changes occur align well with the regions identified using previous GLM analysis for Task activation (see Supplemental Figure 15). We thank the reviewer for bringing this up, as we have not explicitly defined it in the text and have added to section 3.3.2:

To calculate the delta maximum correlation as shown in Figure 4D, we calculated the maximum correlation during the task window (computing the correlation for each spatial temporal point across all ROIs for the first 14.4 seconds after stimulus and taking the maximum) in the unseparated case as well as for each of the candidate separation schemes determined by the coefficients c_1 and c_2 . To determine the delta, we took the difference between the separated scheme and the unseparated case.

Figure 2 top left: I don't understand the Venn diagram. The equation below the Venn is also a bit confusing the way it is written, but later when you read equation (6)

We thank the reviewer in bringing up the concerns in Figure 2 the graphical abstract where we tried to simplify our methodology without losing clarity. The Venn diagram in Figure 2, shows the gray box, the Measured Signal with components of the Resting state Networks, Task Networks, and the Stimulus activity. We believe that the coupling of the

Stimulus and the Resting state networks can be represented as the Task model and thus we get the equation that the Stimulus plus noise is roughly equivalent to Measured – Rest + Task (gray area not in the stimulus or Resting State networks is assumed to be noise). This is the Active Cortex Model, that we believe to exist.

Equation 6 is our approximation on how to compute this relation with the group models and individual runs in the derivative space. Since we only model the Rest and Task components using ODEs, we have to rewrite this in differential and integral form which is shown in Equation 6. However, both equations are the exactly the same since you can integrate each individual component separately in Equation 6 which would result in the equation given in Figure 2.

We have clarified the difference in the caption:

The Stimulus Activity couples with the Resting State Networks to produce Task Network activity. To isolate the stimulus activity that is associated with the behavioral task, we use the measured timeseries and subtract our estimation for the Resting State Network Activity using an ODE and add back the Task Network Activity as shown by the Venn Diagram relation.

Figure 2 bottom right: It's not entirely clear to me that different task point clouds are more separate in the right panel than the left panel. Especially with a 3D plot, who knows, maybe if you rotate the left one there is a view angle that looks better. I think you need to come up with a quantitative measure (look at the unsupervised clustering literature) and compare it between the 2 panels, instead of just relying on eyeballing.

The UMAP results in Figure 2 are the same as Figure 6 where the differences are quantified using an SVM. Figure 2 and 6 both represent a 2D plot, with 2UMAP components so there should be no rotation possible. However, to make sure and quantify the results we utilized a simple SVM to test how well the points separate on the 2D plane to make sure that the ACM yields better separation between the four tasks. The separation shows an increase of about 20 percent accuracy in classifying the four tasks on the UMAP plane. We also tested the other separation schemes for the Rest and Task baseline model and show in Supplemental Figure 19 that only ACM separates out the FC matrices. For the revision, we also extended the separation for not only the FC, but also the individual per trail timeseries trajectories (bottom of Figure 19).

We have clarified in the captions as follows:

Bottom Right: By removing the Rest processes not present in the specific Task, the functional connectivity separates in high dimensional space providing more interpretability between distinct Tasks visualized using a 2D UMAP. Using an SVM, we

quantify a 20 percent increase in accuracy in the UMAP space between the FC unseparated and FC separated.

Figure 7: are these results on rest or task data?

Figure 7 is on resting state data, but Task data has similar results. We choose to keep the Hyperparameterization values that we determined using the Rest data consistent when constructing models using the Task data to easily be able to interpret the subtraction between them. We have added that information to clarify the Figure in the caption:

These calculations are performed for the Resting State ODE Model but the Task Models give similar results.

The notation in Equation (6) is a bit loose. Please update the writing of the equation to clarify what's integrated over (dt), the limits of the integral, and enclose the integrand in brackets to clarify what is exactly integrated.

We thank the reviewer for their suggestion and have incorporated their suggestions and have rewritten Equation 6 as follows. The integral is performed until the Task window T for a total of 14.4 sec after stimulus. The comparison to RT on the right is generated by correlating at every timepoint with the RT so we represent it using a proportion rather than an equal sign. We have clearly marked the relationship between using the unseparated timeseries and the separated timeseries.

$$\int_{t_{stim}}^{t_{stim}+T} \dot{X}_{unsep}(t) - (c_1 * \Xi_{Rest} + c_2 * \Xi_{Task}) * \theta(X_{unsep}(t)) dt = X_{sep}(t) \propto RT \quad (6)$$

References:

Varshney, L.R., Chen, B.L., Paniagua, E., Hall, D.H., Chklovskii, D.B.: Structural Properties of the *Caenorhabditis elegans* Neuronal Network. *PLOS Computational Biology* 7(2), 1001066 (2011) <https://doi.org/10.1371/journal.pcbi.1001066>
. Publisher: Public Library of Science. Accessed 2024-12-23

Reviewer #1 (Remarks to the Author):

I would like to thank the authors for taking the time to answer my previously raised concerns in detail. As of now, the only previous concern that remains unresolved is that of interpretability and novel insights.

To ameliorate this concern, the authors have performed an additional experiment contrasting ODE results for Working Memory under two different levels of difficulty (0-back vs. 2-back) and shown that such a contrast leads to more specific activations patterns as compared to (2-back vs. rest). Yet, that is what happens on a regular GLM framework; which brings me back to my original comment/question: what is it that we can learn about how the brain responds to a given task that other simpler methods cannot uncover? I believe that clearly exemplifying the power of this method for generating new knowledge beyond that of what more traditional approaches (which are much less computationally expensive and require much less data) can do would be critical for its successful adoption by the community.

We would like to thank the reviewer for the earlier discussion especially related to the previous work contextualizing our findings. We hope to address the remaining concerns about interpretability and novel insights as follows.

The novel contribution of this method lies in its use of an ODE model with biophysical properties to interpret the fMRI signal. It extends existing modeling frameworks by providing insight into how signals propagate through integration, enabling the interpretation of measured time series in terms of both network-driven and non-network processes. It has allowed us to test the association of different identified epoch-types in fMRI, namely Rest and Task, by constructing different ODEs through regression. While we can connect our ACM framework to earlier work that showed similar relationships, i.e Cole et al. (2017) for functional networks and Yan et al. (2021) for structural networks, having a dynamical causal framework allows us to separate and interpret the signal in terms of its network components, which other methods do not provide. We have shown that this separation increases variance, shows a higher HRF response in Task severity specific areas, and increases association across 5 behavioral measures and 14 subtasks.

The results using the Elastic Net (a linear model) on the unseparated data can be seen as an extension of the GLM approach to test brain to behavior association. Our model not only outperforms this framework, but does so in an informative manner, as it is guided by the constraints of an estimated biophysical model (as shown in Figure 4C). We believe that gaining insight and a novel understanding of brain dynamics, structure and behavior is best addressed by creating a single framework where all of these elements can interact in a causal manner. Only when all three of these properties are represented

simultaneously, can we hope to build a working model of the cortex in the future as the space for possible solutions is vast. This is our novel contribution, while elements of these have been modeled before, our method provides a unified framework that takes all of these components into account.

We have changed the conclusion paragraph in the introduction to reflect the argument given above:

The novel contribution to the field is in its use of an ODE model with biophysical properties to interpret the Task fMRI signal. To gain real insight into brain dynamics, structure, and behavior, it is best to address it by creating a single framework where all these elements can interact in a causal manner. Only when all three of these properties are represented simultaneously, can we hope to build a working model of the cortex in the future as the space for possible solutions is vast. This is our novel contribution, while elements of these have been modeled before, our method provides a unified framework that takes all of these components into account.

We have also rewritten the conclusion:

This work presents a novel contribution to the field by introducing an ODE-based model with biophysical properties to interpret Rest and Task fMRI signals within a unified, causal framework. While prior approaches have separately modeled aspects of brain structure, dynamics, or behavior, our method uniquely integrates all three. By grounding the framework in regression, we identify the governing dynamical systems for Rest and Task separately, allowing for a methodologically consistent and scientifically meaningful test of whether a linear combination of these models can explain the measured data. We show that assuming Task is a linear subset of Rest reveals distinct signatures in the isolated Task signal, including heightened hemodynamic response in regions of Task engagement, and increase in variance that have a stronger associations with a wide range of behavioral measures. This suggests the Active Cortex Model as a core principle of cortical dynamics, where the nervous system at Rest consists of all processes being active, and any Task activity is a subset of these processes. Having established this novel regression-based framework, future research can build on it by incorporating more complex components to further refine our understanding of brain function.

I have two minor new concerns. First, the manuscript contains numerous typos, repetitions, unformatted latex, etc. (see a few examples below), which not only clouds interpretation but makes one feel like the writing was quite rushed. Second, likely due to the number of issues raised by reviewers, the manuscript has grown not only in size, but also in complexity and technicality; perhaps making it less accessible to a broader audience. I think that any efforts to amend this could be clearly beneficial for the quality of the text and its impact.

Examples of typos, repetitions, etc.

! appears instead of < when reporting p-values

the only the ACM framework (repeated “the”)

Sometimes the authors refer to Working Memory as WM, sometimes is fully spelled. using model F using complex functions (repeated “using”)

open source open source Python (repeated “open source”)

Our results also align with prior models of Task using Generalized Linear Models (GLM) have assumed that (missing tat before have)

“It has limited the scope“ to “this has limited the scope”

Sometimes you use Metatasks, sometimes meta-task. The inconsistent use of hypens applies to other terms.

\$c_1\$, \$c_2\$ not rendered properly.

We thank the reviewer for providing examples that improve readability and adhere to standardized grammar and formatting. In addition to addressing the specific examples listed, we have also thoroughly proofread the manuscript.

To shorten the manuscript, we removed one figure—the neural abstraction pyramid—from the introduction, streamlined the introduction text, and relocated several paragraphs to the methods section for better focus and clarity. This structure allows readers to more easily understand the overall message and concept of the paper.

Reviewer #3 (Remarks to the Author):

The authors addressed my questions point by point. Below are my opinions regarding their responses and the paper:

We thank the reviewer for their insightful thoughts and discussion regarding our responses. We aim to address their concerns in detail in our point-by-point responses below.

1. The authors provide a justification for using a high-order polynomial model compared to alternative methods. They argue that the since the field is still relatively new, no existing approach fully replicates their results. They also discuss existing methods such as DCM, asserting that their approach offers superior scalability and correlation accuracy. Additionally, they reference deep learning-

based techniques and highlight current limitations in convergence and interpretability. While the response addresses my concerns, it lacks enough empirical and theoretical proofs for demonstration of why their chosen model outperforms other possible approaches.

We acknowledge the reviewer's concern over the empirical justification of our algorithm. Our approach to modeling fMRI combines the *network structure, dynamics, and behavioral variables* into single framework where all of these elements can interact in a causal manner. While other existing approaches have captured aspects of these elements individually, we believe that constraining the large solution space using all of them together enables the construction of more interpretable models of cortical dynamics.

A demonstration of this interpretability is provided by our experiment identifying dynamical systems corresponding to Rest and Task epochs, where we test for a potential linear relationship between them. We show that if Task is assumed to be a linear subset of Rest, the isolated Task signal shows an increase in variance, an increase in HRF in Task severity regions, and an increase in the association to 5 different behavioral measures and 14 subtasks. Since this unified framework across brain structure, dynamics and behavior is novel, we chose regression to identify each component. In this manner, the dynamical systems for each regime, the linear relationship between Rest and Task, as well as the brain to behavior function were all identified to construct the causal framework to link them together. After establishing this novel framework grounded in regression, future research can explore the impact of incorporating more complex components in place of the current elements.

We added this argument to the conclusion:

This work presents a novel contribution to the field by introducing an ODE-based model with biophysical properties to interpret Rest and Task fMRI signals within a unified, causal framework. While prior approaches have separately modeled aspects of brain structure, dynamics, or behavior, our method uniquely integrates all three. By grounding the framework in regression, we identify the governing dynamical systems for Rest and Task separately, allowing for a methodologically consistent and scientifically meaningful test of whether a linear combination of these models can explain the measured data. We show that assuming Task is a linear subset of Rest reveals distinct signatures in the isolated Task signal, including heightened hemodynamic response in regions of Task engagement, and increase in variance that have a stronger associations with a wide range of behavioral measures. This suggests the Active Cortex Model as a core principle of cortical dynamics, where the nervous system at Rest consists of all processes being active, and any Task activity is a subset of these processes. Having established this novel regression-based framework, future research can build on it by incorporating more complex components to further refine our understanding of brain function.

2. The authors acknowledge that Eq. 6 can be reframed as an optimization problem, allowing for a more formal approach to solving for the coefficients c_1 and c_2 . The primary challenge lies in incorporating the Brain-to-Behavior function G , which maps neural signals to reaction time (RT). They have expanded the discussion in Section 4.2 (Limitations), recognizing that future studies could explore nonlinear separation schemes and optimization techniques to refine this approach. From my perspective, the work still requires a more rigorous equation formulation and a more robust solver to make sure their choice of the method is proper and to make sure the result is convincing.

We appreciate the reviewer's interest and thoughtful insights on the topic of linear vs. non-linear separability. We completely agree that this is an important question; however, we believe it warrants dedicated research with its own depth and novel experiments, making it more suitable for future studies. As the reviewer has highlighted, a nonlinear test requires estimating the Brain-to-Behavior function which is unknown and it is not clear how to test Rest and Task nonlinearity independently, as these could arise from either the function G or the nonlinear combination of the two ODE models.

The linear combination of Rest and Task ODE is not meant to capture the full complexity of neural dynamics, but to serve as a unified framework of dynamics, structure and behavior to examine structural overlap between conditions. The models we construct for Rest and Task are derived by separating their respective epochs into structured and unstructured processes using an ODE-based framework. This structured component represents the dominant network dynamics captured during each condition. Since both models are established through regression, it is methodologically consistent — and scientifically meaningful — to test whether a linear combination of these fitted models can explain measured data.

In this context, the use of linear combinations provides a conservative and interpretable mechanism for evaluating the overlap between Task and Rest dynamics. If Task-related processes are indeed embedded within the structure of Rest, then the linear projection of one model onto another becomes a natural way to quantify and test that inclusion. Our results — including improved behavioral relevance after separating stimulus-dependent components — support this hypothesis.

We have included the above paragraph in the limitation section:

The linear combination of Rest and Task ODE is not meant to capture the full complexity of neural dynamics, but to serve as a unified framework of dynamics, structure and behavior to examine structural overlap between conditions. The models we construct for Rest and Task are derived by separating their respective epochs into structured and unstructured processes using an ODE-based framework. This structured component represents the dominant network dynamics captured during each condition. Since both

models are established through regression, it is methodologically consistent — and scientifically meaningful — to test whether a linear combination of these fitted models can explain measured data.

In this context, the use of linear combinations provides a conservative and interpretable mechanism for evaluating the overlap between Task and Rest dynamics. If Task-related processes are indeed embedded within the structure of Rest, then the linear projection of one model onto another becomes a natural way to quantify and test that inclusion. Our results — including improved behavioral relevance after separating stimulus-dependent components — support this hypothesis.

Regarding whether the separated signal genuinely represents resting activity or is merely noise, the authors argue that their method does not perform traditional denoising, as evidenced by an increase in both Functional Connectivity (FC) and signal variance after separation. To validate that the transformation is not simply removing noise, they compare their separated signal to a matched random signal with the same mean and variance as the Active Cortex Model (ACM). The results indicate that the separated signal preserves structured network properties, whereas subtracting a purely random signal does not. Furthermore, the authors compare their approach to ICA, PCA, and filtering techniques, which typically reduce variance by removing structured components. In contrast, their method increases variance while still improving behavioral correlates, reinforcing that the separation process is not merely noise removal. However, a remaining issue is the lack of statistical tests to rigorously validate their argument.

We thank the reviewer for highlighting the absence of a statistical test in our plot. We have now added the p-values to the supplemental figure, demonstrating that these distributions are statistically different. The p-values displayed are given with respect to the original measured signal, but are significant $p < 0.001$ for all comparisons except for Measured vs Measured-Matched in the left plot and Measured vs Measured – Rest in the right plot.

Figure 1 The Separation of the signal using various models. M. stands for Measured. The signal separation using both task and rest ODE, the Active Cortex Model (Rest - Task), is the only separation that yields significantly larger behavioral correlates ($p < 0.001$) than the original Measured Signal. The matched signal represents subtracting the measured signal using a random signal with the same mean and variance as the Active Cortex Model. The figure on the left shows that the variance increases significantly ($p < 0.001$) for the Active Cortex Model after subtraction compared to the original and matched model. The p-values displayed are given with respect to the original measured signal, but are significant $p < 0.001$ for all comparisons except for Measured vs Measured-Matched in the left plot and Measured vs Measured - Rest in the right plot.

Reviewer #3 (Remarks on code availability):

The authors provide a jupyter notebook to visualize their results appearing in the article. They also provide a file for data preprocessing. A readme is needed to understand how to use them to replicate the results.

We have added a README document explaining how to utilize the uploaded files, as per the reviewer's suggestion.